# QBOi El Niño-Southern Oscillation experiments: Overview of experiment design and ENSO modulation of the QBO

Yoshio Kawatani[1], Kevin Hamilton[3], Shingo Watanabe[2], Masakazu Taguchi[4], Federico Serva[5], James A. Anstey[6], Jadwiga H. Richter[7], Neal Butchart[8], Clara Orbe[9], Scott M. Osprey[10], Hiroaki Naoe[11], Dillon Elsbury[12,13], Chih-Chieh Chen[7], Javier García-Serrano[14,15], Anne Glanville[7], Tobias Kerzenmacher[16], François Lott[17], Froila M. Palmerio[14,18], Mijeong Park[7], Stefan Versick[16], Kohei Yoshida[11]

[1]Faculty of Environmental Earth Science, Hokkaido University, Sapporo, Japan
[2]Japan Agency for Marine-Earth Science and Technology, Yokohama, Japan
[3]International Pacific Research Center and Department of Atmospheric Sciences, University of Hawaii, Honolulu, USA
[4]Aichi University of Education, Kariya, Japan
[5]Institute of Marine Sciences, National Research Council (ISMAR-CNR), Rome, Italy
[6]Canadian Centre for Climate Modelling and Analysis (CCCma), Victoria BC, Canada
[7]U. S. National Science Foundation National Center for Atmospheric Research (NSF NCAR), Boulder, CO, USA
[8]Met Office Hadley Centre, Exeter, UK
[9]NASA Goddard Institute for Space Studies (GISS), New York, USA
[10]National Centre for Atmospheric Science (NCAS), University of Oxford, Oxford, UK
[11]Meteorological Research Institute (MRI), Tsukuba, Japan
[12] Cooperative Institute for Research in Environmental Sciences, University of Colorado Boulder, Boulder, Colorado
[13]NOAA/Chemical Sciences Laboratory, Boulder, Colorado
[14] Group of Meteorology, Universitat de Barcelona, Barcelona, Spain
[15]Barcelona Supercomputing Center (BSC), Barcelona, Spain
[16]Institute of Meteorology and Climate Research–Atmospheric Trace Gases and Remote Sensing (IMK-ASF) Karlsruhe Institute of Technology (KIT), Karlsruhe, Germany
[17]Laboratorie de Météorologie Dynamique (LMD), Paris, France
[18]CMCC Foundation - Euro-Mediterranean Center on Climate Change, Italy.

*Correspondence to*: Yoshio Kawatani (kawatani@ees.hokudai.ac.jp)

**Abstract**

The Atmospheric Processes And their Role in Climate (APARC) Quasi-Biennial Oscillation initiative (QBOi) has conducted new experiments to explore the modulation of the QBO by El Niño-Southern Oscillation (ENSO). This paper provides an overview of the experiment design and investigates the modulation of the QBO by ENSO using nine climate models used in QBOi. A key finding is a consistent lengthening of the QBO period during La Niña compared to El Niño across all models, aligning with observational evidence. Although several models simulate QBO periods that deviate from the observed mean of approximately 28 months, the relative difference between La Niña and El Niño remains interpretable within each model. The simulated La Niña periods range from 6.9 % to 42.5 % longer than those during El Niño, compared to an observed difference of approximately 27.2 %. However, the magnitude of this lengthening shows large intermodel differences. By contrast, even the sign of the ENSO effect on QBO amplitude varies among models. Models employing variable parameterized gravity wave sources generally exhibit greater sensitivity of the QBO amplitude to the presence of ENSO than those models using fixed sources. The models capture key observed ENSO-related characteristics, including a weaker Walker circulation and increased equatorial precipitation during El Niño compared to La Niña, as well as a characteristic response in zonal mean zonal wind and temperature. All models also simulate stronger equatorial tropical upwelling in El Niño compared to La Niña up to ~10 hPa, consistent with ERA5 reanalysis. These modulations influence the propagation and filtering of gravity waves. Notably, models with variable parameterized gravity wave sources show stronger wave forcing during El Niño, potentially explaining the shorter QBO period modulation in these models. Further investigation into the complex interplay between ENSO, gravity waves, and the QBO can contribute to improved model formulations.

**1 Introduction**

The familiar Quasi-Biennial Oscillation (QBO), characterized by alternating easterly and westerly prevailing stratospheric winds in the tropics, dominates the large-scale circulation of the tropical stratosphere. The QBO influences the stratospheric polar vortex by modulating the propagation of extratropical planetary waves, consequently impacting storm tracks and surface pressure patterns in mid-to-high latitudes (Baldwin and Dunkerton, 2001; Anstey and Shepherd 2014; Kidston et al., 2015). In the equatorial troposphere, the Madden-Julian Oscillation (MJO), a large-scale tropical cloud activity pattern, may exhibit enhanced activity during the QBO easterly phase at 50 hPa (Yoo and Son, 2015). Other QBO-related effects beyond the tropical stratosphere may include the modulation of the mid-latitude subtropical jet by off-equatorial secondary circulations, filtering of propagating gravity waves into the upper stratosphere and mesosphere that then affect the Semiannual Oscillation (SAO) near and above the stratopause and even acting to generate a mesopause QBO (MQBO) (Baldwin et al., 2001; Anstey et al., 2022). Indeed recent research indicates that the QBO facilitates troposphere-stratosphere coupling and influences a wide range of dynamical and chemical processes spanning the equator to the poles and from the surface to the mesopause (Anstey et al., 2022).

The QBO in equatorial prevailing wind has been observed (at least up to mid-stratospheric levels) for over seven decades (Naujokat, 1986). It is clear over this record that the QBO differs somewhat from cycle to cycle (e.g. Quiroz, 1981) and there have been efforts to try to see if the cycle-to-cycle variations may systematically depend on such factors as solar activity, volcanic eruptions or the El Niño/Southern Oscillation (ENSO) cycle of the tropical troposphere (Dunkerton, 1983; Geller et al., 1997; Salby and Callahan, 2000; Hamilton, 2002, Kane, 2004; Taguchi, 2010). It seems that the strongest empirical connection that has been observed is between the QBO period and the ENSO phase. Notably Taguchi (2010) analyzed radiosonde observations of zonal wind from 70 hPa to 10 hPa over Singapore (1.3°N) for the period 1953-2008. He found that QBO signals exhibit faster phase propagation during El Niño compared to La Niña conditions, along with a weaker QBO amplitude during El Niño. Yuan et al. (2014) largely confirmed these findings using radiosonde data from ten near-equatorial stations but noted that the ENSO influence on the QBO amplitude appears less robust than its influence on the QBO period.

The present paper will discuss our investigation of the dynamics of the ENSO-QBO connection, using simulations from comprehensive global atmospheric general circulation models (AGCMs). First, we very briefly review here some key aspects of the current understanding of the dynamics of the QBO and its representation in AGCMs. The QBO is believed to be primarily driven by atmospheric waves, predominantly gravity waves generated by tropical cumulus convection. Eastward and westward propagating gravity waves, excited by active convection in the equatorial regions, propagate vertically from the troposphere to the stratosphere. Upon dissipation, they deposit mean momentum into the background zonal wind, accelerating it and generating the westerly (eastward wind) and easterly (westward wind) phases of the QBO. Large-scale Kelvin waves and inertia-gravity waves contribute to the QBO westerly acceleration phase, while small-scale gravity waves primarily drive the easterly accelerations (Hamilton et al., 1999; Kawatani et al., 2010a, 2010b; Evan et al., 2012; Pahlavan et al 2021).

Some global climate models with moderate horizontal and vertical resolutions have been able to simulate the QBO by representing the effects of sub-grid scale non-orographic gravity waves through parameterization. However, despite advancements in observational techniques, our understanding of the gravity wave field in the tropical stratosphere remains limited, hindering our ability to fully determine their geographical distribution, temporal variations, and sources (e.g., Alexander et al., 2010). Consequently, non-orographic gravity wave parameterizations (GWPs) must rely on simplified physical assumptions, frequently assuming constant gravity wave sources and/or launch levels in space and time.

The Brewer-Dobson mean meridional circulation (BDC) in the stratosphere, ascends in the equatorial region, flows poleward toward both hemispheres, and descends at high latitudes. The equatorial upwelling within the BDC can slow down or even temporarily halt the downward propagation of the QBO phase as it descends from the upper to the lower stratosphere (Coy et al. 2020). Generally, shorter QBO periods are associated with stronger zonal wave forcing and/or weaker tropical upwelling, and vice versa (Dunkerton, 1997). Previous research suggests that tropical upwelling is actually stronger during El Niño (Randel et al., 2009; Calvo et al., 2010; Simpson et al., 2011). Therefore, it seems likely that the shorter QBO periods observed during El Niño result from increased wave driving of the mean flow accelerations (Schirber, 2015; Kawatani et al., 2019).

The Atmospheric Processes And their Role in Climate (APARC) Quasi-Biennial Oscillation initiative (QBOi), launched in 2015 (Anstey et al. 2022; Butchart et al., 2018), aims to compare the representation of the QBO in climate models and comprises five core papers in its first phase. QBOi was originally initiated as a project under SPARC (Stratosphere–troposphere Processes And their Role in Climate), which was later renamed APARC in January 2024. These papers focused on: (1) evaluating the QBO in participating AGCMs (Bushell et al., 2020); (2) investigating the QBO's response to a warming climate (Richter et al., 2020); (3) evaluating tropical waves and their forcing of the QBO (Holt et al., 2020); (4) investigating QBO teleconnections (Anstey et al., 2021); and (5) evaluating seasonal forecast prediction skills for the QBO (Stockdale et al., 2020).

For paper 2 (Richter et al. 2020), eleven climate models participated in a global warming experiment (doubled $CO_2$ and globally uniform +2K Sea Surface Temperature (SST) increase, and a quadrupled $CO_2$ experiment with a +4K SST increase). Consistent with previous studies (Kawatani et al., 2011, 2012; Watanabe and Kawatani, 2012; Kawatani and Hamilton, 2013), all models showed a weakening of the QBO amplitude in the lower stratosphere with global warming. However, significant variability existed among models regarding changes in the QBO period. Recent global warming experiments using climate models have produced mixed results regarding the QBO period under increasing $CO_2$ and SST conditions. Some models project a shorter period, others a longer period, and still others project no change or even the disappearance of the QBO in the warmer mean climate (Richter et al., 2020; Butchart et al., 2020; DallaSanta et al. 2021; Lee et al. 2024). This discrepancy may stem from differing assumptions regarding GWP in different models, in addition to varying responses of resolved waves, precipitation, and large-scale circulations like the BDC and Walker circulations (Richter et al., 2020). It is worth noting that the tropical Pacific circulation response to warming has an El Niño like pattern (Vecchi and Soden, 2007), while this is not the case for the extratropical circulation (Lu et al., 2008).

While almost all climate models incorporate non-orographic GWP to simulate the QBO, most fix the sources of parameterized gravity waves, implying that gravity wave activity remains unaffected by changes in SSTs and $CO_2$ concentration. However, some recent models utilize GWP with variable sources linked to cumulus convection, reflecting the real-world relationship between convective activity and gravity wave generation. Verifying the modulation of resolved and parameterized waves in a future climate is impossible. However, investigating how the simulated QBO is modulated in individual models under El Niño and La Niña conditions is feasible.

The MIROC model without non-orographic GWP has successfully simulated the QBO by utilizing higher vertical resolution (~300 m or 550 m) and a modified Arakawa-Schubert type cumulus parameterization (e.g., Kawatani et al., 2005; Watanabe et al., 2008; Kawatani et al., 2011). Kawatani et al. (2019; hereafter K2019) conducted an ENSO-QBO experiment using the MIROC-AGCM with 100-year integrations of both El Niño and La Niña conditions, which simulated a shorter QBO period during El Niño compared to La Niña, consistent with the observations of Taguchi (2010) and Yuan et al. (2014). K2019 found that in their AGCM equatorial upwelling associated with the BDC strengthens during El Niño, and gravity waves contributing to the QBO become more prominent due to increased precipitation directly above the equator. Analyses based on Transformed Eulerian Mean (TEM) equations revealed that the effect of gravity waves overcomes that of tropical upwelling, leading to a shorter QBO period during El Niño compared to La Niña. Furthermore, K2019 conducted spectral analyses of high temporal and spatial resolution satellite cloud observational data (CLAUS, TRMM), revealing that convective activity in the spectral domain with slow horizontal phase speeds (say less than about 10 m s$^{-1}$) is more pronounced during El Niño compared to La Niña – a characteristic also qualitatively simulated in the MIROC-AGCM. This modulation of convective activity results in the excitation of more slow-phase-speed gravity waves during El Niño, facilitating the QBO descent to the lower stratosphere. Additionally, the generally weaker Walker circulation during El Niño compared to La Niña creates favorable conditions for gravity waves generated in the troposphere to propagate into the stratosphere and effectively drive the QBO.

Conducting a common ENSO-QBO experiment across a range of QBO-resolving climate models could help elucidate the role of non-orographic GWP in driving the oscillation. This exercise would be beneficial for evaluating both GWP schemes and model responses to ENSO, particularly in terms of whether a model accurately simulates shorter QBO periods during El Niño compared to La Niña. While previous work (e.g., Richter et al., 2020) has highlighted large inter-model differences in QBO responses to climate warming scenarios due to divergent representations of non-orographic GWP, such future scenarios lack observational benchmarks. In contrast, the QBOi-ENSO experiments are informed by well-documented observational comparisons, particularly the observed shortening of QBO periods during El Niño compared to La Niña (e.g., Taguchi, 2010; Yuan et al., 2014). In this sense, the ENSO-focused experiments offer a scientifically more tractable approach to evaluating the role of GWP in the models' tropical stratosphere compared to warming scenario experiments.

Nine models participated in the QBOi-ENSO experiments and provided datasets. We aim to present three core papers for this project: (i) QBOi-ENSO experimental design and basic characteristics of ENSO modulation of the QBO (this paper); (ii) teleconnections of the QBO during El Niño and La Niña; and (iii) Madden-Julian Oscillation modulation associated with ENSO and the QBO.

The research groups in the QBOi-ENSO project conducted long, continuous model integrations with annually-repeating prescribed SSTs characteristic of either El Niño or La Niña conditions, following K2019. Although this approach does not fully capture real-world SST evolution (e.g., the SST field of a mature El Niño at the end of a calendar year directly transitioning to the developing phase of another El Niño in the following year), this simplified design ensures a diverse sampling of QBO phases relative to the annual cycle. Basing our prescribed SST anomalies on composites of numerous actual historical months enables us to compare our atmospheric simulations with observations under our perpetual El Niño and La Niña model runs.

As explained in the following section, some model groups were unable to provide the complete set of model data necessary for analysis, and this limits our ability to conduct a comprehensive analysis and comparison across all models. Detailed zonal-time spectral analyses of model fields, like those performed in K2019, remain a subject for future study. This paper, the first core paper of the QBOi-ENSO experiments, focuses on the representation of fundamental ENSO modulation of the QBO and discusses the possible roles of GWP in each model. This paper is structured as follows: Section 2 describes the model and experimental design. Section 3 examines ENSO modulation of the QBO and Section 4 discusses climatological differences in mean fields in our experiments. Section 5 investigates wave forcing and residual mean meridional circulations in El Niño versus La Niña. Finally, Section 6 presents a discussion followed by a summary and concluding remarks.

## 2. Model Description and Experimental Design

The experimental design for the QBOi phase-1 project, including model names, domain and resolution, information on non-orographic GWP, and requested output variables, is detailed in Butchart et al. (2018). Phase-1 of QBOi consists of five experiments. Experiment 1 is an AMIP-type simulation using observed sea surface temperatures (SSTs) for 1979–2009. Experiment 2 employs climatological annual cycles of SSTs, sea ice, and external forcings, while Experiments 3 and 4 explore global warming scenarios. Experiment 5 consists of seasonal prediction experiments with perturbed initial conditions. QBOi experiment 2 employed repeated annual cycles of SST, sea ice, and external forcings. The current ENSO-QBOi experiments follow the same framework as experiment 2 but incorporate annual cycle anomalies of El Niño and La Niña SSTs into the climatological annual cycles of SSTs.

The prescribed SST anomalies used in these experiments were derived using the same procedure outlined in K2019. El Niño SST anomalies were calculated as a function of the time of year using AMIP SST data from 1950-2016. We computed a composite SST anomaly for each calendar month averaged over all times classed as El Niño conditions. The same procedure was used to construct annually repeating La Niña SST anomalies. This ENSO state characterization follows the definitions used by the Japan Meteorological Agency (JMA). El Niño and La Niña months are selected individually for each calendar month (i.e., each January, February, …, December) based on the JMA definition. Monthly SST data are weighted by the NINO.3 SST deviation values and then averaged.

The result of this procedure is a composite SST anomaly of 1.92 K in January for the NINO.3 region, which corresponds to "moderately strong" El Niño conditions. To produce more pronounced effects on the QBOi model integrations, the calculated composite SST anomalies were amplified. In a change from the procedure employed by K2019, in the present study the El Niño composite anomalies were multiplied by a factor of 1.8 and the La Niña composite anomalies by a factor of 1.4. These scaling factors bring the peak composite SST anomalies closer to those observed during the most intense historical El Niño and La Niña events. This approach allows for long integrations of QBOi models under prescribed perpetual El Niño and La Niña conditions, while avoiding the introduction of other sources of interannual variability. Further details are provided in Supplementary Section S1.

Figures 1a and 1b show the annual mean composite SST deviations from climatology for El Niño and La Niña, respectively. Figures 1c and 1d depict the composite NINO.3 SST anomalies throughout the year for El Niño and La Niña, along with the maximum and minimum observed monthly values from 1950 to 2016. To illustrate the annual cycle clearly, a two-year period is displayed (simply repeating the same composite values). The El Niño anomalies are weaker during boreal summer and intensify during boreal winter (Fig. 1c). It is important to note that the applied procedure cannot fully capture the development, mature phase, and decay of all observed El Niño events, as the evolution of an event can span more than a year. Nevertheless, the simulated time evolution generally resembles that of real events, with El Niño amplitudes tending to peak during boreal winter. La Niña, on the other hand, does not exhibit such a clear standard seasonal development pattern (Fig. 1d). The SST anomalies used are confirmed to remain within the observed range of ENSO magnitudes, representing the upper end of past variability.

The monthly El Niño and La Niña SSTs used in the model integrations are generated by adding the composite monthly

SST anomaly to the monthly climatological SSTs used in the QBOi experiment 2 for each model. The imposed SSTs are the only difference between our El Niño and La Niña runs; other prescribed fields, such as sea ice and stratospheric ozone distributions, remain identical. The global average SST anomaly is +0.176 K for El Niño and -0.134 K for La Niña. These differences in global mean SST are significantly smaller than those in typical global warming experiments. For instance, global SST increases of +2 K (+4 K) with doubled (quadrupled) CO2 concentrations were utilized in QBOi experiments 3 and 4

(Richter et al., 2020).

Nine models participated in the QBOi-ENSO experiments: CESM1 (WACCM5-110L), EC-EARTH3.3, ECHAM5sh, EMAC, GISS-E2-2G, LMDz6, MIROC-AGCM-LL, MIROC-ESM and MRI-ESM2.0. For clarity and conciseness, we refer to these models in the text as CESM1, EC-EARTH, ECHAM, EMAC, GISS, LMDz, MIROC-AGCM, MIROC-ESM and MRI, respectively. The original model names are retained in figures and tables. Table 1 summarizes the model information.

Five models (EC-EARTH, ECHAM, EMAC, MIROC-ESM, and MRI) employed fixed sources of parameterized gravity waves, while three models (CESM1, GISS and LMDz) used variable sources. Launch levels for parameterized gravity waves varied across models, ranging from 450 to 700 hPa or 1000 to 100 hPa (see Table 1). Notably, MIROC-AGCM does not incorporate non-orographic GWP; therefore, the QBO in this model is driven solely by resolved waves. Note that, while MIROC avoids the arbitrariness involved with GWP, the T106 horizontal resolution (1.125°) in this model is insufficient to

represent small-scale gravity waves, which are thought to be particularly important for driving the QBO easterly phase (Kawatani et al., 2010a,b). Model integration periods also varied, ranging from 40 to 100 years. Climatological means were calculated using all available data. For example, in the case of GISS, which comprises three ensemble members with 30-year integrations, data from all members were analyzed separately and then averaged to create climatological fields.

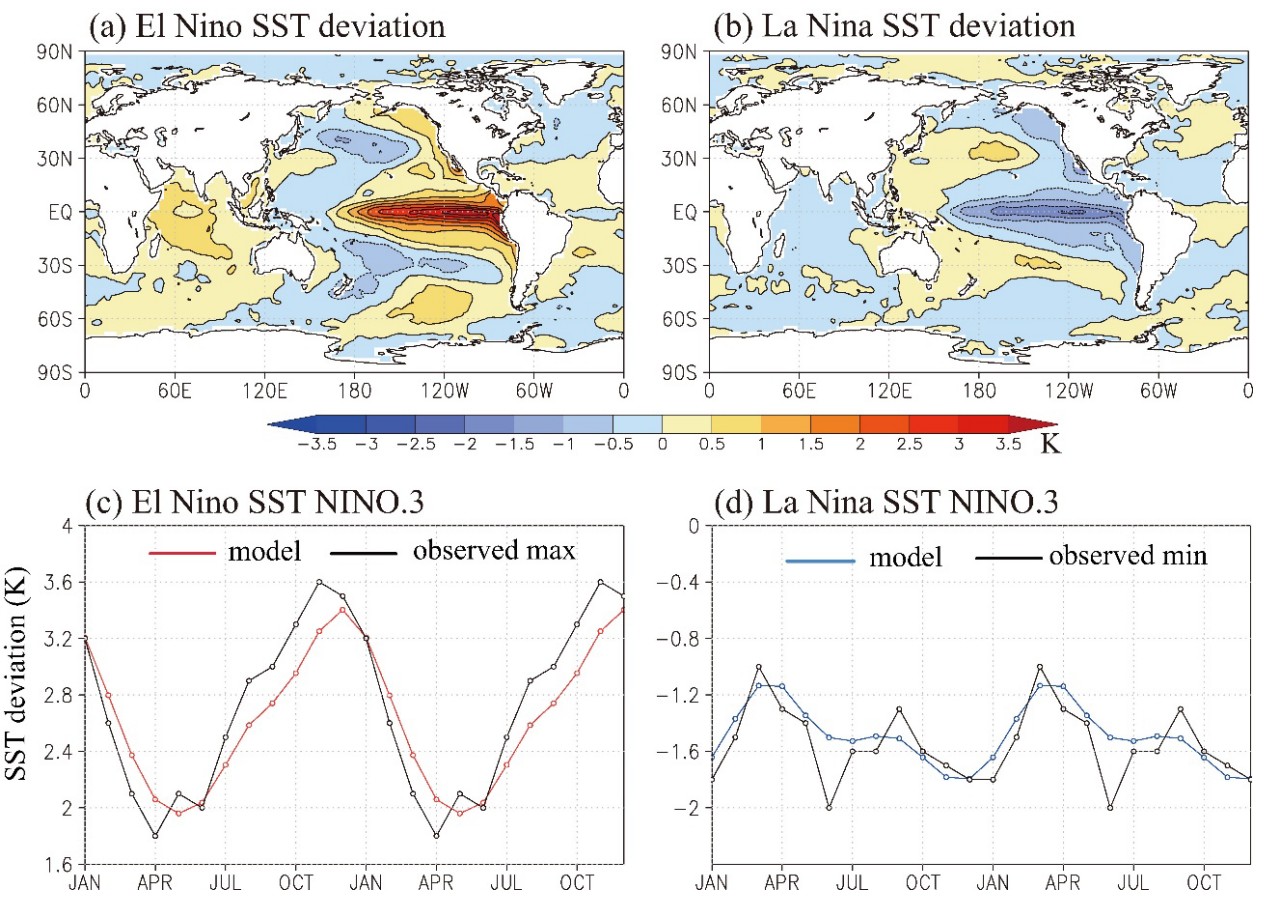


**Figure 1: Annual mean composite SST deviations from climatology for the (a) El Niño and (b) La Niña composites. The contour interval is 0.5 K. The annual cycle of NINO.3 (5°S-5°N, 90°W-150°W) SST deviation from climatology for our (c) El Niño (red line) and (d) La Niña (blue line) composites. Black lines represent the maximum/minimum observed monthly values during 1950-2016. For visualization, two full repeating cycles are shown.**

More detailed model information for CESM1, EC-EARTH, ECHAM, EMAC, LMDz, MIROC-AGCM and MIROC-ESM can be found in Butchart et al. (2018). In the QBOi-ENSO experiment, MRI-ESM2.0 was used instead of the MRI-ESM2 described in Butchart et al. (2018). MRI-ESM2.0 (Yukimoto et al., 2019) is an updated version of the model documented in Butchart et al. (2018), and it includes changes aimed at improving the modelled QBO (Naoe and Yoshida, 2019). Updated information about EC-EARTH3.3, compared to EC-EARTH3.1 used in QBOi phase-1 (Butchart et al., 2018; Stockdale et al.,

2020), can be found in Palmeiro et al. (2022) and Lott et al. (2024). While not included in QBOi phase-1, GISS joined the QBOi-ENSO experiments. This model incorporates non-orographic GWP, representing gravity waves generated by convection, shear, and deformation. Refer to Rind et al. (2020) for a more detailed description of the model configuration.

    The requested spatial and temporal resolution and output period (e.g., monthly, daily, 6-hourly three- or two-dimensional data) align with those outlined in Butchart et al. (2018). However, a limited number of new diagnostics were

added in the QBOi-ENSO experiment to facilitate the analysis of the Madden-Julian Oscillation (MJO) and the QBO-MJO teleconnection (c.f., Elsbury et al. 2025): (1) Daily variables: zonal wind, meridional wind at 200 hPa and 850 hPa, and outgoing longwave radiation (OLR) within the 30°S-30°N latitude band. (2) 3-hourly variables: OLR and precipitation. (3) 6-hourly data: temperature, zonal wind, meridional wind, and vertical wind within the 15°S-15°N latitude band, covering an altitude range of 150 to 0.4 hPa (compared to 100 to 0.4 hPa in QBOi phase-1, see Butchart et al. 2018).

While fundamental variables such as precipitation, zonal wind, and temperature were available for all models, some variables crucial for diagnosing QBO behavior were not available for some models. These include resolved wave forcing, parameterized non-orographic and orographic wave forcing, parameterized eastward and westward gravity wave fluxes, and residual mean velocity in the TEM equation (Table 2).

| Model name | Horizontal resolution | Δz (10-25 km) | non-orographic GW scheme | GWP sources | GWP launch level | year integration |
|---|---|---|---|---|---|---|
| CESM1 (WACCM5-110L) | 1.25°×0.94° | 500 m | Richter et al. 2010 | variable | 1000-100 hPa | 100-yr |
| EC-EARTH3.3 | T255 | 0.8-1.1 km | Scinocca 2003 | fixed | 450 hPa | 100-yr |
| ECHAM5sh | T63 | 600-700 m | Hines 1997 | fixed | 600 hPa | 40-yr |
| EMAC | T42 | 600-700m | Hines 1997 | fixed | 650hPa | 100-yr |
| GISS-E2-2G | 2°×2.5° | 0.5-1.0 km | Rind et al. 2007 | variable | 1000-100 hPa | 3×30-yr |
| LMDz6 | 2°×1.25° | 0.9-1.1km | Lott et al. 2012 | variable | 500 hPa | 82-yr |
| MIROC-AGCM-LL | T106 | 550 m | – | – | – | 100-yr |
| MIROC-ESM | T42 | 680 m | Hines 1997 | fixed | 650 hPa | 100-yr |
| MRI-ESM2.0 | T159 | 500-700 m | Hines 1997 | fixed | 700 hPa | 50-yr |


**Table 1. Models participating in the QBOi-ENSO experiment, including information on horizontal resolution, vertical level spacing (Δz) over 10-25 km altitude, references for non-orographic gravity wave parameterizations, whether parameterized gravity wave sources are fixed or variable, launch level of parameterized gravity waves and available model integration length (years). Note that non-orographic GWP is not used in the MIROC-AGCM-LL.**


    We will examine the climatological annual mean differences between the El Niño and La Niña runs across various fields. Emphasis will be placed on regions where these differences are statistically significant at a 95% confidence level. Statistical significance is determined using a two-sided Student's t-test, sampling the maximum individual yearly mean data (e.g., 100 data points for models with 100-year integrations) for both the El Niño and La Niña runs. For comparison, zonal

wind and temperature data from the ERA5 reanalysis dataset (Hersbach et al., 2020) and precipitation data from the CPC

Merged Analysis of Precipitation (CMAP; Xie and Arkin 1997) datasets are used. Observed El Niño and La Niña composites for each calendar month were computed using reanalysis data spanning 1979-2022. To ensure that each calendar month contributes equally to the annual mean, the same compositing procedure used for SST was applied. Specifically, El Niño and La Niña months were identified separately for each calendar month based on the definition provided by JMA, and monthly means were computed using all years satisfying each ENSO condition. These twelve monthly means were then averaged to obtain the annual mean. Unlike the SST calculation, no NINO.3-based weighting was applied when averaging ERA5 and CMAP data, in order to avoid disproportionately emphasizing months with large SST anomalies, which could bias the composite toward particular QBO phases. Importantly, the composite ERA5 and CMAP data were not scaled by factors of 1.8 and 1.4 for El Niño and La Niña, respectively, unlike the SSTs used as boundary conditions in the QBOi model simulations, which were multiplied by these factors. Consequently, the observed El Niño minus La Niña differences are expected to be smaller than those in the QBOi models. Nevertheless, these observational datasets remain valuable for evaluating the qualitative characteristics of the model differences, such as anomaly distributions.

For the analysis of QBO amplitude and period, long-term observational zonal wind data from equatorial radiosonde stations are used. These data were historically maintained and distributed by the Free University of Berlin (FUB) (Naujokat, 1986). Although the FUB dataset was discontinued in November 2021, the data provision has been continued by the Karlsruhe Institute of Technology (KIT). The full merged record covering January 1953 to December 2022 is used in this study.

| Model name | Resolved & non-orographic GWP forcing | orographic GWP forcing | eastward/westward GWP flux | Residual stream function | Residual vertical velocity |
|---|---|---|---|---|---|
| CESM1 (WACCM5-110L) | N/A | ✓ | ✓ | N/A | N/A |
| EC-EARTH3.3 | N/A | N/A | N/A | N/A | ✓ |
| ECHAM5sh | ✓ | ✓ | ✓ | N/A | ✓ |
| EMAC | N/A | N/A | N/A | N/A | N/A |
| GISS-E2-2G | ✓ | N/A | N/A | ✓ | ✓ |
| LMDz6 | ✓ | ✓ | N/A | ✓ | ✓ |
| MIROC-AGCM-LL | ✓ | ✓ | – | ✓ | ✓ |
| MIROC-ESM | ✓ | ✓ | ✓ | ✓ | ✓ |
| MRI-ESM2.0 | ✓ | ✓ | ✓ | ✓ | ✓ |

**Table 2. Model datasets used in this study (i.e. those available as of September 2024). Datasets can include zonal momentum forcing due to resolved and non-orographic GWP, zonal forcing due to orographic GWP, eastward and westward momentum fluxes of parameterized waves, residual meridional circulation stream function, and residual vertical velocity.**

## 3. ENSO Modulation of the QBO

This section discusses ENSO modulation of the QBO and climatological mean field differences between the perpetual El Niño and La Niña experiments. Figure 2 shows a time-height cross-section of the monthly and zonal mean zonal winds over the equator in the El Niño and La Niña simulations for each model. Red and blue colors correspond to westerlies and easterlies, respectively. For simplicity, only results from the first 20 years of each experiment are shown. QBO-like oscillations are found in all models. In the lower stratosphere, the westerly phase duration is generally longer in the La Niña

simulations compared to the El Niño simulations, particularly noticeable in EC-EARTH. The downward propagation of QBO
westerly and easterly phases to the lower stratosphere is more rapid during El Niño, which is a common characteristic among
the models.

In the ECHAM El Niño experiment, dominant zonal winds are westerlies above 20 hPa, easterlies at 20-30 hPa, and
westerlies around 30-50 hPa. Downward phases of easterlies and westerlies occasionally occur (around years 7-10 and 13-15).
These characteristics are not observed in the real atmosphere but are somewhat similar to those in their global warming
experiments (i.e., QBOi phase-1 experiments 3 and 4; see Fig. 4 in Richter et al., 2020). In contrast, the ECHAM La Niña
experiment simulates a much more realistic QBO with continuous downward phase propagation of both easterly and westerly
phases.

In the El Niño runs, GISS and LMDz simulate rather stable QBO phases extending to the lower stratosphere, but the
QBO in their La Niña experiments is more irregular, and westerly phases sometimes fail to propagate into the lower
stratosphere. In GISS, westerlies are continuously formed around 50 hPa. This suggests weaker zonal wave forcing in their La
Niña runs, as discussed later. Around 5-10 hPa, QBO westerly phases are much weaker in MRI, and QBO easterly phase
durations are much longer than those of westerly phases. Longer easterly phases of the QBO are also visible in CESM1.

Figure 3 presents the periods of individual QBO cycles in the El Niño and La Niña runs for each model's entire
integration period. For each simulated cycle, a period with an integral number of months is computed from the first month,
during which the monthly and zonal mean zonal winds at 20 hPa change from westerly to easterly, to the last month, defined
as one month before the next transition at 20 hPa. Note that GISS provided data for 30 years in each of its three ensemble
members, which are continuously drawn in Fig. 3 (i.e., ensemble 1 starts from months 1 to 360, ensemble 2 starts from months
361 to 720, and ensemble 3 starts from months 721 to 1080 on the abscissa). The values of mean QBO period ± one standard
deviation among QBO cycles are provided within each panel of Fig. 3.

The mean QBO period differs among models, and several models simulate QBO periods that fall notably outside the
observed range, which has a mean of approximately 28 months and varies from 18 to 34 months (Baldwin et al., 2001; Anstey
et al., 2022). In particular, MIROC-AGCM (16.6 to 19.7 months for El Niño and La Niña means) and MIROC-ESM (22.5 to
24.9 months) exhibit systematically shorter periods than observed. The mean QBO periods in both El Niño and La Niña runs
for EMAC and GISS are also somewhat shorter than ~28 months. While these models reproduce realistic downward
propagation of QBO phases (Fig. 2), the short simulated QBO period constitutes a structural limitation that should be taken
into account when interpreting the model results. Nevertheless, the primary focus of this study is on the relative differences in
QBO characteristics between El Niño and La Niña conditions within each model, rather than on absolute agreement with
observed QBO behavior. Accordingly, even models with biases in mean QBO period can still provide meaningful insights into
the modulation of the QBO if they produce internally consistent and interpretable differences between the two ENSO phases.

The simulated mean QBO period in the La Niña runs is longer than in the El Niño runs for each model, and the
percentage difference relative to El Niño periods is shown at the bottom of each pair of panels (e.g., 42.5% means that QBO
periods in La Niña are 1.425 times longer than those in El Niño).

Consistent with the observational study by Taguchi (2010), all models simulated longer periods during La Niña
compared to El Niño runs, a difference statistically significant at the ≥ 99% confidence level for each model (based on a two-
sided Student's t-test using the number of QBO cycles in each simulation as the degrees of freedom). However, the degree of
sensitivity of the QBO periods to ENSO differs among the models. Models with relatively large percentage La Niña versus El
Niño differences are EC-EARTH (42.5%), LMDz (27.9%), and ECHAM (24.6%), while those with smaller differences are
EMAC (4.3%), MIROC-ESM (6.9%) and MRI (8.5%). The analysis of observed near-equatorial winds by Yuan et al. (2014)
estimated long-term means for the QBO period of 25 months for El Niño conditions and 31.8 months for La Niña conditions,
corresponding to a 27.2% difference. Only three of the nine models (EC-EARTH, LMDz, and ECHAM) simulate La Niña–El

Niño differences in QBO period that approach this observed sensitivity, even under the amplified ENSO forcing used in this study.

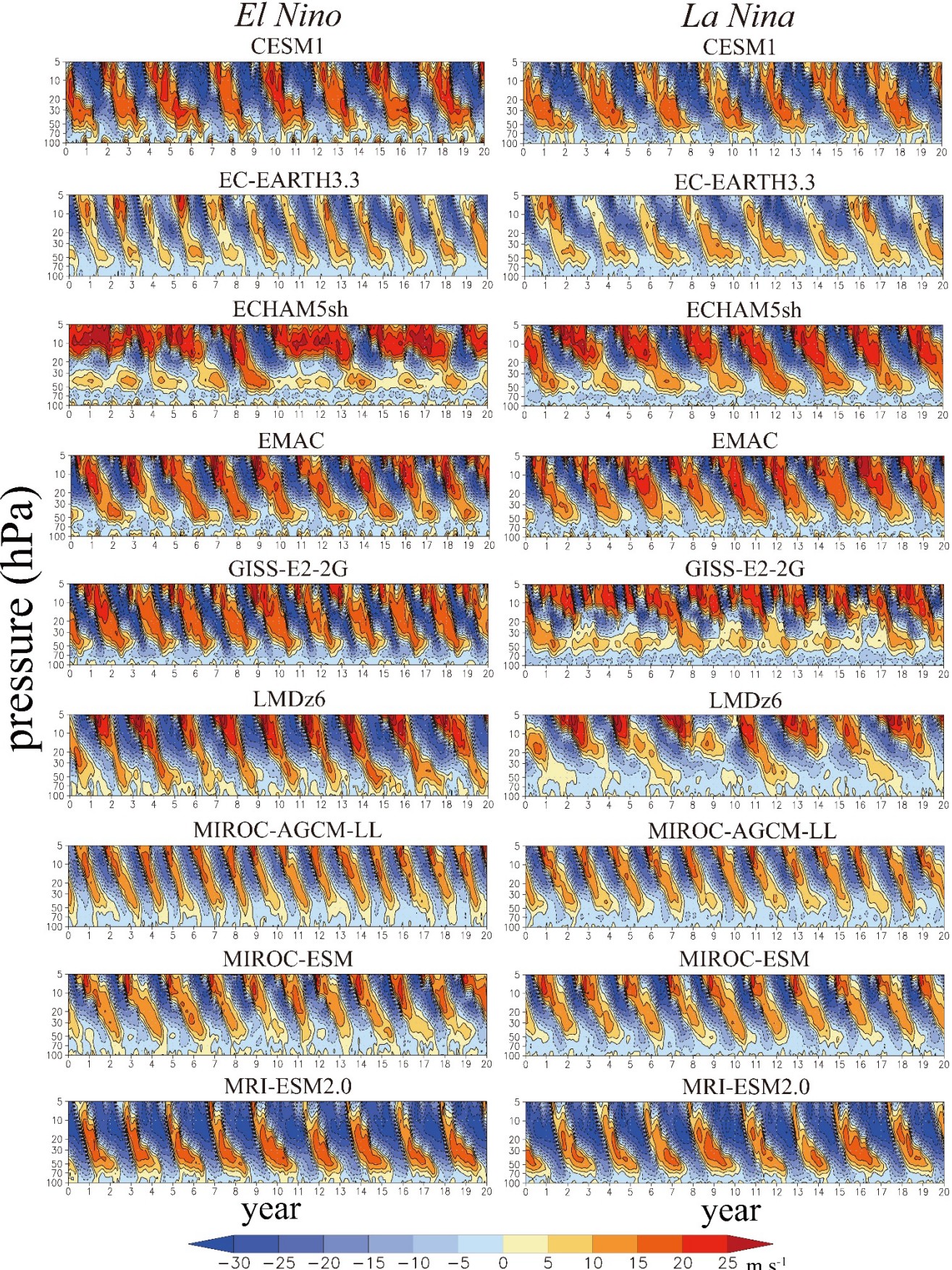

**Figure 2: Time-height sections of the monthly mean, zonal mean zonal wind over the equator in the (left) El Niño and (right) La Niña runs for each model. Results from the first 20 years of one ensemble member for each model are shown. Model names are noted above each panel. The contour interval is 5 m s⁻¹.**

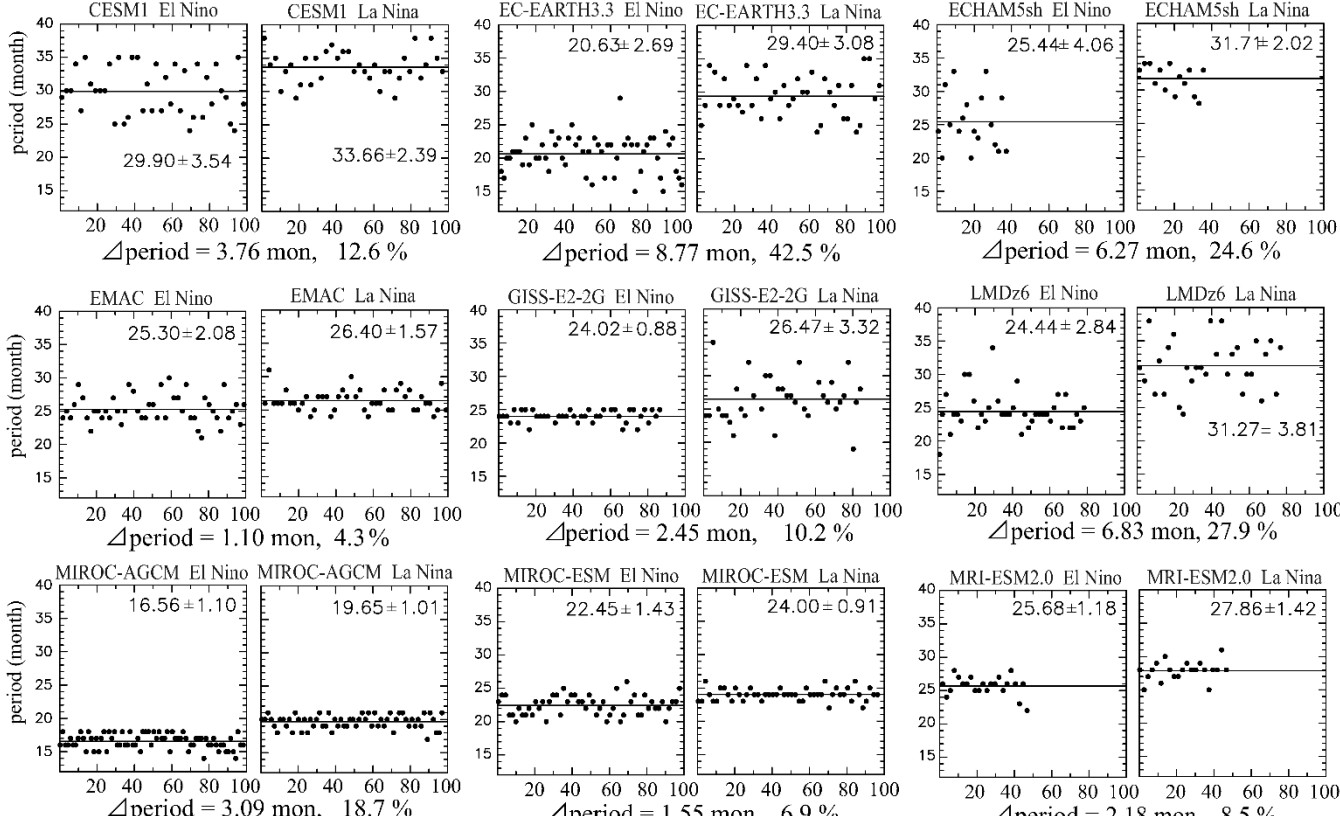

**Figure 3: Each pair of panels shows the time series of the QBO periods in the (left) El Niño and (right) La Niña runs for one model. Model names are noted above each panel. Solid horizontal lines show the mean QBO periods. Three ensemble datasets, each spanning 30-years, are used for GISS-E2-2-G model (i.e., the time series is not continuous around 30 and 60 years). One ensemble member is used for other models. Values of mean periods ± one standard deviation are shown within each figure. Differences in the QBO period between La Niña and El Niño, expressed in months, and as a percentage of the La Niña value, are shown at the bottom of each pair of figures. Differences of the mean QBO periods between El Niño and La Niña have statistical confidence levels ≥ 99% for all models.**

The cycle-to-cycle variability of the simulated QBO periods differs among models. The variability is relatively small in MIROC-AGCM, MIROC-ESM, and MRI. Seasonal locking of the QBO may preferably occur when its period is close to 2-year cycles, such as in GISS during the El Niño run and MIROC-ESM during the La Niña run. Cycle-to-cycle variability differences, evaluated from the standard deviation, are much larger (more than 30% larger) during El Niño in ECHAM, MIROC-ESM, and CESM1, while they are much larger during La Niña in GISS and LMDz, in which the QBO becomes more unstable during La Niña runs, as seen in Fig. 2.

In this context, it is also worth noting that a comprehensive evaluation of QBO period characteristics across multiple climate models participating in the QBOi project was conducted by Bushell et al. (2020). That study analyzed QBO periods in both QBOi Experiment 1 (AMIP-type simulations with observed SSTs) and Experiment 2 (simulations with climatological SSTs). Their results provide a broader reference for understanding how model formulation and boundary conditions influence simulated QBO periodicity. Readers interested in the model-dependent behavior of QBO periods across these different experimental designs are encouraged to consult Bushell et al. (2020) for further context.

Next, we consider the ENSO modulation of the QBO amplitude, which is known to be less robust than that for QBO periods (Serva et al. 2020). Following Dunkerton and Delisi (1985), at each level, we first calculate the standard deviation ($\sigma$) of the monthly mean time series after removing the mean seasonal cycle and then estimate the amplitude as $\sqrt{2}\sigma$. Figure 4 shows latitude-height cross-sections of QBO mean amplitude differences between El Niño and La Niña. Colored areas correspond to differences with a statistical confidence level of ≥ 95%, and contours show QBO amplitude in El Niño runs. ECHAM is an outlier model that shows a significantly weaker QBO amplitude during El Niño (i.e., negative differences) throughout the stratosphere, which might result from the unrealistic QBO in the El Niño run, making a simple comparison to other models difficult.

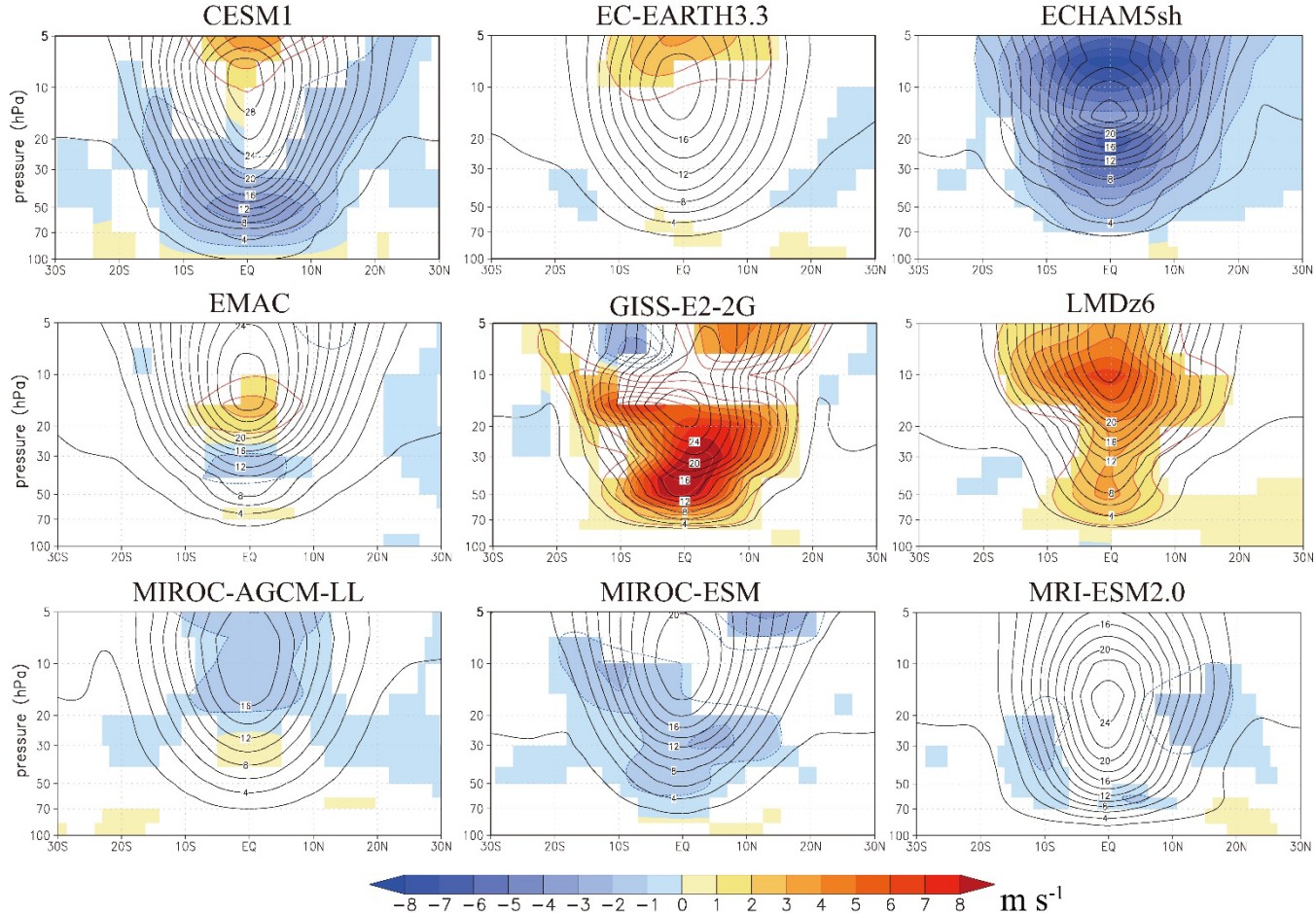

**Fig. 4: Contours show the latitude-height cross sections of the mean QBO amplitude in El Niño runs, with intervals of 2 m s⁻¹. The El Niño minus La Niña QBO amplitude differences are shown with colored shading with interval of 1 m s⁻¹. Color shading is shown only where the differences have a statistical confidence ≥ 95%.**

Except for ECHAM, the models showing large ENSO differences in QBO amplitude are GISS, LMDz, and CESM1, all of which have variable GWP sources. GISS and LMDz show larger amplitude throughout most of the stratosphere during El Niño compared to La Niña, while CESM1 shows weaker amplitude in the lower stratosphere and larger amplitude above 10 hPa during El Niño. Given that QBO amplitude depends on the zonal phase velocity of gravity waves in addition to wave source strengths, the discrepancy among these three models could be associated with zonal phase velocity differences in resolved and/or parameterized gravity waves. EC-EARTH, EMAC, MIROC-ESM, and MRI, which used fixed GWP sources, show relatively smaller differences. MIROC-AGCM, in which the QBO is driven only by resolved waves, also shows smaller differences, as shown in K2019. Due to its limited resolution (T106L72), MIROC-AGCM cannot fully capture the high-frequency, small-scale gravity wave spectrum that is typically represented by parameterized GWP schemes. As a result, much of the gravity wave effects that would influence the QBO remain unresolved.

To examine the influence of ENSO on different phases of the QBO, the method as described in Taguchi (2010) was applied. This approach is based on an empirical orthogonal function (EOF) analysis of equatorial zonal wind anomalies between 70 and 10 hPa, and identifies the two leading vertical modes that account for the majority of QBO variance. Profiles of monthly mean zonal wind anomalies are represented as linear combinations of these two fixed vertical patterns, with time-dependent coefficients describing the relative contribution of each mode. These coefficients serve as coordinates in a two-dimensional space that characterizes the state of the QBO at each time. The space is divided into four sectors corresponding to distinct QBO phases, including westerly and easterly regimes and their respective transition phases. From this representation, two scalar diagnostics are derived. One is the QBO amplitude, which is defined as the distance from the origin in the two-dimensional space. The other is the phase progression rate, which corresponds to the temporal rate of change of the phase

angle. These quantities are categorized by QBO phase and season. For the FUB/KIT observational dataset, only months classified as El Niño or La Niña are selected based on the NINO3 index provided by JMA. In contrast, all model simulations are conducted under perpetual El Niño or La Niña conditions, allowing straightforward compositing. A full description of the method, including mathematical details and physical interpretation, is provided in Taguchi (2010).

Figures 5 and 6 present composites of the mean phase progression rate and amplitude, respectively, classified by season and QBO phase at 50 hPa (see Fig. S1 for the QBO phase definition). Red shading indicates categories in which El Niño conditions are associated with faster downward propagation (Fig. 5) or stronger QBO amplitude (Fig. 6) compared to La Niña. The amplitude defined in the phase-space method reflects the overall QBO strength averaged over the 70–10 hPa layer and does not capture vertical or latitudinal variations. In contrast, Fig. 4 presents zonal-mean climatological QBO amplitude differences between El Niño and La Niña conditions as a function of latitude and height, highlighting spatially localized responses not represented by the phase-space diagnostics. These two approaches provide complementary perspectives. The phase-space method offers a compact representation of temporal evolution, whereas the latitude–height analysis provides a more detailed view of the vertical and meridional structure of the QBO modulation by ENSO.

The simulated QBO phase progression rates (Fig. 5) are interpreted in relation to the QBO period differences shown in Fig. 3. In the FUB dataset, phase progression is generally faster during El Niño, particularly in the W and WE phases. This behavior is consistent with the findings of Taguchi (2010), which were based on observations from 1953 to 2008. The present analysis uses the FUB/KIT dataset through 2022 and confirms that this characteristic remains robust throughout the full period. It is also noted that the FUB/KIT dataset is based on in-situ single station observations, while the model results reflect zonal mean zonal wind fields. Given the predominantly zonally symmetric nature of the QBO, this difference is not expected to substantially affect the comparison.

The model simulations show a range of responses in phase progression rate under ENSO conditions. EC-EARTH and LMDz show strong and systematic acceleration of QBO phase progression under El Niño, as illustrated in Fig. 5. This acceleration is consistent with the large differences in QBO period between El Niño and La Niña conditions shown in Fig. 3. Notably, EC-EARTH and LMDz show the largest percentage increases in QBO period during La Niña compared to El Niño, with values of 42.5 % and 27.9 %, respectively. MIROC-AGCM also shows robust acceleration of phase progression across all QBO phases and seasons, with uniformly positive differences, particularly during the MAM season. In contrast, MIROC-ESM, EMAC, and MRI-ESM exhibit minimal changes in phase progression rate between El Niño and La Niña conditions, consistent with the small QBO period differences seen in Fig. 3. CESM1 and GISS display intermediate behavior, with some phase- and season-dependent acceleration.

In the case of ECHAM5, slower (faster) phase acceleration during the E and WE (W and EW) phases may be explained by the distortion of its QBO structure in phase space under El Niño conditions. In Fig. 3, this model exhibits shorter QBO periods during El Niño on average; however, the distribution shows marked variability across categories, indicating substantial differences depending on phase and season. The EOF coefficients deviate markedly from a smooth elliptical pattern in phase space (not shown), resulting in a highly deformed or collapsed configuration. Equatorial zonal wind in this El Niño experiment (see Fig. 2) indicate that the westerly phase is weak around 20–30 hPa, and the easterly phase is vertically confined. Because the phase progression rate is calculated as the angular displacement from the origin in this space, such distortion likely leads to inaccurate or biased estimates of downward propagation speed. This structural inconsistency should be taken into account when interpreting the results from the ECHAM5 simulation.

Phase- and season-dependent modulation of QBO amplitude, as shown in Fig. 6, is the focus of the following analysis. Although this diagnostic does not resolve vertical structure, it reveals patterns that are not evident from the climatological fields shown in Fig. 4. Climatological differences indicate that GISS and LMDz exhibit stronger QBO amplitude during El Niño throughout the 70–10 hPa layer. This enhancement appears largely independent of QBO phase and season, as reflected in the uniformly positive values across most categories in the phase-space analysis. In contrast, ECHAM displays weaker

amplitude under El Niño conditions across the vertical domain, with a similarly consistent reduction across QBO phases and seasons.

MIROC-AGCM and MIROC-ESM both show slightly weaker amplitude during El Niño, with a broadly consistent
suppression pattern evident in many phase–season combinations. EC-EARTH, despite showing little climatological difference from 70 to 10 hPa, exhibits substantial amplitude variability depending on QBO phase and season. EMAC also shows limited climatological contrast, but the phase-space composite reveals stronger amplitude under El Niño in the WE and W phases and weaker amplitude in the E and EW phases. CESM1 exhibits weaker QBO amplitude during El Niño in the 70–30 hPa layer based on climatological fields, whereas the phase-space diagnostics show enhancement during the EW phase and suppression
in all other QBO phases. In contrast, MRI shows little difference in climatological amplitude overall, but the phase-space analysis reveals a clear phase dependence, with amplitude increasing during El Niño in the W and EW phases and decreasing in the E and WE phases.

These results highlight that, unlike the relatively uniform ENSO-related changes in QBO period, climatological amplitude differences vary considerably among models. Moreover, the phase-space diagnostics reveal that most models exhibit
QBO amplitude modulation with distinct dependencies on QBO phase and season, even when climatological differences are minimal. It is also important to recognize that some simulations, such as the El Niño runs in GISS and LMDz and the La Niña run in MIROC-ESM, produce QBO periods that are tightly clustered around 24 months (Fig. 3). This suggests the presence of seasonal locking. In such cases, the interpretation of phase–season composites requires caution, as the apparent seasonal dependence may be shaped by the regularity of QBO phase transitions. Furthermore, the QBO in the real atmosphere as
represented in the FUB/KIT dataset is influenced by year-to-year variability in sea surface temperatures. In contrast, the model simulations are based on perpetual ENSO forcing. This difference in experimental design limits the comparability between observations and models, a point considered further in the discussion section 6.1 below.

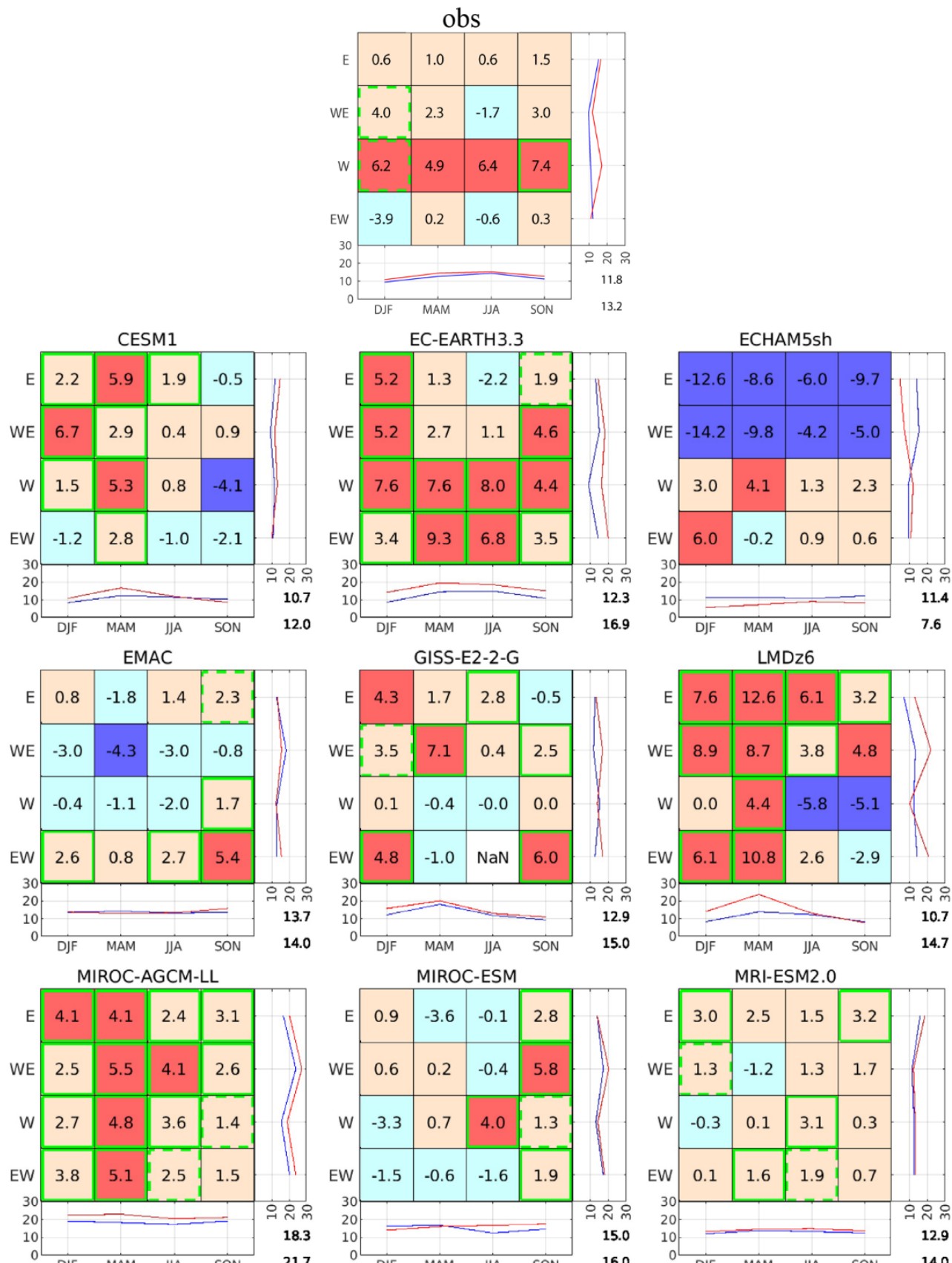

**Fig. 5: Composite differences in QBO phase progression rate (unit: degrees per month) between El Niño and La Niña conditions, classified by QBO phase at 50 hPa and season. The top-center panel shows results from radiosonde observations, and the remaining panels show outputs from nine QBOi models. Red (blue) shading indicates faster (slower) phase progression during El Niño. Green outlines denote statistically significant differences at the 90 % confidence level. Right and bottom subpanels show seasonal and QBO-phase averages, respectively. The two values in the lower-right corner indicate mean values for La Niña (top) and El Niño (bottom), and the line plot shows their distributions (blue for La Niña, red for El Niño).**

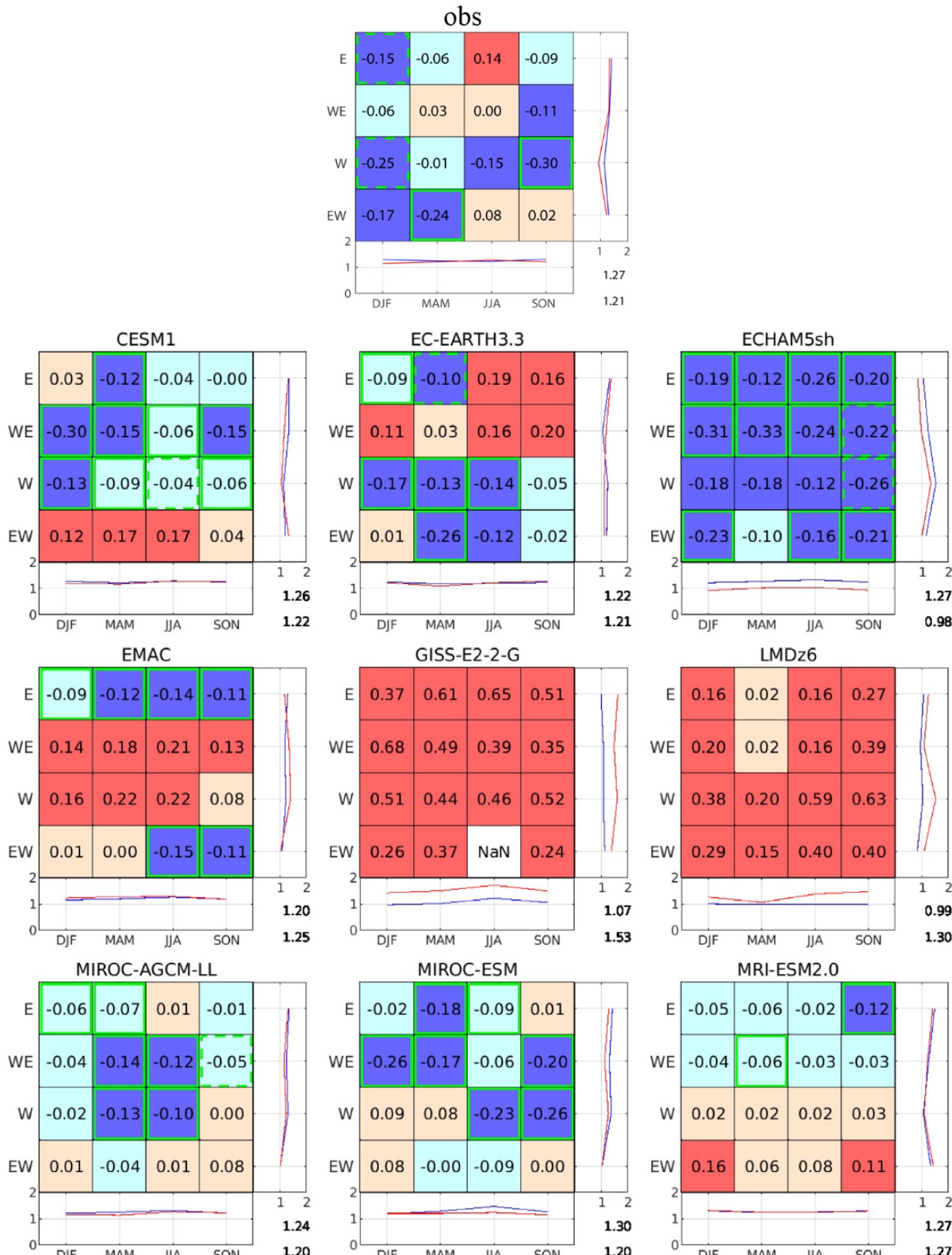

**Fig. 6: Same as Fig. 5, but for QBO amplitude, defined as the distance from the origin in the two-dimensional phase space constructed from the first two EOF components of equatorial zonal wind anomalies. Red (blue) shading indicates stronger (weaker) amplitude during El Niño. Mean values and composite lines follow the same format as in Fig. 5.**

## 4. Climatological Mean Field Differences

Next, we examine the long term annual mean differences between the El Niño and La Niña runs. Figure 7 presents a representation of the Walker circulation, namely longitude-height cross-sections of the long term annual mean, zonal wind at 10°N–10°S for the El Niño run, the La Niña run, and their difference (El Niño minus La Niña). Also shown are the same quantity computed from ERA5 fields composited over El Niño and La Niña periods. These ERA5 results will be discussed first and may be expected to show smaller El Niño minus La Niña differences compared to the QBOi models, since, as mentioned in Section 2, the QBOi simulations imposed SST anomalies based on composites scaled by factors of 1.8 and 1.4 for El Niño and La Niña. The ERA5 results in Fig. 7 show the expected Walker circulation differences between El Niño and La Niña conditions. In the upper troposphere both easterlies in the eastern hemisphere and westerlies in the western hemisphere are stronger in La Niña than in El Niño. Since the Walker circulation filters gravity waves propagating from the troposphere to the stratosphere, a weaker Walker circulation during El Niño could provide more favorable conditions for gravity wave propagation from the troposphere into the stratosphere due to reduced filtering (see Figs. 6 and 7 of Kawatani et al., 2010b). This argument assumes critical-level absorption of otherwise weakly damped, vertically propagating waves, similar to the ideal model of Lindzen and Holton (1968). Over the Pacific, around 120°E–90°W, significantly large easterly differences in El Niño minus La Niña are found from ~500 hPa to 80 hPa, and westerly differences are found below the region with easterly differences. At other longitudes above ~500 hPa, westerly differences are found, associated with smaller easterly differences below.

For both El Niño and La Niña conditions there is some degree of agreement in the Walker circulation in the ERA5 composite and in the QBOi model simulations. However, there is considerable variability in the Walker circulation among the individual model simulations. Most notably, in the ECHAM El Niño runs, the observed easterly jets in the eastern hemisphere are absent, while westerlies in the western hemisphere are much stronger than those in the ERA5 composite. One possible explanation for the unrealistic El Niño QBO simulation in this model, in which westerly phases do not propagate downward (Fig. 2), could be much stronger filtering of eastward waves within the troposphere. During El Niño conditions, the EMAC, GISS, LMDz, and MRI models also show relatively weak easterlies in the eastern hemisphere, while EMAC, GISS, MIROC-ESM, and CESM1 exhibit stronger westerlies in the western hemisphere. During La Niña, all models qualitatively represent one maximum easterly jet around 90°E and two maximum westerly jets around 140°W and 30°W.

Despite variations in the simulated Walker circulation structures between El Niño and La Niña across models, the El Niño minus La Niña difference patterns exhibit remarkable similarity across models and with ERA5. These results suggest that while the specific El Niño and La Niña circulations may be model-dependent, the ENSO-induced responses in the Walker circulation are well-represented in the QBOi models. All models except CESM1 exhibit stronger easterly differences over the central Pacific compared to westerly differences elsewhere. In CESM1, however, easterly anomalies over the central Pacific are much weaker, while westerly anomalies dominate at other longitudes.

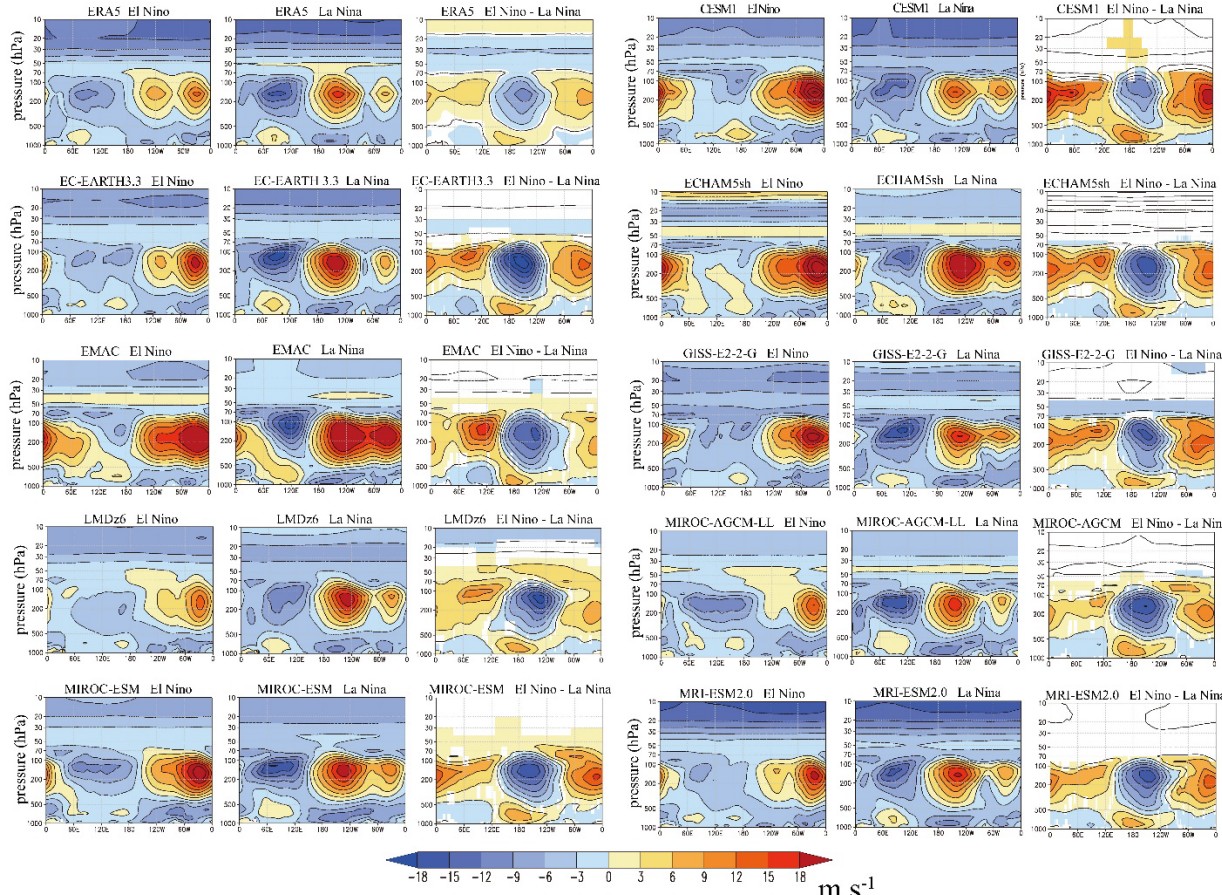

**Fig. 7: Each group of three panels shows longitude-height cross sections of the climatological annual mean zonal wind averaged over 10ºN–10ºS during (left) El Niño, (middle) La Niña, and (right) their difference. The top left row of panels shows results for ERA5 observations, and others show results from individual models. For the model results color shading is included where differences are judged to have a statistical confidence ≥ 95%. For the ERA5 observations shading is included where the magnitude of the difference**

**exceeds 1 m s⁻¹.**

Figure 8 shows climatological annual mean precipitation differences (El Niño minus La Niña) in CMAP data and in each of the models. Note that ENSO composites of the CMAP data are not multiplied by the 1.8 and 1.4 factors applied to the El Niño and La Niña SST anomalies imposed in the model experiments. All models show positive precipitation differences

(note that blue colors correspond to positive values) in the equatorial eastern Pacific and negative precipitation around the Maritime Continent, which are quite similar to those in CMAP. From the central to eastern Pacific a region of slightly reduced precipitation during El Niño conditions extends east to west, north of the equatorial positive differences (a feature clearly largest in MRI and CESM1). Note that this off-equatorial region of negative precipitation differences is related to the southward displacement of the Intertropical Convergence Zone (ITCZ) during El Niño as compared to the northward displacement during

La Niña (e.g. Trenberth et al. 1998).

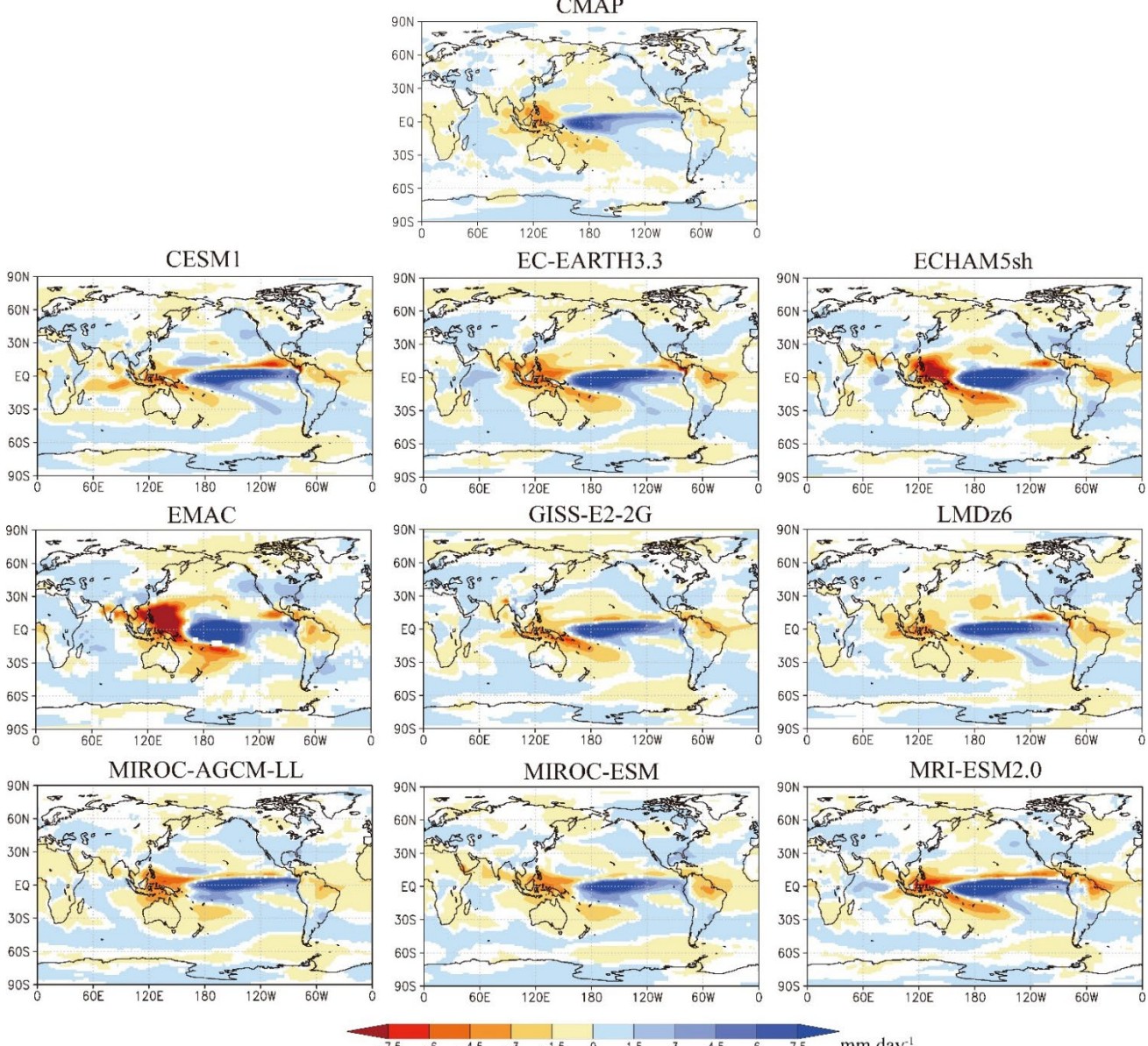

**Fig. 8. Climatological annual mean precipitation differences for El Niño minus La Niña conditions for (center-top) CMAP and (others) individual models. Differences with absolute values ≥ 0.1 mm day⁻¹ are shaded in CMAP, and those with a statistical confidence ≥ 95% are shaded for models.**

Figures 9a-c present longitudinal variations in El Niño and La Niña mean precipitation and their differences, averaged over 10ºN–10ºS. Longitudinal variations are qualitatively similar between observations and models in both El Niño and La Niña runs, although models tend to simulate the precipitation peak to the east of the observed one over the central Pacific in the El Niño run. The magnitude of the precipitation peak is also generally larger in the models than in observations, which

may reflect the amplified SST anomalies used in the simulations. The longitudinal variations of precipitation differences in 10°S-10°N are also qualitatively similar between CMAP and the models.

Figure 9d shows latitudinal variations in zonal mean precipitation differences for CMAP and each model. All models simulate larger precipitation during El Niño (i.e., positive differences) over the equator, consistent with observations. Note again that the observed composite here is being compared with model runs forced with amplified El Niño and La Niña SST

anomalies. So, for example, the differences in MIROC-AGCM-LL (yellow line) are larger than in CMAP in this experiment, while in the earlier K2019 experiments (without amplification of the imposed SST anomalies) the simulated ENSO differences in rainfall agreed well with the CMAP result. Larger precipitation over the equator in El Niño compared to La Niña is a favorable condition for generating equatorially symmetric waves, mostly Kelvin waves (e.g., Kawatani et al. 2009).

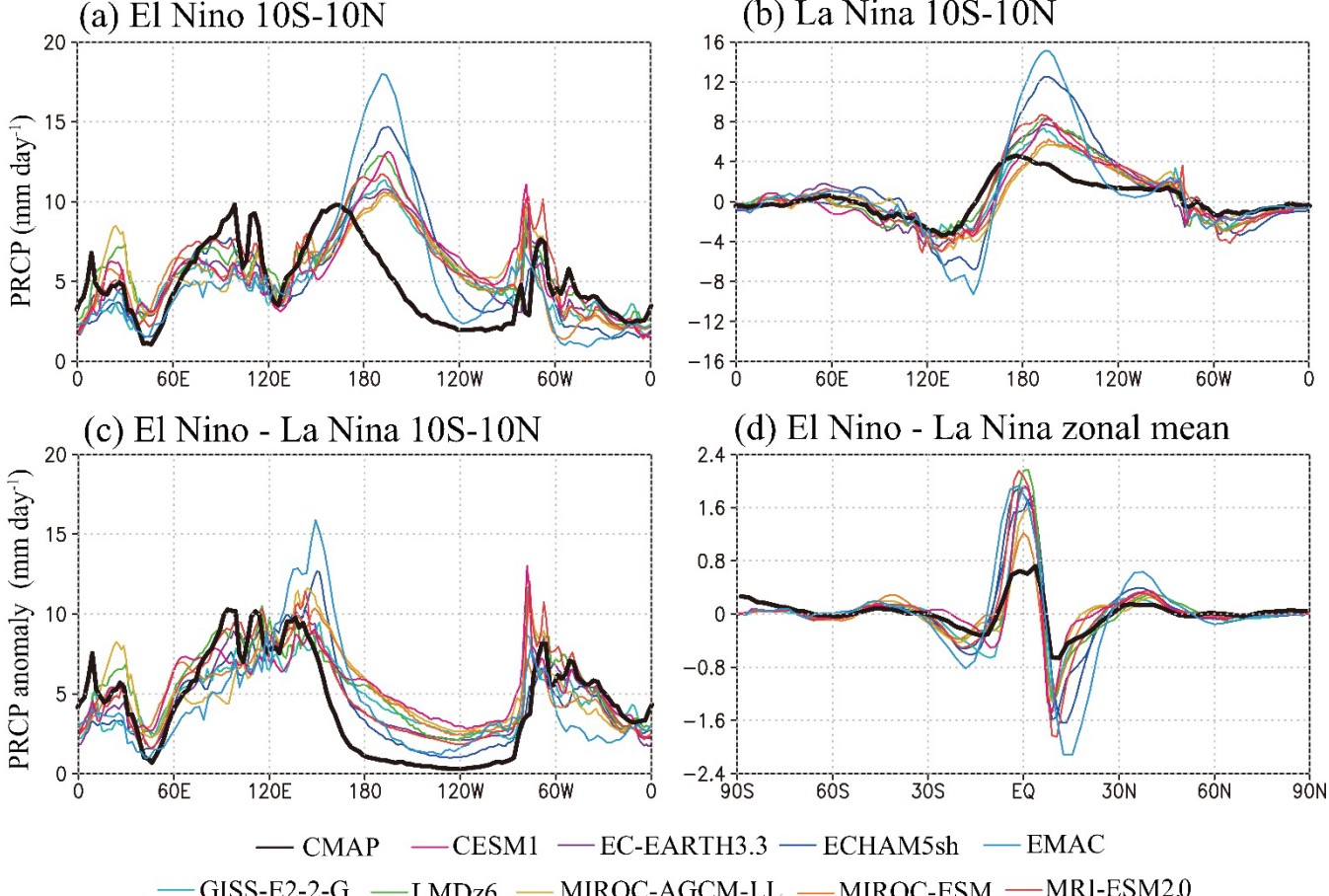

**Fig. 9. Longitudinal variations of annual mean precipitation averaged over 10ºN–10ºS in (a) El Niño conditions, (b) La Niña conditions and (c) their differences. (d) Latitudinal variations of zonal mean El Niño minus La Niña precipitation differences. Black line shows CMAP observations and color lines display results for the individual models.**

To contextualize the ENSO-related differences in wave forcing and stratospheric circulation discussed in the next section, Supplementary Section S3 (Figs. S2 and S3) presents the long term annual mean, zonal mean temperature and zonal wind in El Niño and La Niña conditions. In both ERA5 and all model simulations, the tropical troposphere tends to be warmer and the stratosphere cooler during El Niño compared to La Niña. Mid-to-high latitude stratospheric warming and an upward shift of the zero line for the zonal mean zonal wind are also consistently seen across most models, although the structure and extent of these anomalies vary among models. Notably, the enhanced westerly winds in the lower stratosphere at midlatitudes during El Niño resemble those seen in simulations of global warming, suggesting that similar dynamical mechanisms, such as altered wave propagation and mean flow interaction, are likely to contribute to changes in the Brewer–Dobson circulation.

**5 Contrasting Wave Forcing and Residual Mean Meridional Circulations in El Niño and La Niña**

In this section, we discuss the ENSO related effects on the wave forcing of zonal mean momentum and residual mean meridional circulation in the TEM formulation. The mean flow forcing from explicitly resolved waves as well as GWP are considered. The Eliassen–Palm flux (EP-flux) in spherical and log-pressure coordinates is used (Andrews et al. 1987):

$$F^{(\phi)} = \rho_0 a \cos\phi \left( \bar{u}_z \overline{v'\theta'}/\bar{\theta}_z - \overline{u'v'} \right) \tag{1}$$

$$F^{(z)} = \rho_0 a \cos\phi \left\{ \left[ f - (a\cos\phi)^{-1} (\bar{u}\cos\phi)_\phi \right] \overline{v'\theta'}/\bar{\theta}_z - \overline{u'w'} \right\} \tag{2}$$

$$\boldsymbol{\nabla} \cdot \boldsymbol{F} = (a\cos\phi)^{-1} \, \partial/\partial\phi \left( F^{(\phi)} \cos\phi \right) + \partial F^{(z)}/\partial z \tag{3}$$

The zonally averaged momentum equation in terms of the TEM formulation is expressed as:

$$\bar{u}_t = \bar{v}^* \left[ f - (a\cos\phi)^{-1} (\bar{u}\cos\phi)_\phi \right] - \bar{w}^* \bar{u}_z + (\rho_0 a\cos\phi)^{-1} \, \boldsymbol{\nabla} \cdot \boldsymbol{F} + \overline{OGW} + \overline{NOGW} + \bar{X} \tag{4}$$

In the above equations, $\rho_0$, $a$, $\phi$, $z$, $u$, $v$, $w$, $\theta$, and $f$ are the (height dependent) mean density, the mean radius of the Earth, latitude, log-pressure height, zonal wind, meridional wind, vertical wind, potential temperature, and Coriolis parameter ($f = 2\Omega \sin \phi$, where $\Omega$ is the rotation rate of the Earth), respectively. The subscripts $\phi$, $z$, and $t$ denote the meridional, vertical, and time derivatives, respectively. The mean residual circulations of the meridional and vertical components are expressed by $\bar{v}^*$ and $\bar{w}^*$. Eastward and westward resolved wave forcing of the mean flow correspond to the EP-flux divergence and convergence (i.e., $\nabla \cdot \boldsymbol{F} > 0$ and $\nabla \cdot \boldsymbol{F} < 0$), respectively. $\overline{OGW}$ and $\overline{NOGW}$ are the zonal forcing due to orographic and non-orographic GWP, respectively. The $\bar{X}$ term represents any other unresolved forcing including explicitly parameterized diffusion and any other contributions from the numerical schemes employed. These variables are not consistently available across all models, as shown in Table 2. Therefore, the figures in this section will only present data from models that include these specific variables. In ERA5, the EP-flux divergence is calculated using data with 0.5° horizontal resolution, 137 vertical levels, and 6 hourly output data. ERA5 provides parameterized zonal forcing, including $\overline{OGW}$ and $\overline{NOGW}$, as well as other source such as horizontal diffusion etc. In the region dominated by the QBO, most of parametrized zonal forcing is expected to arise from $\overline{NOGW}$.

To better understand the dynamical mechanisms underlying the ENSO-related modulation of the stratospheric circulation, Supplementary Section S4 (Fig. S4) presents the annual mean differences between El Niño and La Niña conditions for resolved and parameterized wave forcing and for the residual mean circulation for each model. In particular, westward forcing in the extratropical stratosphere associated with resolved waves, and westward forcing due to $\overline{OGW}$ are larger during El Niño in most models. These changes in wave forcing are accompanied by changes in the residual mean circulation, with models such as MIROC-AGCM and MIROC-ESM showing differences primarily in the shallow branch of the Brewer–Dobson circulation, while LMDz and MRI exhibit more vertically extended responses. The spatial structures and magnitudes of the anomalies vary among models and presumably reflect differences in wave parameterizations, but many of the key features are qualitatively consistent with ERA5.

To investigate the mean ascent in the equatorial lower stratosphere, we analyzed the residual vertical velocity in the TEM formation defined as:

$$\bar{w}^* = \bar{w} + (a\cos\phi)^{-1}\left(\cos\phi\,\overline{v'\theta'}/\bar{\theta}_z\right)_\phi \tag{5}$$

Figure 10 shows the vertical profiles of $\bar{w}^*$ during El Niño and La Niña, their differences (El Niño minus La Niña), and their ratio (El Niño divided by La Niña), averaged over 20°S–20°N for ERA5 and each model. This latitude band was chosen to reduce noise from the secondary meridional circulation associated with the QBO. The main conclusions are not sensitive to the choice of meridional averaging width.

All models simulate a local minimum $\bar{w}^*$ near 50 hPa in both El Niño and La Niña simulations, consistent with ERA5. However, the magnitude of this minimum, which is approximately 0.25 mm s$^{-1}$ in ERA5, varies considerably among the models, ranging from approximately 0.2 mm s$^{-1}$ in MIROC-AGCM to approximately 0.4 mm s$^{-1}$ in LMDz. In both the ERA5 and all the models there is stronger equatorial tropical upwelling in El Niño compared to La Niña, extending up to ~10 hPa. Specifically, large El Niño-La Niña differences are found around 100-50 hPa, ranging from 0.06 mm s$^{-1}$ (MIROC-ESM) to 0.15 mm s$^{-1}$ (GISS) (Fig. 10c), while in terms of fractional change the differences are as large as ~22% (LMDz) and even 40% (EC-EARTH). In the middle to upper stratosphere, the $\bar{w}^*$ differences become smaller, but LMDz and MRI show relatively large differences associated with the ENSO related modulation that extends to the deep branch of the BDC (Fig. S4). The differences seen in ERA5 are broadly consistent with the range of model results.

This inter-model spread in tropical upwelling may reflect differences in the strength of wave–mean flow interaction, which is critical for simulating the QBO with a realistic period (e.g., Kawatani et al., 2010a). While a detailed examination of this aspect is beyond the scope of the present study, it may partly explain model differences in QBO characteristics.

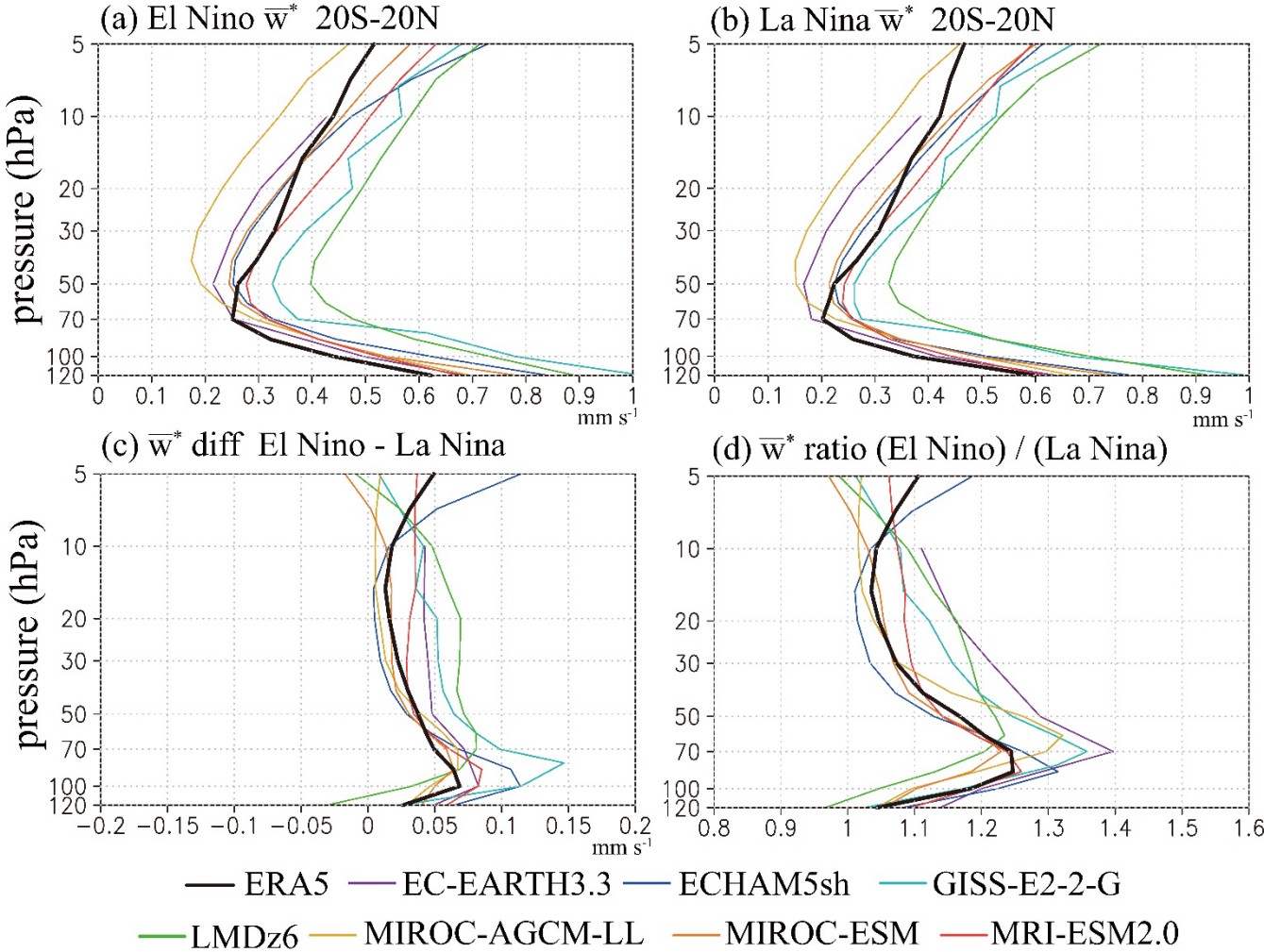

**Fig.10. Vertical profiles of residual vertical velocity averaged over 20°S-20°N in (a) El Niño experiment, (b) La Niña experiment, (c) El Niño minus La Niña values and (d) ratio of El Niño to La Niña values. The units for panels (a-c) are mm s⁻¹.**

If wave forcing relevant to the QBO remains constant, $\overline{w}^*$ could be the primary determinant of the QBO period (Dunkerton, 1997). The increase in equatorial $\overline{w}^*$ during El Niño runs, by itself, is expected to lead to a lengthening of the QBO period. However, it remains unclear which specific altitudes of $\overline{w}^*$ change have the strongest influence on the overall QBO period. Notably, in any event our results demonstrate significantly shorter QBO periods in El Niño simulations (Fig. 3), which is opposite of what would be expected simply from an intensification of the mean upwelling. Presumably the effects of the ENSO related changes in the wave driving of the QBO must be considered.

Next, we examine differences in parameterized wave fluxes between El Niño and La Niña. Relevant results are summarized in Fig. 11 by 4 panels for each model. The upper two panels for each model in Fig. 11 are 100 hPa horizontal maps showing both the absolute values of parameterized non-orographic gravity wave momentum flux during El Niño (contours), as well as the differences between El Niño and La Niña (color shading). Results are shown separately for the eastward and westward propagating parameterized gravity waves (left and right panels, respectively). The lower panels present the longitudinal variations of the absolute values of eastward and westward 100 hPa momentum fluxes averaged over 10°S-10°N during El Niño and La Niña. Datasets from only four models: ECHAM, MIROC-ESM, MRI-ESM, and CESM1, provide these variables (see Table 2). Note again that ECHAM, MIROC-ESM, and MRI-ESM used Hines-type non-orographic GWP, with fixed sources of parameterized gravity waves, but wave fluxes are modulated by filtering effects of background winds above the launched level. CESM1 utilized variable parameterized gravity wave sources, related to convective heating.

The Walker circulation plays a significant role in filtering gravity waves propagating from the troposphere to the stratosphere (e.g., Kawatani et al., 2009, 2010b). Eastward propagating gravity waves are preferentially filtered by background

eastward winds (i.e., westerlies), and westward waves by westward winds (i.e., easterlies). The Walker circulation in the middle to upper troposphere is easterly in the eastern hemisphere and westerly in the western hemisphere (Fig. 7), creating favorable conditions for eastward waves propagating upward in the eastern hemisphere and for westward waves in the western hemisphere.

In ECHAM, MIROC-ESM, and MRI-ESM, eastward fluxes during El Niño reach a maximum near the equator from ~30°E to ~120°W, while westward fluxes are at their minimum over the equator at these longitudes (see contour lines in Fig. 11). This finding corresponds well with the middle to upper tropospheric easterlies associated with the Walker circulation during El Niño (see the leftmost panel in each model in Fig. 7).

        Longitudinal variations in eastward and westward fluxes (see the lower panels for each model in Fig. 11) are
qualitatively similar across these three models for both El Niño and La Niña. Eastward fluxes are weaker from ~0°E to ~150°E and stronger from ~150°E to ~120°W during El Niño compared to La Niña. Conversely, westward fluxes are weaker from ~140°E to ~110°W and stronger at other longitudes during El Niño compared to La Niña. Consequently, differences in eastward fluxes (El Niño minus La Niña; see colors in the upper panels) are particularly large and positive around the central Pacific, associated with negative anomalies to the east and west. In contrast, differences in westward fluxes are largely negative
around the central Pacific. These characteristics are well correlated with background zonal wind differences associated with the Walker circulation (Fig. 7) in these three models (which all use Hines-type GWP).

        The distribution of parameterized gravity wave fluxes in CESM1, and their differences between El Niño and La Niña, differ from other models and exhibit more locally distinct structures. This should be because non-orographic gravity wave sources in CESM1 are related to parameterized convective heating. Differences in wave fluxes seen at 100 hPa are a result of
both variable wave sources and filtering caused by the Walker circulation. During El Niño, eastward fluxes peak around the central Pacific, associated with maximum precipitation (compare Fig. 9a with the red lines in the lower panels of Fig. 11). In contrast, maximum westward fluxes shift to the eastern Pacific. During La Niña, the maximum of both eastward and westward fluxes shifts to the western Pacific, corresponding to the precipitation shift (compare Fig. 9b with the blue lines in the lower panels of Fig. 11).

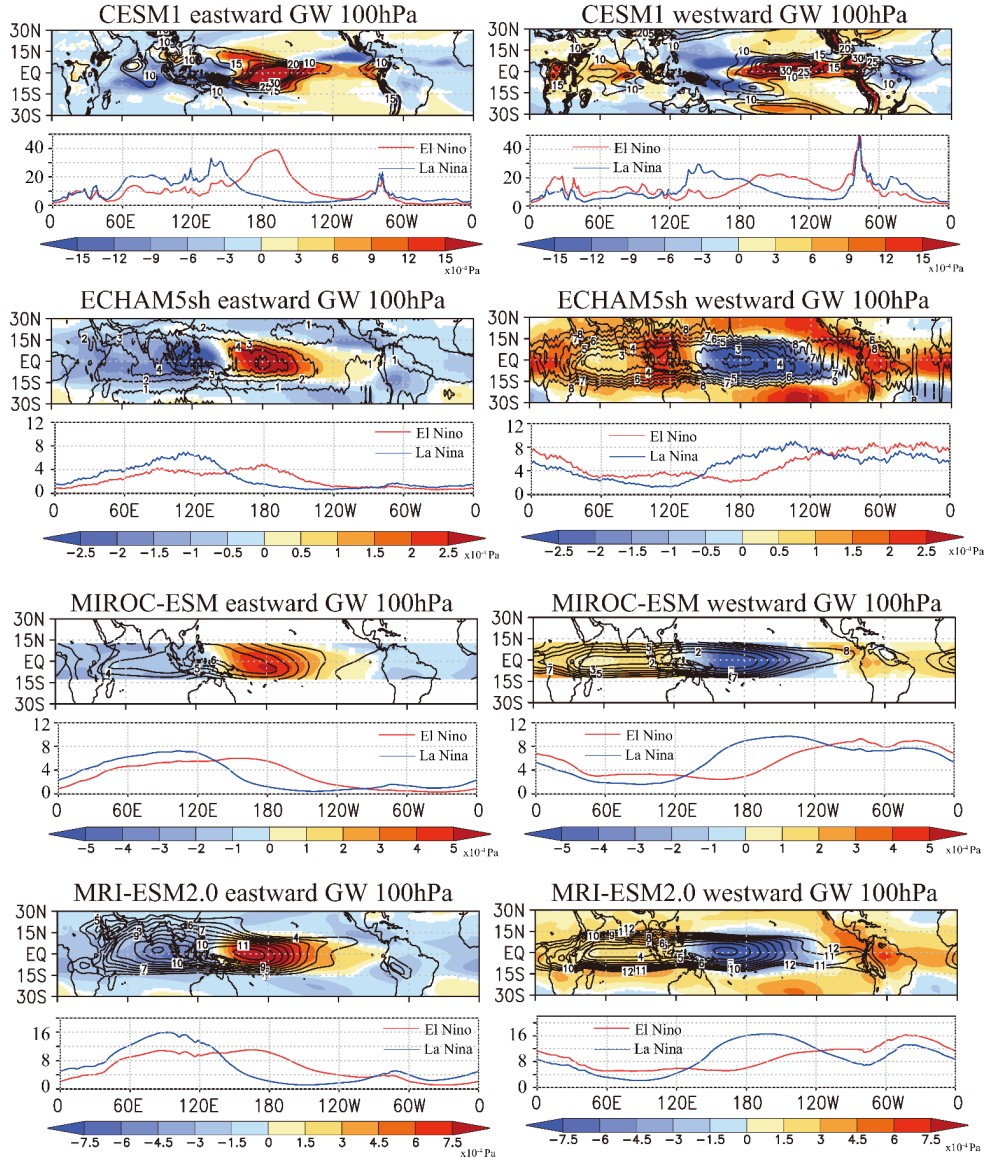

**Fig. 11. Results for the parameterized non-orographic gravity wave 100 hPa fluxes of zonal momentum in four individual models. The left (right) columns are for eastward (westward) waves. The upper panels for each model are maps showing the absolute value of the fluxes in the El Niño experiment (contours) and the El Niño minus La Niña value (color shading). Contour intervals are $1\times 10^{-5}$ (Pa) except CESM1 which are $5\times 10^{-5}$ (Pa). Color intervals are 0.5, 1, 1.5 and $3\times 10^{-5}$ (Pa) for ECHAM5sh, MIROC-ESM, MRI-ESM2.0 and CESM1, respectively, and color shading is shown only where the difference is significant with statistical confidence $\geq$ 95%. The lower panels of each model show the longitudinal dependence of wave fluxes averaged over 10°S-10°N in (red) El Niño and (blue) La Niña experiments. Note that the vertical axis scales differ among the panels.**

The El Niño minus La Niña differences in eastward fluxes show positive values from the central to east Pacific (see colors in the upper panels of Fig. 11), where maximum easterly anomalies of the Walker circulation are located (Fig. 7). Negative values correspond to areas of westerly anomalies of the mid-to-upper tropospheric Walker circulation at other longitudes. These differences are somewhat similar to those found in ECHAM, MIROC-ESM, and MRI. In contrast to eastward waves, differences in westward fluxes in CESM1 have large positive values in the eastern Pacific associated with positive precipitation anomalies (see Fig. 8). V-shaped negative anomalies are found around 150°E, also associated with negative precipitation anomalies. In CESM1, as well as in other models, the point of confluence between the ITCZ and the South Pacific Convergence Zone (SPCZ) shifts westward during La Niña compared to El Niño, consistent with observations (not shown).

While output data of parameterized gravity wave fluxes in LMDz were not available at the time of this analysis, this model, which also uses variable parameterized wave sources related to precipitation activity, showed similar structures affected by precipitation distributions (Dr. Lott, personal communication). Variable sources of parameterized non-orographic gravity wave fluxes in the equatorial region likely have a significant influence on both precipitation and Walker circulations, potentially impacting the representation of the QBO. A detailed investigation of the three-dimensional distributions of parameterized wave fluxes modulated by ENSO, including model dependence, would be of interest and remains a topic for future research.

Next, we discuss wave forcing of the mean flow in QBO composite cycles for El Niño and La Niña runs. The composite was defined based on the phase of the zonal wind QBO. Month zero of the composite is taken to be when the zonal mean wind at 20 hPa in the deseasonalized and smoothed (5-month running mean) zonal wind series changes from westerly to easterly. Composite values of the original unsmoothed data were then computed for ±18 months around these zero months. Figure 12 shows composited zonal mean zonal wind, zonal wave forcing due to resolved waves (i.e., EP flux divergences) and parameterized non-orographic GWP in 10°S-10°N during El Niño and La Niña. Note that these composite fields are meridionally averaged from 10°N to 10°S, and thus the structure of the QBO is somewhat different compared with that just over the equator (Fig. 2).

As shown in Fig. 2, the ECHAM QBO during El Niño is somewhat unrealistic (see contour lines), and the structure of the composite QBO does not show continuous downward propagation with weakening amplitude (i.e., westerly QBO has two maxima around 40 hPa and 7 hPa), although a more realistic QBO is found during La Niña. These differences are related to those of QBO amplitude (Fig. 4), in which only ECHAM shows large negative ENSO differences (i.e., El Niño amplitude is weaker) throughout the stratosphere with two minima at ~30 hPa and ~7 hPa. Zonal wave forcing due to both resolved and parameterized waves is larger during La Niña than El Niño at 10-40 hPa. ECHAM is the only model in which wave forcing during La Niña is larger than El Niño at these altitudes. Since zonal wave forcing distribution is closely related to the vertical shear of the zonal wind as well as wave source strength, it is difficult to compare the results of this model to others based its QBO response to El Niño.

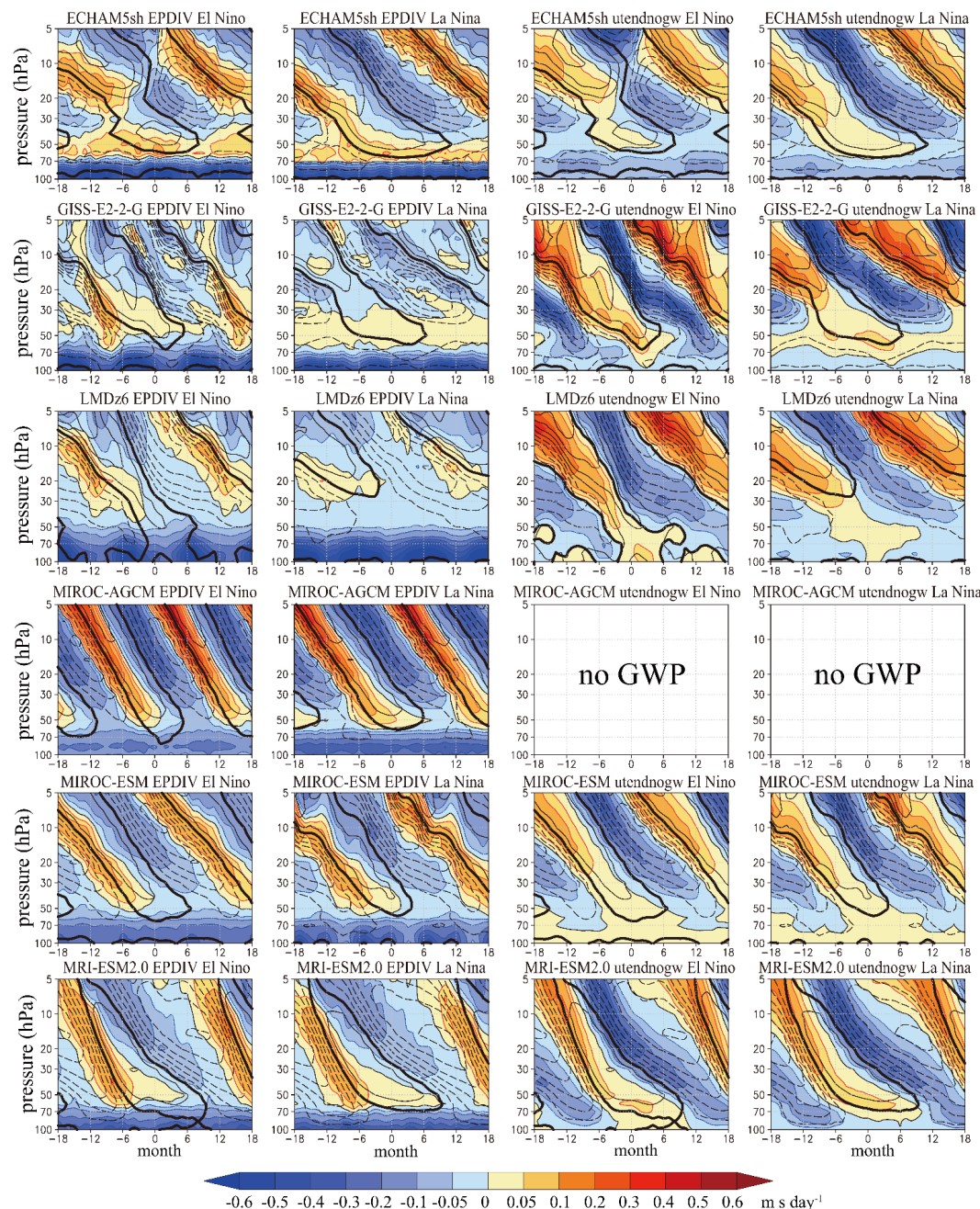

**Fig. 12: The contours in each panel show the QBO cycle composites of the zonal mean zonal wind averaged over 10°S–10°N for one model in either El Niño (columns 1 and 3) or La Niña (columns 2 and 4) experiments. The color shading in columns 1 and 2 shows the EP flux divergence from resolved motions averaged over 10°S–10°N. The color shading in columns 3 and 4 shows the mean flow forcing from the non-orographic gravity wave drag parameterizations also averaged over 10°S–10°N. The QBO composite in each case is made relative to month zero which corresponds to the westerly-to-easterly phase transition of the zonal wind at 20 hPa. The contour interval is 5 m s⁻¹. The color intervals are ±0.05, 0.1, 0.15, 0.2, 0.3, 0.4, and 0.5 m s⁻¹ day⁻¹.**

The GISS and LMDz models, which both use variable sources of parameterized GWP, produce similar behavior in their simulations of the QBO. For both models the descent of the westerly phase of the QBO into the lower stratosphere is more evident during El Niño compared to La Niña. As seen in Fig. 4, the QBO amplitude in both models is larger from ~70 hPa to ~10 hPa during El Niño. Both models show larger eastward forcing due to resolved waves during El Niño, specifically around 20-50 hPa in GISS and 10-30 hPa in LMDz. As seen in Fig. 9d, precipitation over the equator is significantly larger during El Niño than La Niña, which is a favorable condition to generate Kelvin waves, which can contribute to driving the westerly phase of the QBO (e.g., Kawatani et al. 2010b, 2019). In the companion paper, Elsburly et al. (2025), it is shown that convectively coupled Kelvin waves are more active in El Niño than La Niña runs. Other characteristics for these two models

are that zonal wave forcing by parameterized waves is much larger than that by resolved waves in both eastward and westward accelerations. In addition, parameterized wave forcing is stronger during El Niño than La Niña for both eastward and westward directions, especially below 30 hPa.

On the other hand, in MIROC-ESM and MRI, differences in resolved and parameterized wave forcing are not as large as those in GISS and LMDz. In MRI, westward forcing by parameterized waves is generally larger than that by resolved waves, especially in the lower stratosphere. This is consistent with previous studies (e.g., Giorgetta et al. 2002), as the easterly phases of the QBO are mainly driven by small-scale gravity waves (e.g., Kawatani et al. 2010). In the MIROC-ESM, parameterized eastward wave forcing is significantly smaller than that from resolved waves at 30-50 hPa.

In MIROC-AGCM, in which the QBO is driven by resolved waves only, asymmetry of wave forcing between El Niño and La Niña is found below ~40 hPa for westward forcing. K2019 show that these differences are mainly due to gravity waves with zonal wavenumbers greater than 42, i.e., smaller-scale gravity waves that could not be resolved by a moderate resolution T42 model. In this model the mean QBO period difference between El Niño and La Niña is ~3 months (Fig. 3), which makes only modest differences in the composite. However, the slightly delayed downward phase progression of the easterly regime around 30-50 hPa in La Niña relative to El Niño seen earlier in Fig. 2 is also found in these composites (see the contour lines around 30–50 hPa during months 3–9 in Fig. 12).

Figure 13 presents vertical profiles of mean eastward and westward forcing due to resolved and parameterized waves during El Niño and La Niña averaged over 10°S-10°N. Note that in ERA5, the parameterized forcing includes not only the effects of non-orographic gravity waves but also other types of zonal forcing. For simplicity, the mean eastward and westward wave forcing at a specific altitude was calculated by summing the forcing when the sign was positive and negative for all periods in each model, respectively, and then averaging the results. Here, resolved, parameterized, and total (i.e., resolved plus parameterized) wave forcing is shown separately during El Niño and La Niña.

Observations, as represented by the ERA5 assimilated fields display larger total wave forcing during El Niño than during La Niña above approximately 25 hPa for eastward forcing and above approximately 15 hPa for westward forcing. For eastward forcing, both resolved and parameterized components are stronger in El Niño than in La Niña in the middle and upper stratosphere. For westward forcing, the parameterized component becomes notably larger than the resolved component above 15 hPa. This may reflect the reduced influence of observational constraints at higher altitudes, where the radiosonde observation density is low and the zonal wind fields are more strongly shaped by the parameterizations used within the reanalysis system.

As discussed for Fig. 13, ECHAM is the only model which shows significantly larger total wave forcing during La Niña (black dashed lines) compared to El Niño (black solid lines) at most altitudes between 70-5 hPa as also seen in Fig. 13, but we do not discuss this further due to unrealistic structures of the QBO during El Niño in this model. In GISS and LMDz, we confirm again that parameterized wave forcing (red lines) is generally much larger than resolved wave forcing (blue lines), especially above ~30 hPa (note that exact altitudes of positive or negative differences depend on eastward or westward forcing as well as the model). Total wave forcing is larger during El Niño below ~20 hPa in both eastward and westward directions in GISS, and is larger at all altitudes for eastward waves during El Niño and up to ~15 hPa for westward waves in LMDz.

In MIROC-ESM, eastward wave forcing by resolved waves is larger than parameterized waves, while westward wave forcing due to resolved and parameterized waves is comparable above ~40 hPa. Differences in total wave forcing in this model are almost identical between El Niño and La Niña, except for slightly larger westward forcing in El Niño at 40-70 hPa. In the MRI, total eastward wave forcing in El Niño is significantly larger above 20 hPa, which is mainly due to differences in resolved wave forcing between El Niño and La Niña. For westward waves, parameterized wave forcing is much larger than resolved wave forcing. Differences in total wave forcing are found below 40 hPa, with larger forcing during El Niño. In MIROC-AGCM, both eastward and westward resolved wave forcing is larger during El Niño than La Niña, with particularly clear differences found above 40 hPa for eastward and 40-60 hPa for westward waves.

Finally, we briefly discuss the QBO period modulation associated with differences in tropical upwelling and wave forcing (ECHAM is omitted here for the reasons mentioned above). All models consistently simulate longer QBO periods during La Niña than El Niño. Both GISS and LMDz, which utilize variable sources of parameterized GWP, exhibit distinctly larger wave forcing during El Niño, despite having quantitatively different vertical profiles of this forcing (Fig. 13). Tropical
upwelling $\overline{w}^*$ differences are larger in GISS compared to LMDz up to ~50 hPa but are smaller above this level (i.e., LMDz simulate stronger $\overline{w}^*$, see Fig. 10). The QBO period is 10.2% longer during La Niña in GISS and 27.9% longer in LMDz (Fig. 3), indicating LMDz shows a much stronger ENSO effect on the simulated modulation of QBO period. As seen in Fig. 2, the GISS QBO in the La Niña run shows continuous westerly around 50 hPa, while downward propagation of QBO westerly phases sometimes stalls around 20 hPa in LMDz. These different QBO modulations make it difficult to simply judge wave
forcing and $\overline{w}^*$ differences to evaluate the factors responsible for the distinct QBO period differences between the two models.

MIROC-ESM and MRI, which utilize Hines-type GWP with fixed wave sources, exhibit smaller differences in wave forcing. Notably, these two models demonstrate the least modulation of the QBO period in association with ENSO, with variations of 6.9% and 8.5% for MIROC-ESM and MRI respectively (Fig. 3). MIROC-AGCM shows larger resolved wave forcing during El Niño than La Niña, although the differences are not very large and QBO period modulation by ENSO is
18.7%. We cannot definitively conclude that models with variable GWP sources generally have a simulated period that is more sensitive to ENSO modulation than those with fixed sources. Indeed, some models with fixed GWP sources, such as EC-EARTH, and some with variable sources, such as CESM1, exhibit substantial modulations, suggesting that other factors beyond the variability of the GWP source are also important. Further investigation of these models is hampered by the incomplete model variables in the available data sets.

This simple analysis with limited model output data cannot fully explain quantitative differences in QBO periods between El Niño and La Niña. More detailed wave analyses, based on TEM diagnostics using high temporal datasets, zonal wavenumber versus frequency spectra of EP-flux and precipitation, parameterized wave forcing properties including their 3-dimensional variations, etc., are required.

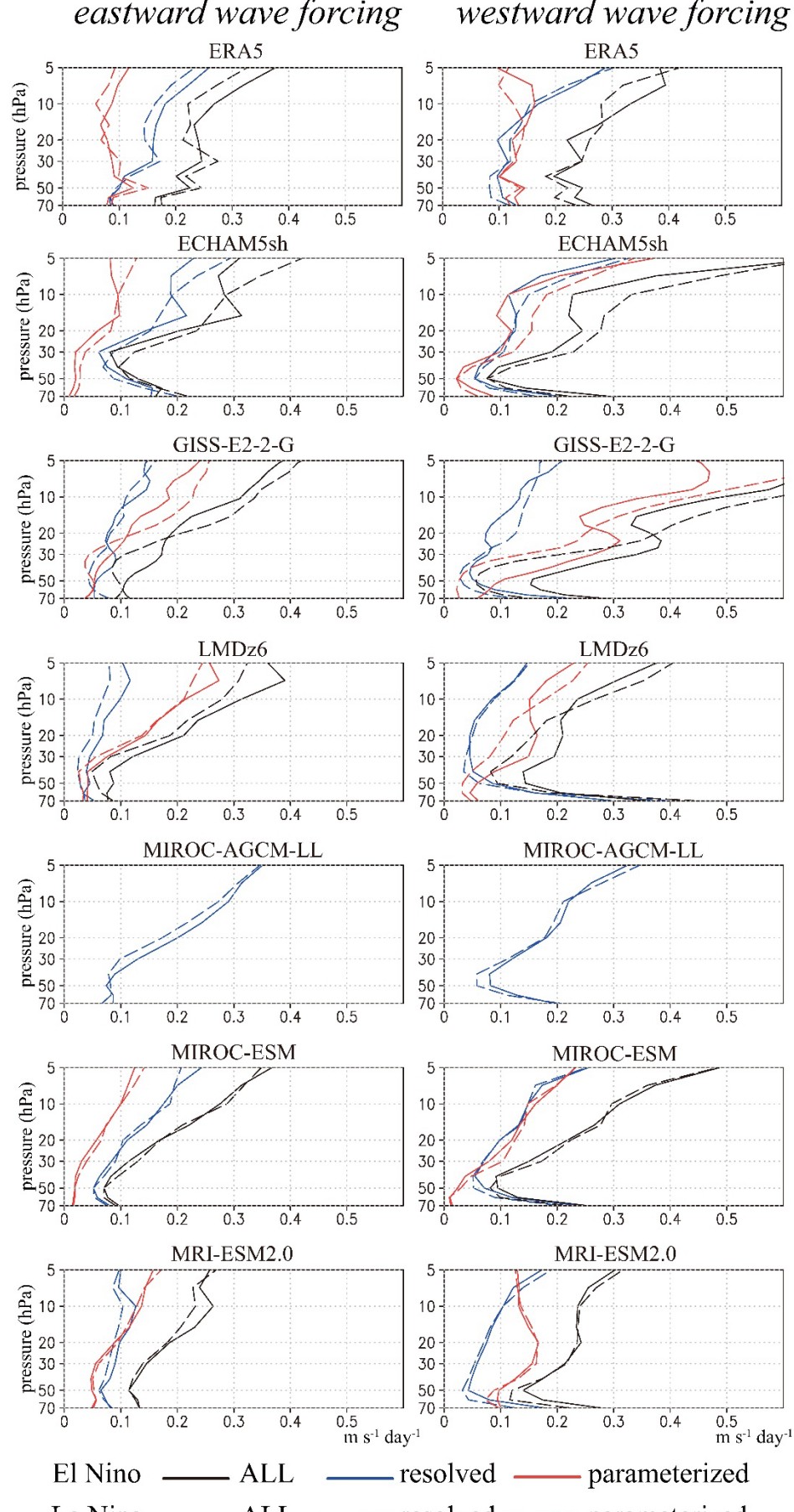

**Fig. 13: Vertical profiles of the 10°S–10°N averaged wave forcing of the zonal mean flow from (blue) resolved forcing, (red) parameterized non-orographic gravity waves and (black) their sum. Solid and dashed lines correspond to El Niño and La Niña runs. Note that in ERA5, the parameterized forcing includes not only non-orographic gravity waves but also other types of zonal forcing.**

## 6. Discussion and Summary

### 6.1 Discussion

The influence of SST anomaly magnitude on the QBO response to ENSO forcing is first discussed in this subsection. Taguchi (2010) addressed this issue in an observational context by increasing the threshold for defining El Niño and La Niña events from ±0.5°C to ±0.8°C. Although the corresponding results were not shown, Taguchi (2010) noted that "the results are generally insensitive to the definitions of the ENSO cases, since the weaker amplitude and faster phase propagation of the QBO are also obtained for the stronger EL conditions." This suggests that QBO responses to ENSO variability are robust

across different ENSO thresholds. However, the limited length of the observational record introduces uncertainties and limits definitive assessment of the degree of nonlinearity in the response to tropical SST anomalies.

      Further insight into the role of SST amplitude can be gained by examining model simulations under different ENSO forcing strengths. For this purpose, simulations from MIROC-AGCM and MIROC-ESM are compared between the QBOi-ENSO experiments, which used strongly amplified SST anomalies scaled by factors of 1.8 (El Niño) and 1.4 (La Niña), and

earlier integrations reported in K2019, where composite SST anomalies were imposed without such scaling. In K2019, both the El Niño and La Niña experiments were conducted under perpetual SST forcing and integrated for 100 years each.

      While the MIROC-AGCM simulations in the two experiments were based on the same model version, they differ slightly in parameter settings. In contrast, the MIROC-ESM simulations used identical model versions and configurations across both experiments. In the ENSO experiments, the period of the QBO is shorter during El Niño than during La Niña by

approximately 3.09 months in MIROC-AGCM and 1.55 months in MIROC-ESM. In contrast, the K2019 experiments with more modest SST anomalies show smaller differences of 2.2 and 0.4 months, respectively, with the latter not being statistically significant. These results imply that stronger SST forcing leads to more pronounced QBO responses.

      A complementary approach for investigating the influence of ENSO SST amplitude on the QBO is provided by QBOi Experiment 1 (Exp1). This experiment consists of AMIP-type simulations in which each model is forced by observed monthly

sea surface temperatures, sea ice, and external radiative forcings over the 1979–2009 period (Butchart et al., 2018). In contrast to the perpetual ENSO experiments that impose amplified and fixed SST anomalies, Exp1 incorporates the full temporal evolution of observed SST anomalies, including the natural development and transitions between ENSO phases.

      The integration period for each ensemble member is 31 years. One ensemble member is available for each of ECHAM5sh, LMDz, and MRI-ESM2.0, while three ensemble members are available for MIROC-AGCM, MIROC-ESM, and

CESM1. El Niño and La Niña months in Exp1 were identified using the definition provided by JMA, in a manner consistent with the observational analyses. However, the number of El Niño and La Niña months identified in Exp1 is considerably smaller than in the perpetual ENSO experiments, particularly when stratified by QBO phase and season, which introduces sampling uncertainty and limits the robustness of direct comparisons.

      The results of Exp1 are summarized in Supplementary Section S5 (Figs. S5–S7). In general, the tendency for faster

QBO descent during El Niño, as measured by the phase progression rate, is found in most models and is consistent with the results from the ENSO experiments. While the overall pattern of ENSO influence on QBO downward propagation appears robust, the sensitivity of QBO amplitude and phase progression to SST anomaly magnitude is more difficult to assess in Exp1. This is likely due to the limited sample size, the presence of other sources of interannual and decadal variability (e.g., volcanic eruptions and solar fluctuations), and the non-perpetual nature of the boundary conditions.

Given these limitations, while Exp1 provides a useful point of comparison under more realistic boundary conditions, the results are not sufficient to determine whether the modulation of the QBO by ENSO forcing scales linearly with the amplitude of SST anomalies. The relatively small sample size of El Niño and La Niña months, the presence of additional sources of interannual and decadal variability such as volcanic eruptions and solar fluctuations, and the non-perpetual nature of the boundary conditions all contribute to increased uncertainty in quantifying SST sensitivity. As a result, the Exp1

simulations do not allow for a clear assessment of nonlinear responses in QBO amplitude or phase progression rate. These

findings underscore the need for targeted sensitivity experiments with systematically varied SST anomaly amplitudes and long integration periods in order to rigorously evaluate the degree of linearity or nonlinearity in the ENSO–QBO relationship.

Despite these remaining questions, the results from the QBOi-ENSO experiments provide several robust insights into how ENSO modulates the QBO in QBO-resolving climate models. These main findings are summarized below.


## 6.2 Summary and Concluding Remarks

This study investigates how ENSO modulates the QBO using nine climate models that participated in QBOi . The experimental design builds upon Experiment 2 of QBOi phase-1 (see details in Butchart et al. 2018), but employs annually repeating SST patterns characteristic of El Niño and La Niña conditions. These SST anomalies, derived from observed data
(1950-2016), were amplified to enhance their impact on QBO simulations. Note that, other prescribed fields like sea ice and ozone remained unchanged between El Niño and La Niña runs, and so the imposed SST anomalies represent the sole difference in boundary conditions.

The models differ in their representation of gravity wave processes, with five models (EC-EARTH, ECHAM, EMAC, MIROC-ESM, MRI) using fixed sources of parameterized gravity waves, three (GISS, LMDz, CESM1) incorporating variable
gravity wave sources, and one (MIROC-AGCM) simulating the QBO solely through resolved wave dynamics. The study analyzes a comprehensive suite of atmospheric variables from the model simulations, comparing them to ERA5 reanalysis data and CMAP precipitation data to evaluate the models' ability to capture observed ENSO modulations.

The absolute values of the simulated QBO periods differ among models. Some models, such as MIROC-AGCM and MIROC-ESM, exhibit substantially shorter mean periods than the observed value of approximately 28 months. Others,
including EMAC and GISS, also show somewhat shorter periods. While these discrepancies represent structural limitations in simulating realistic QBO behavior, the relative differences between El Niño and La Niña remain internally consistent and interpretable. Therefore, these models can still provide useful insight into ENSO's modulation of the QBO despite biases in simulated QBO period.

A key finding is that all models consistently simulate longer QBO periods during La Niña compared to El Niño, in
basic agreement with observations. In contrast, the modulation of QBO amplitude varies significantly among models, with GISS, LMDz and CESM1 (which used variable parameterized gravity wave sources) exhibiting the most pronounced differences, while EC-EARTH, EMAC, MIROC-ESM and MRI show more modest changes. This finding contrasts the results from global warming experiments performed with the same QBOi models, in which projected QBO amplitude reductions were found to be much more consistent across models, but projected QBO period changes are inconsistent, showing shorter, longer,
or even disappearing QBOs depending on the model (Richter et al. 2020).

The long term mean differences of the basic wind, temperature and precipitation fields associated with ENSO are qualitatively similar among models and broadly consistent with observations. The models consistently capture the observed pattern of increased precipitation over the equator during El Niño, conducive to generating waves that can effectively interact with the QBO. Furthermore, differences in zonal-mean temperature and zonal wind patterns between El Niño and La Niña are
evident, consistent with observed ENSO-related changes. Overall, these results indicate that the QBOi models successfully capture fundamental features of ENSO's influence on the QBO and mean climate, despite variations in the magnitude and details of these responses across different models.

Focusing on equatorial tropical upwelling, the models consistently show stronger upwelling during El Niño compared to La Niña, particularly up to approximately 10 hPa, with the most significant differences observed around 100-50 hPa. While
these differences generally become smaller in the middle to upper stratosphere, LMDz and MRI stand out, exhibiting relatively larger differences due to the modulation extending to the deep branch of the BDC.

The Walker circulation, characterized in the upper troposphere by easterly winds in the eastern hemisphere and westerly winds in the western hemisphere, plays a significant role in filtering gravity waves propagating from the troposphere

to the stratosphere. A weaker Walker circulation is evident during El Niño, potentially allowing for enhanced gravity wave propagation into the stratosphere due to reduced filtering. The representation of the Walker circulation during El Niño and La Niña shows some variation among models, but the El Niño minus La Niña differences are very similar across models, with easterly anomalies over the central Pacific and westerly anomalies in other longitudes, consistent with observations.

The three models using Hines-type parameterized GWP (ECHAM, MIROC-ESM, and MRI-ESM) show similar patterns in longitudinal variations of eastward and westward fluxes during both El Niño and La Niña at 100 hPa. Eastward flux differences between El Niño and La Niña are notably positive around the central Pacific, with negative anomalies to the east and west. In contrast, westward flux differences are predominantly negative around the central Pacific. These variations align well with the differences in background zonal winds associated with the Walker circulation.

CESM1 exhibits a distinct, localized distribution of parameterized gravity wave fluxes compared to other models, likely due to its formulation of non-orographic GWP which features sources assumed dependent on the simulated adiabatic convective heating. This results in ENSO related variations influenced by both variable sources and Walker circulation filtering. The difference in eastward fluxes between El Niño and La Niña reveals positive values from the central to east Pacific, aligning with the location of maximum easterly anomalies of the Walker circulation. Conversely, negative values correspond to regions of westerly anomalies in the mid-to-upper tropospheric Walker circulation at other longitudes. These differences show some resemblance to those observed in ECHAM, MIROC-ESM, and MRI. Unlike eastward waves, differences in westward fluxes in CESM1 exhibit large positive values in the eastern Pacific and negative anomalies in the western Pacific, associated with positive precipitation anomalies.

Zonal wave forcing due to resolved and parameterized waves were composited based on QBO phase and these composite fields display some differences among models. In particular, GISS and LMDz show larger eastward forcing due to resolved waves during El Niño compared to La Niña. Increased precipitation over the equator is a favorable condition for generating more Kelvin waves, which contribute to driving the westerly phase of the QBO. Both models, with variable parameterized gravity wave sources, have zonal wave forcing by parameterized waves that is much larger than that by resolved waves in both eastward and westward accelerations. In contrast, MIROC-ESM and MRI show smaller differences between resolved and parameterized wave forcing. In MIROC-AGCM, a slight delay in the downward phase progression of the easterly phases during La Niña compared to El Niño is observed around 30-50 hPa, associated with smaller westward resolved wave forcing.

The vertical profiles of mean eastward and westward forcing due to resolved and parameterized waves during El Niño and La Niña also exhibit inter-model differences. GISS and LMDz display the dominance of parameterized wave forcing over resolved wave forcing, particularly above 30 hPa. Total wave forcing is generally stronger during El Niño in both GISS and LMDz. In MIROC-ESM, resolved eastward wave forcing surpasses parameterized forcing, while resolved and parameterized westward wave forcing is comparable. Differences in total wave forcing are small between El Niño and La Niña, except for slightly stronger westward forcing during El Niño between 40-70 hPa. In MRI, total eastward wave forcing during El Niño is significantly larger above 20 hPa, primarily driven by differences in resolved wave forcing. Parameterized westward wave forcing dominates over resolved forcing, and differences in total wave forcing are observed below 40 hPa, with larger forcing during El Niño. In MIROC-AGCM, both eastward and westward resolved wave forcing is stronger during El Niño than La Niña, with notable differences above 40 hPa for eastward forcing and 40-60 hPa for westward forcing.

The QBO period is consistently longer during La Niña than El Niño across all models. GISS and LMDz, which utilize parameterized GWPs with variable sources, exhibit significantly stronger wave forcing during El Niño and a correspondingly larger ENSO modulation of the simulated QBO period, consistent with previous studies (Geller et al. 2016; Zhou et al. 2024). Of the nine models, EMAC, MIROC-ESM and MRI, which use Hines-type parameterized GWP schemes with fixed wave sources, show the weakest modulation of the QBO period by ENSO. While MIROC-AGCM, which relies solely on resolved waves, also produces a longer QBO period during La Niña, its results need to be considered in light of the model's moderate

T106 horizontal resolution. Two other models, EC-EARTH (fixed GWP sources) and CESM1 (variable GWP sources), exhibit larger QBO period modulations, but further analysis is limited by data availability. It is noted here that only three of the nine models (EC-EARTH, LMDz, and ECHAM) simulate La Niña–El Niño differences in QBO period that approach the observed
sensitivity (~27 %), even under the amplified ENSO forcing. The remaining six models exhibit more modest ENSO modulation of the QBO period.

In contrast to the consistent QBO period response, the sign of the ENSO effect on QBO amplitude varies among models. However, those employing variable parameterized gravity wave sources generally exhibit greater sensitivity of the QBO amplitude to ENSO than those using fixed sources (compare GISS, LMDz, and CESM1 to EC-EARTH, EMAC,
MIROC-ESM, and MRI in Fig. 4).

While this analysis provides initial insights into the QBO response to ENSO through a multi model comparison, further investigation is needed to fully understand the quantitative differences in QBO periods between El Niño and La Niña. This requires more detailed wave analyses, incorporating datasets with high temporal sampling. Notably such high frequency data could be applied to examine zonal wavenumber versus frequency spectra of EP-flux and precipitation, and to analyze
parameterized wave forcing properties, including their three-dimensional variations.

Future research efforts should prioritize several key areas to enhance our understanding of the complex interplay between ENSO and the QBO, ultimately contributing to more accurate climate change predictions. Firstly, different schemes of parameterized gravity waves lead to substantial differences among models. Comprehensive comparisons across various models are needed to refine these parameterizations and reduce these discrepancies. Secondly, further research is necessary to
unravel the sources of gravity waves. While gravity waves are generated through various processes, the mechanisms behind their formation are not fully understood. This is particularly true for gravity waves generated by convective activity, which requires further investigation. Finally, conducting much higher resolution simulations is essential. High-resolution simulations allow for more detailed analysis of gravity wave propagation and their impact on the QBO. In addition, comparing non-orographic parameterized gravity waves to observations, such as constant level balloons as done by Lott et al. (2024), would
be valuable. By addressing these research priorities, we can gain a more comprehensive understanding of the complex interactions between ENSO and the QBO, which could contribute to more accurate climate change predictions.

**Data availability**

The JMA data and data description are provided in the web-pages:

*El Niño monitoring and outlook*: http://ds.data.jma.go.jp/tcc/tcc/products/elnino/index.html

*Download El Niño Monitoring Indices*: http://ds.data.jma.go.jp/tcc/tcc/products/elnino/index

Storage for the QBOi multi-model data set is provided by the Centre for Environmental Data Analysis (CEDA) whose data and processing service is called JASMIN. Interested users must obtain a JASMIN login account and take the necessary steps to access the QBOi group workspace within JASMIN, which contains the perpetual ENSO simulations.


**Code availability**

GFD-DENNOU Club codes are available at https://dennou-h.gfd-dennou.org/index.html.en

Grid Analysis and Display System (GrADS) is available at http://cola.gmu.edu/grads/

**Author contributions**

YK and KH wrote the manuscript. YK and MT drew figures. FS calculated TEM-related quantities in ERA5. YK, KH, SW, JA, JR, NB and SO contributed to the conceptualization of this study. YK, SW, JR, CO, HN, CC, JGS, AG, TK, FL, FP, FS, SV and KY did the model experiments. All authors contribute to review and edit the manuscript.

**Competing interests**

The authors declare that they have no conflict of interest.

**Acknowledgement**

YK was supported by JSPS KAKENHI (JP22K18743) and the Environment Research and Technology Development Fund (JPMEERF20242001) of the Environmental Restoration and Conservation Agency provided by Ministry of the Environment of Japan. YK and SW were supported by JSPS KAKENHI (JP22H01303 and JP23K22574). SW was supported by MEXT-Program for the advanced studies of climate change projection (SENTAN) Grant Number JPMXD0722681344. KH were supported by the Japan Agency for Marine-Earth Science and Technology (JAMSTEC) through its sponsorship of research at the International Pacific Research Center. TK and SV acknowledge support by the state of Baden-Württemberg through bwHPC. This work was supported by the National Center for Atmospheric Research (NCAR), which is a major facility sponsored by the National Science Foundation (NSF) under Cooperative Agreement 1852977. Portions of this study were supported by the Regional and Global Model Analysis (RGMA) component of the Earth and Environmental System Modeling Program of the U.S. Department of Energy's Office of Biological and Environmental Research (BER) via NSF Interagency Agreement 1844590. FS thanks C. Cagnazzo for discussions on the setup of ECHAM5sh simulations, carried out thanks to an ECMWF Special project. The numerical simulations of MIROC models were performed using the Earth Simulator. The GFD-DENNOU Library and GrADS were used to draw the figures.

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
