# Peer review of "QBOi El Niño-Southern Oscillation experiments: Overview of experiment design and ENSO modulation of the QBO"

_EGUsphere, 2024_

## Referee Comment (RC1)

Review of the article *QBOi El Nino Southern Oscillation experiments Part I: Overview of experiment design and ENSO modulation of the QBO* by Yoshio Kawatani et al.

General comments

This work is a continuation of the publication series produced by the QBOi project, based on an experimental protocol and several models known to simulate the QBO. This work is focused on the El Niño/La Niña effects on the QBO as simulated in 9 models. Most of the article describes the common and different features found in the different model simulations, and its structure follows the work of Kawatani et al. (2019), hereafter K2019, where they investigated the El Niño/La Niña effects on the QBO in MIROC models.

The experimental design chosen here is however deviating from that of K2019. Here they decided to uses amplified mean El Niño/La Niña SST anomalies. This makes any direct comparison to K2019 and to observations difficult. Whether or not the QBO response should be linear to the amplitude of the SST anomaly pattern is not discussed. Probably it would have been better to use the same anomalies as in K2019. (Or an entirely different design based on SST fields of selected El Niño and La Niña years.)

They find that El Niño/La Niña effects on the QBO period are qualitatively similar with respect to the period, with El Niño leading to a shorter period despite of the increased tropical upwelling in the tropical lower stratosphere, from which it is clear that El Niño must also produces a stronger wave meanflow interaction. No common response is found for the QBO amplitude.

An interesting part is the discussion and analysis of the reasons for the described results: The more equatorial precipitation and the weaker Walker circulation found during El Niño conditions. These features are found in all models, and they probably are independent of the skill of a model to simulate a QBO. The discussion of the wave meanflow processes is however more complicated, because of the rather different ways that resolved and parameterized waves generate the QBOs in the different models. And therefore not so much can be learned from this part, except that there exists still a considerable difference in the way that models generate QBOs. Further, as acknowledged by the authors, more detailed model diagnostics would be needed to learn more about the underlying reasons for the found behaviours. But this additional diagnostics was not part of the protocol, or the modelling groups could not produce these diagnostics.

Overall I think that the publication is worthwhile, as it creates a baseline for further work on this topic. Some minor corrections are needed before publication.

Detailed comments and questions

**Abstract**

L40 Stratosphere-troposphere Processes And their Role in Climate (SPARC) …
As we know QBOi has been started as a SPARC project. But SPARC has changed its name to APARC and QBOi is now listed as an APARC project. Maybe it is worth to add a remark or a footnote on this aspect.

L45 …  models -models  … should probably be … models. Models …

**1 Introduction**

L64 … that QBO facilitates …  →  … that the QBO facilitates …

L140 … Conducting a common ENSO-QBO experiment across a range of QBO-resolved climate models could help elucidate the role of non-orographic GWP in driving the oscillation. …

The work of Richter et al. (2020) on the climate warming effects on the QBO unfortunately showed that the differences between GWPs are considerable and probably responsible for the rather different QBO responses to the warming. As it seems it was not possible to decide which GWPs were "wrong" or "right". Now a similar exercise is presented aiming at El Niño/La Niña variations in SST as the external forcing instead of a warmer SST and increased atmospheric $CO_2$. Why should we expect a scientifically more robust result if Richter et al. (2020) have shown that differences in parameterizing non-orographic gravity drag can lead to very different results? Simply because El Niño/La Niña cycles exist in the historical period for which the models have been tuned?

**2. Model Description and Experimental Design**

L179 … These factors bring the peak composite anomaly SSTs closer to the anomalies observed during the most intense El Niño and La Niña events. …

Using amplification factors is problematic. This makes a comparison to observations or to the work by K2019 difficult. It seems necessary to add some remarks about the linearity between the SST pattern amplitude and the response of the QBO. Can this be assumed?

Alternatively you could have chosen specific years with strong El Niño and La Niña SST anomalies. Then there would be no need to amplify the SST anomaly, and there would be less of a risk to construct an SST anomaly pattern that mixes the different types of El Niños, which are discussed in literature.

L199 … For clarity and conciseness, we will refer to these models as CESM1, EC-EARTH, ECHAM, EMAC, GISS, LMDz, MIROC-AGCM, MIROC-ESM and MRICESM1, respectively. …

The abbreviated model names are introduced here, but not used consequently. Tables, Figures, and also soe sentences use the full model names. Please decide whether short names shall be used or not. But if you decide to use short names, then please use these in all places: tables, figures, and text.

L203 … Launch levels for parameterized gravity waves varied across models, ranging from 450 to 700 hPa or 1000 to 100 hPa. …

To which model(s) do the two pressure ranges relate? Please clarify.

Table 1. lunched level → launch level

Table 2. What does the entry for GISS-E2-2G and Residual stream function (✓5-1115) mean?

L243 … from the ERA-Interim (ERA-I; Dee et al. 2011) reanalyses …

Why is ERA-I used for this comparison, when ERA-5 is now available? Newer reanalyses are generally improved compared to earlier ones.

L247 … Importantly, the composite ERA-I and CMAP data were not scaled …

This is a kind of a flaw in the experimental design. If the response to the SST anomaly patterns is non-linear to the amplitude, then the applied scaling is hindering a direct comparison to observations or analyses. If, however the signal is linear, then the signals derived from ERA-I should be scaled like the SST patterns used for the simulations.

**3. ENSO Modulation of the QBO and Climatological Mean Field Differences**

L266 – L276 These lines discuss deficiencies in the structure of the simulated QBO, as occurring in El Niño or La Niña simulations of ECHAM, GISS, and LMDz. In my opinion it is necessary to point out another deficiency, which is an unrealistic period, although a regular pattern of downward propagating westerly and easterly jets is simulated. Taking the displayed 20 years (Fig. 2) of the El Niño and La Niña simulations together, we have 40 years for which on average (40years / 28 months) we would expect about 17 cycles. A count of the cycles shown in Figure 2 can now serve as an additional measure for the quality of the simulations. If we allowed a range of 15 to 19, then the following models (here excluding ECHAM, GISS and LMDz) would fail: EC-EARTH: 20, MIROC-AGCM-LL: 26, MIROC-ESM: 21. Please extend your discussion of problematic simulations in this direction, so that the reader knows from the beginning which model simulations need to be viewed a bit more critically.

L311 … Next, we consider ENSO modulation of QBO amplitude, which is known less robust … → … Next, we consider the ENSO modulation of the QBO amplitude, which is known to be less robust …

L323 … GISS, LMDz, and CESM1, all of which have variable GWP sources. …
I think it should be added that MIROC-AGCM-LL has variable gravity waves too, though these are explicitly simulated, within the given resolution, instead of parameterized. Thus variability of gravity waves not necessarily leads to a strong amplitude difference between El Niño and La Niña. And one needs to wonder if a strong change is indicating that the variability of gravity waves is important aspect for a GWP, or whether this effect is rather a result of other aspects of parameterizing gravity wave. Please add some thoughts on this problem.

L383 … although models tend to simulate the precipitation peak to the east of the observed one over the central Pacific in the El Niño run. …
It should also be mentioned that the precipitation peak in the model simulations is higher than in observations, which indicates that the local forcing by latent heat release in the simulations is higher than that explained by the observed precipitation. Quite likely this is related to the amplified El Niño/La Niña SST strength.

L422 … significantly deep westerly difference … → … significantly deeper westerly difference …

L432 …  for (left top) ERA-I  …
ERA-I is "center top"

**4 Contrasting wave forcing and residual mean meridional circulations in El Niño and La Niña from QBOi models**

L458 … The $\overline{X}$ term represents any other unresolved forcing. …
Do you mean here parameterized momentum diffusion and effects from numerical diffusion and damping operators?

L470 … in El Niño and La Niña simulations. …
La Niña simulations are nor shown, but differences of the El Niño and La Niña simulations.

L477 … La Niña c annual … → … La Niña annual …

L498 … which both use variable sources in their GWP, …
Do you mention this because you think that this is the reason for the differences? Often other differences in the formulation of the non-orographic gravity wave drag parameterizations can cause substantial differences already.

L509 … averaged over 20°S–20°N …
Maybe it is worth to explain why a band of 20°S – 20°N is chosen, while earlier diagnostics/figures used narrower bands. (I guess this is made in order to remove residuals of the secondary meridional circulation of the QBO.)

L510 … ranging from approximately 0.2 mm s⁻¹ in MIROC-AGCM to approximately 0.4 mm s⁻¹ in LMDz. …
This strong difference in the tropical upwelling implies also a strong difference in the strength of wave meanflow interaction that is necessary to simulate a QBO with a realistic period. This aspect is not discussed here, and maybe this El Niño/La Niña related article is the wrong place. Still it directly shows that the wave meanflow interaction must work at different strengths.

L522 … However, the specific altitudes at which $\overline{w}^*$ changes would most strongly influence the overall QBO period remain unclear. …
Sentence unclear.

L579 … While output data of parameterized gravity wave fluxes in LMDz were not available at the time of this analysis, this model, which also uses variable parameterized wave sources related to precipitation activity, showed similar structures affected by precipitation distributions (Dr. Lott, personal communication). …
Francois Lott is a co-author of this study. Please include the LMDz results in Figure 12.

L620 … parameterized wave forcing below is stronger …
What does "below" refer to? Maybe the sentence needs to be rephrased.

L639 … As discussed for Fig. 14, … → … As discussed for Fig. 13, …

**5. Summary and concluding remarks**

L685 … remained consistent …
"consistent" seems to be the wrong term, because this could have different meanings. (If El Niño/La Niña influences the ozone distribution, then the same ozone field cannot be consistent with El Niño and La Niña conditions at the same time.) "unchanged" would express more clearly that these fields simply have not been changed.

---

## Referee Comment (RC2)

Review of the manuscript "QBOi El Nino Southern Oscillation experiments
Part I: Overview of experiment design and ENSO modulation of the QBO" by Kawatani et al.

**Summary**

The manuscript presents an overview of the experimental design from the new SPARC QBOi project and examines the modulation of the QBO by ENSO using nine climate models. The findings indicate that the QBO period is longer during La Niña compared to El Niño across all models, consistent with observations. However, changes in the QBO amplitude remain inconclusive. Overall, I find the experiment intriguing, the manuscript well-written, and the results clearly explained. Most of my comments are minor and focus on improving consistency between different parts of the manuscript and aligning the figures with the text.

One major comment, however, concerns the lack of deeper insights into ENSO modulation of the QBO. The authors attribute this to the simplicity of the analyses and the limited availability of model output data, which they suggest prevents a full explanation of the quantitative differences in QBO between El Niño and La Niña. While future studies are mentioned as a potential avenue to address this, I argue that if a more detailed analysis is feasible, it should be included in this paper, as it was the primary motivation for the experiment and study.

**Major Comments**

1) Lack of additional insight into the mechanisms of ENSO modulation of the QBO

My primary concern is whether this paper and the associated experiments provide any additional insight into the mechanisms by which ENSO modulates the QBO. At the start of the paper, I had hoped—likely in line with the motivation behind designing and implementing these experiments—that this study would offer a deeper understanding of these mechanisms. However, the study appears to be an extension of Kawatani et al. (2019), with potentially more models included beyond MIROC, yet missing important analyses due to data limitations.

There are repeated statements such as: "Further investigation of these models is hampered by the incomplete model variables in the available data sets", "This simple analysis with limited model output data cannot fully explain quantitative differences in QBO periods between El Niño and La Niña", and "Detailed zonal-time spectral analyses of model fields, like those performed in Kawatani et al. (2019), remain a subject for future study."

If such analyses are indeed possible, this paper is the appropriate venue to present them, rather than postponing them to future studies. For example, as the authors mentioned, detailed spectral analyses of the EP flux, gravity wave parameterization fluxes, precipitation, or momentum

budgets based on the TEM framework could offer crucial insights into the intermodel spread of QBO period and amplitude.

To provide further context on my disappointment, Kawatani et al. (2019) noted: "It would be interesting to analyze the ENSO modulation of the three-dimensional wave forcing as well as tropical upwelling, which must show large differences between El Niño and La Niña. This may be investigated in a future study". Now, five years later, this study states: "A detailed investigation of the three-dimensional distributions of parameterized wave fluxes modulated by ENSO, including model dependence, would be of interest and remains a topic for future research."

It feels like an opportunity has been missed to address these outstanding questions. If there is a way to conduct these analyses, I strongly encourage the authors to include them in this paper.

2) Lack of use of recent data and citations of recent studies

Some aspects of the study, including citations and the data used, feel somewhat outdated. For instance, the use of ERA-I instead of ERA5. Additionally, the study only uses observed data up to 2012. If this limitation is due to avoiding the QBO disruptions, there are still several years of data available between 2012 and the end of 2015, as well as between 2020 and 2024. While including these additional years may not change the main conclusions of the paper, it would enhance the robustness of the analysis, particularly for slowly evolving phenomena like ENSO and QBO, where even a few more samples could provide valuable insights. Moreover, the citations miss some relevant and recent studies, such as Zhou et al. (2024), and a few others noted in my review.

**Minor comments**

L47-49: It can also be mentioned that "all models simulate stronger equatorial tropical upwelling in El Niño compared to La Niña up to ~10 hPa".

L85: Small-scale gravity waves also contribute significantly to the forcing of the QBO westerly (e.g., Pahlavan et al. (2021))

L95: As a good reference on this you can cite Coy et al. (2020).

In general, the figures can be significantly improved by reducing redundancy, which would allow for larger, clearer panels. For instance, in Figure 2, use "El Niño" as the title for the left column and "La Niña" for the right, rather than repeating them for each panel. Similarly, list model names only on the left side of the figure and show the y-axis (0–20) only on the bottom

panels, instead of repeating it in every panel. These changes can enhance readability and apply to other figures as well.

The other general issue with the figures is the presence of too many contours, which reduces readability. In particular, in Figures 4, 8, 9, 10, and 13, the contours over the shadings can be removed, similar to Figure 12, to improve clarity.

Figure 3: Have you analyzed each phase of the QBO separately? For example, do both phases of the QBO become shorter during El Niño?

Figure 3: Have you considered using a Fourier Transform to determine the period instead of relying on zero wind line crossing (e.g., as done in Lee et al. (2024))? While it likely won't change the conclusions, it might be a better option, particularly when the QBO becomes more irregular/unrealistic, as seen during El Niño in ECHAM.

L412: Will the cooler anomaly around 60°N–90°N in ERA-I, which is not observed in the models, change if more data is included, such as using ERA-5 from 1940 to 2024?

For Figures 10, 11, 13, and 14, you could consider including results from reanalysis (e.g., ERA5) as a reference, similar to what is done in Figures 8 and 9.

Figure 12: Could you add the total flux for El Niño and La Niña (i.e., averaged over 10°S–10°N and all longitudes) to the bottom panels? If so, is it consistently larger during El Niño?

**Editorial comments**

L84: "respectively" seems redundant.

L107: You can cite (Richter et al., 2020) again to avoid ambiguity.

L108: SST is not yet defined.

L140: "QBO-resolved" -> "QBO-resolving"

L164: GWP is already defined.

L165: What is experiment 2?

L165: SST needs to be defined at L108.

L198: The model name "CESM15-110L" is mentioned here, while "WACCM5-110L" is used in the results (figures and tables). I suggest selecting one naming convention for consistency.

L200: Using the concise version of model names is a great choice, but it would be helpful to maintain this approach consistently in the results (figures and tables) as well. Currently, there is a discrepancy where the text uses concise names while the results use the full model names, making it harder to follow.

L198: "CESM15-110L" is mentioned here but in the results (figures and tables) "WACCM5-110L" is used. I suggest choosing one for consistency.

L200: It is great to use the more concise version of the model names but it would be great to use the concise version in the results as well (figures and tables) to make it easier to follow. Now, there is this discrepancy between the text and results, the former using the concise version, and the latter the full name of the models.

L220: Palmerio et al. (2022) is not in the bibliography.

L238: TEM is already defined.

Table 2: what is "5-1115" in front of GISS.

L284: "larger" -> "longer"(?)

L295: "with" -> "for"

L375: ITCZ not defined yet.

L378: "(left-top)" -> "(center-top)"

L392: Any reference for this statement?

Figure 7: "PRCP" not defined.

L427: BDC is already defined.

L431: "(left-top)" -> "(center-top)"

L442: A point after 4 is missing. "In El Niño and La Niña from QBOi models" is redundant. Also, capitalize the first letter to maintain consistency with the other titles.

L443: "eddy forcing" -> "wave forcing", to be consistent with the other sections.

L443: "mean zonal" -> "zonal mean"

L444: TEM is already defined.

L445: "eddies" -> "waves"

L566-567: "WACCM" -> CESM1

L577: ITCZ should be defined earlier at L375.

L620: "below" should be removed.

L621: "below" -> "above" (?)

L639: "Fig. 14" -> "Fig. 13"

Caption of Fig. 14: "eddy" -> "wave". "resolved motions" -> "resolved forcing".

**References**

Coy, L., Newman, P. A., Strahan, S., & Pawson, S. (2020). Seasonal Variation of the Quasi-Biennial
Oscillation Descent. *Journal of Geophysical Research: Atmospheres*, *125*(18), e2020JD033077.
https://doi.org/10.1029/2020JD033077

Geller, M. A., Zhou, T., & Yuan, W. (2016). The QBO, gravity waves forced by tropical convection, and
ENSO. *Journal of Geophysical Research: Atmospheres*, *121*(15), 8886–8895.
https://doi.org/10.1002/2015JD024125

Kawatani, Y., Hamilton, K., Sato, K., Dunkerton, T. J., Watanabe, S., & Kikuchi, K. (2019). ENSO
Modulation of the QBO: Results from MIROC Models with and without Nonorographic Gravity
Wave Parameterization. https://doi.org/10.1175/JAS-D-19-0163.1

Lee, H.-K., Chun, H.-Y., Richter, J. H., Simpson, I. R., & Garcia, R. R. (2024). Contributions of
Parameterized Gravity Waves and Resolved Equatorial Waves to the QBO Period in a Future
Climate of CESM2. *Journal of Geophysical Research: Atmospheres*, *129*(8), e2024JD040744.
https://doi.org/10.1029/2024JD040744

Pahlavan, H. A., Wallace, J. M., Fu, Q., & Kiladis, G. N. (2021). Revisiting the Quasi-Biennial
Oscillation as Seen in ERA5. Part II: Evaluation of Waves and Wave Forcing. *Journal of the
Atmospheric Sciences*, *78*(3), 693–707. https://doi.org/10.1175/JAS-D-20-0249.1

Richter, J. H., Butchart, N., Kawatani, Y., Bushell, A. C., Holt, L., Serva, F., et al. (2020). Response of the Quasi-Biennial Oscillation to a warming climate in global climate models. *Quarterly Journal of the Royal Meteorological Society*, *n/a*(n/a). https://doi.org/10.1002/qj.3749

Zhou, T., DallaSanta, K. J., Orbe, C., Rind, D. H., Jonas, J. A., Nazarenko, L., et al. (2024). Exploring the ENSO modulation of the QBO periods with GISS E2.2 models. *Atmospheric Chemistry and Physics*, *24*(1), 509–532. https://doi.org/10.5194/acp-24-509-2024

---

## Editor Comment (EC1)

Dear Prof. Kawatani and co-authors:

Both Anonymous referees have posted their comments on your manuscript (WCD 2024-3270). As per WCD policy, you are now to post a response on how you will address the referee's comments – after which I will make a decision on the manuscript.

Both reviewers have made excellent comments on the manuscript and call for major revisions which I agree with, particularly concerning the issues of "what is new? (Reviewer #2)", the unexplored impact of unrealistically large forcing on the response (Reviewer #1), the lack of a discussion of agreement between the simulated QBOs and those observed (particularly the period) (Reviewer #1), and the use of very old reanalysis data throughout the manuscript (both Reviewers). To provide guidance in revising the manuscript so that it is acceptable for publication in WCD, below I itemize the issues that I expect will addressed in a revised manuscript. I will also post these on the WCD page for the manuscript. `

1. Reviewer #1 asks for an examination and justification of the impact of applying an ENSO forcing with an unrealizable amplitude. This should be addressed in the revised manuscript. One way to do this quantitatively is to run some of the models for ~20 years with realistic forcing and show how this impacts the wave forcing, say by producing Figs. 11 and 12 and comparing the amplitude of the wave forcing in the 3x El Nino with that with a 1x El Nino (and ditto for La Nina forcing).

2. Reviewer #1 calls for a necessary discussion of a deficiency in half the model models to simulate a QBO with a period that is consistent with that observed (see Reviewer #1 comments on Lines 266-76), and I agree. The reviewer's argument shows that four of the models have QBO periods that are unrealistic ( EC-EARTH: 20, MIROC-AGCM-LL: 26, MIROC-ESM: 21). I note that GISS is on the edge of being disqualified by this measure, and the unrealistic QBO in the La Nina simulations is a reason to add this model to the disqualified list. Hence, three of the six models in which wave driving is examined in Figs. 13 and 14 have QBOs with unrealistic periods. These issues should be mentioned in the abstract, in section 3, and in the discussion/summary in section 5.

3. Reviewer #2 asks whether the changes in the QBO period are sensitive to the method used to define a QBO cycle and whether ENSO impacts all phases of the QBO simulated by the models. I expect the method used to provide a robust period. However, it is not clear from the analysis presented whether El Nino accelerates all phases of the QBO, as is seen in the observations – or whether it impacts certain phases (such as the downward propagation of the westerly shear zone). In most of the models, the slope of the constant phase lines in the vertical-time plots with El Nino forcing experiment is indistinguishable from the slope in the La Nina forcing experiment (e.g., in Fig 2 for CESM1, EMAAC, MIROC-AGCM, MIROC-ESM and MRI-ESM2). Only in EC-Earth3.3 is the slope of the phase lines steeper with El Nino forcing than with La Nina forcing, as is also the case in the observations (Taguchi 2010, cf Fig. 9a with 9b). Repeating the straightforward analysis of Taguchi on the model results in this study would add considerable information on this issue.

4. Though not explicitly discussed by the Anonymous Reviewers, the changes in the period of the QBO due to the phase of ENSO (El Nino vs. La Nina) in six of the nine models are small compared to that observed, despite the 3x forcing applied. This should be noted in the abstract, in section 3, and noted and discussed in the summary in section 5. For example, the observational analysis in Taguchi (2010) suggests that a QBO in a perpetual El Nino would have a period of 25 months – 7 months faster than during a perpetual La Nina (a 26% change). Only three of the models in this study feature this amount of change (even under 3 times the observed ENSO forcing), only one of which has a GWD parameterization that responds to changes in convective activity. [Interestingly, all three of these models are also the only models to have an average QBO period that is consistent with that observed (~28 months).]

5. Both reviewers call for ERA 1 reanalysis data to be replaced with ERA 5 data throughout the manuscript and I agree. Also, Reviewer #2 provides references to recent literature that is relevant to this study.

6. Reviewer #2 questions what is learned from this study, given that it is already well established that gravity waves that cannot be explicitly resolved in (even high resolution) AGCMs are important for the driving of the observed QBO and that the response of the QBO to forcing is sensitive to the gravity wave parameterization scheme. The reviewer laments that this phase of the project did not deliver on the promise of a quantitative analysis of the spectral properties of the wave driving in each of the models, which would have made the current study novel. Though I am sympathetic to the Reviewer's concerns, I do see value in the current study, but the revised manuscript should persuasively argue for the merits of the study, given the superficial nature of the analysis. [Certainly, the inability of 4 or 5 of 9 models to simulate a QBO with a realistic period is further evidence of the sensitivity of the QBO to the parameterization of gravity waves (see comment 3 above). See also point 7 below.]

7. Reviewer #2 has made some good suggestions to improve the figure presentations. Moreover, adding observational results (from ERA5) to Figures 11, 12 and 14 would add important observational evidence for how ENSO actually does affect the wave driving, and provide important information for evaluating the efficacy of the ENSO impact on wave driving in the models. These plots would be sufficient to assuage Reviewer #2's comment "What is new?".

8. Figures 8, 9 and 10 are not necessary for the paper. That AGCMs reproduce the observed zonal average changes in circulation has been documented over and again, and the changes in these figures and not useful/used in understanding the impact of ENSO on the QBO (Figs. 11 and 12 are sufficient). Similarly, the text on lines 443-504 should be deleted (it detracts from paper).

Finally, a minor comment on statistical significance: On Line 240, we find " Emphasis will be placed on ... statistically significant at the 95% confidence level." But elsewhere you mention 99% (e.g., Fig. 3 caption and on Line 287). Which is it? Line 241-242 goes on to say "Statistical significance is determined using a two-sided Student's t-test, sampling the maximum individual yearly mean data (e.g., 100 data points for models with 100-year integrations) for both the El Niño and La Niña runs". This is fine for differences in the climatological mean, but not for discerning whether the period of the QBO is different in El Nino vs. La Nina, which has degrees of freedom equal to the number of QBO cycles (minus 1) in each respective regime. Using these degrees of freedom for each model, I find that all of the differences in Fig. 2 are indeed statistically significant at 99%.

Regards,

David Battisti

---

## Author Comment (AC1)

**Reply to the queries and comments of Dr. Paul Pukite**

We very much appreciate Dr. Pukite's comments to our manuscript. In our reply below, we reproduce *the Dr. Pukite's comments in blue italics*, while our replies are in a standard font.

*Decades ago Richard Lindzen came to the conclusion that the quasi-biennial oscillation (QBO) could not be caused by tidal forcing, despite it's obvious potential as a driving mechanism. Negative results are often difficult to find in the literature, but Lindzen mentioned this in two passages:*

- *"For oscillations of tidal periods the nature of the forcing is clear" - Lindzen, Richard D. "Planetary waves on beta-planes." Monthly Weather Review 95.7 (1967): 441-451.*
- *".. it is unlikely that lunar periods could be produced by anything other than the lunar tidal potential" - Lindzen, Richard S., and Siu-shung Hong. "Effects of mean winds and horizontal temperature gradients on solar and lunar semidiurnal tides in the atmosphere." Journal of the atmospheric sciences 31.5 (1974): 1421-1446.*

*At this point of QBO historical data there were only 6 to 8 complete cycles to draw from, yet Lindzen apparently missed the possibility of nonlinear aliasing the lunar cycle against the annual cycle. The only candidate due to QBO wavenumber=0 group symmetry arguments is the 27.2122 day nodal (aka draconic) lunar cycle, which generates a (365.242/27.212) mod 1 = 2.37 year physically aliased repeat period. This matches the historical record, continuing decades later from these early Lindzen studies, see Ref [1].*

*From EGUSPHERE-2024-3270, this passage needs clarification:*

- *"It is clear over this record that the QBO differs somewhat from cycle to cycle (e.g. Quiroz, 1981) and there have been efforts to try to see if the cycle-to-cycle variations may systematically depend on such factors as solar activity, volcanic eruptions or the El Niño/Southern Oscillation (ENSO) cycle of the tropical troposphere (Dunkerton, 1983;*

    *Geller et al., 1997; Salby and Callahan, 2000; Hamilton, 2002, Kane, 2004; Taguchi, 2010). "*

*By "solar activity", one can't imply that is related to sunspot activity, as that is minor compared to the annual or seasonal solar cycle. In fact, the annual solar cycle figures into the same nodal symmetry group as the Semi-Annual Oscillation (SAO) which exists directly above the QBO in altitude. The topological similarity in the nodal driving force behind both the SAO and QBO (the former solar nodal, and the latter solar+lunar nodal) is described in detail in Ref [1].*

**References**

[1] Pukite, P., Coyne, D. and Challou, D. (2018). Wind Energy. In Mathematical Geoenergy (John Wiley & Sons). https://doi.org/10.1002/9781119434351.ch11

We appreciate the interest of Dr. Paul Pukite in our paper.

His points seem to be (a) the QBO might be significantly forced by the lunar gravitational effects in the atmosphere, and (b) that the effects of the 11-year solar activity variation on the atmosphere (which we mention in passing as a topic that has been studied by earlier authors as a possible source of inter-cycle variability in the QBO) are less important than the annual cycle of solar heating, and that this point needs clarification in our manuscript.

With respect to (a), note that we are not addressing this issue in our paper which focuses on the effects of ENSO on the QBO.

With respect to (b) note that the interest of this paper is in the cycle-to-cycle variability of the QBO and so the relevant concern is the interannual variability of solar input into the atmosphere, which is what we refer to as "solar variability". Again, this issue only arises in our background discussion, mentioned as a subject that has been addressed in earlier papers.

---

## Author Comment (AC2)

**Authors' response to the Reviewers' comments on "QBOi El Nino Southern Oscillation experiments: Overview of experiment design and ENSO modulation of the QBO"**
**by Y. Kawatani et al.**

Corresponding author: Yoshio Kawatani (kawatani@ees.hokudai.ac.jp)

We are grateful to the two official referees for their helpful comments/suggestions and to Prof. David Battisti for itemizing the issues addressed in a revised manuscript. Here, we respond to these comments and outline how our proposed revised manuscript will address their concerns. In this reply, we reproduce each Reviewer's comments in blue italics, while our responses are in standard font. We include some figures and tables as part of this response. These figures and tables are labelled "Fig. R1", "Fig R2", "Fig. R3", "Fig. R4" and "Table R1". The other figure and table numbers refer to the original manuscript.

Before presenting our responses to each referee, let us address some overall issues that we would like to share under the general discussion points (I) through (V) below.

**(I) Role of this paper within the QBOi program**

The QBOi Phase 1 project focused on evaluating various aspects of QBO simulation (quality of the QBO in control integrations, modulation associated with global warming, seasonal projection, wave activity etc; see references below) across a range of global circulation models. This foundation enabled the next phase that involves more targeted studies, such as the present QBOi-ENSO project, to explore specific forcings and their impact on the QBO.

The QBOi-ENSO experiments utilize a simplified framework, adding somewhat typical intense ENSO SST anomalies to the annual cycle climatological SST from QBOi Phase-1 Experiment 2 (QBOi Phase-1 experiment 2 used annually repeated climatological SSTs while Experiment 1 employed observed SSTs from January 1979 to February 2009, see Butchart et al. 2018 in more details).

This deliberate simplification isolates the impact of ENSO on the QBO, allowing for a clearer interpretation of the results. While direct comparison with K2019 and observations is indeed more challenging with this design, the amplified anomalies are

expected to increase the signal-to-noise ratio of the response, providing a robust signal to identify key mechanisms and model sensitivities.

This present paper is meant as the initial component of several linked papers, and it specifically aims to provide a comprehensive overview of the QBOi-ENSO experimental design, participating models, and fundamental ENSO influences on the QBO and related meteorological phenomena. This foundational information will be essential for subsequent QBOi-ENSO publications, just as Butchart et al. (2018) served as an overview of the experiment design for QBOi Phase 1. In this paper, Section 2 includes details on the experimental protocol, model descriptions, and data information, similar to Butchart et al. (2018), but also expands upon this with some scientific analysis of ENSO modulation of the QBO, as indicated in the title.

We acknowledge the reviewer's point regarding the complexity of wave-mean flow interactions and the need for more detailed diagnostics. While such in depth analysis was not within the scope of the present paper, it is an important next step. We are currently planning follow-up studies involving detailed wave analyses, similar to Holt et al. (2020) in QBOi Phase-1, using the QBOi-ENSO datasets to address these specific questions. We believe this tiered approach, starting with a broad overview and followed by more specialized investigations, will be the most effective way to disseminate the findings of the QBOi-ENSO project. Therefore, we have reserved more detailed wave analysis for these future publications.

On the other hand, we understand the reviewer's concern and we plan for significant new analyses that will be presented in our revised manuscript. The additional analysis is described in general discussion points (III) and (IV) below

---QBOi phase-1 project----

<Protocol paper>
Butchart, N. et al., 2018: Overview of experiment design and comparison of models participating in the SPARC Quasi-Biennial Oscillation initiative (QBOi), GMD, https://doi.org/10.5194/gmd-11-1009-2018

<Five core papers>
Bushell, A. C. et al. 2020: Evaluation of the Quasi‑Biennial Oscillation in global climate models for

the SPARC QBO‑initiative, QJRMS, https://doi.org/10.1002/qj.3765.

Richter, J. H. et al., 2020: Response of the quasi-biennial oscillation to a warming climate in global climate models, QJRMS, https://doi.org/10.1002/qj.3749.

Anstey, J. A. et al., 2021: Teleconnections of the quasi‑biennial oscillation in a multi‑model ensemble of QBO‑resolving models, QJRMS, https://doi.org/10.1002/qj.4048.

Holt, L. et al. 2020: An evaluation of tropical waves and wave forcing of the QBO in the QBOi models, QJRMS, https://doi.org/10.1002/qj.3827.

Stockdale, T. N., et al. 2020: Prediction of the quasi‑biennial oscillation with a multi‑model ensemble of QBO‑resolving models, QJRMS, https://doi.org/10.1002/qj.3919.

---QBOi-ENSO project----

<papers to be submitted within a period of a few months>

Kawatani et al. 2025: QBOi El Nino Southern Oscillation experiments Part I: Overview of experiment design and ENSO modulation of the QBO, EGUsphere, https://doi.org/10.5194/egusphere-2024-3270, 2024. (Current manuscript. Note that "Part I" will be removed in the revision as we asked)

Naoe et al. 2025: QBOi El Niño Southern Oscillation experiments: Teleconnections of the QBO, WCD discussion, submitted

Elsbury et al. 2025: QBOi El Niño Southern Oscillation experiments: Assessing relationships between ENSO, MJO, and QBO, EGUsphere, https://doi.org/10.5194/egusphere-2024-3950, 2025.

**(II) Strength of ENSO SSTs**

We understand concern regarding the strength of the ENSO SST anomalies used in the QBOi-ENSO experiments as compared to those used in Kawatani et al. 2019 (referred to as K2019, hereafter). We chose to use amplified anomalies to maximize the ENSO signal in the QBO response and clarify the underlying mechanisms. We will explain our rationale and the procedure in detail.

The SST anomalies in K2019 represent a "moderate" ENSO based on observations from all El Niño/La Niña SST anomalies. ENSO SST composites were constructed using data from 1950-2016, based on the Japan Meteorological Agency (JMA) ENSO indices. For each calendar month, El Niño and La Niña events were identified according to the JMA definition. Monthly SST anomalies were then weighted by the corresponding NINO.3 index and averaged to create monthly composite SST anomalies.

This process resulted in "moderate" composite ENSO SST anomalies, as illustrated by the January El Niño example. Seventeen January El Niño events were identified between 1950-2016, with NINO3.4 anomalies ranging from 0.4K to 3.2K. The resulting composite NINO.3 SST anomaly for January was 1.92K, representative of a moderate El Niño event. In the observational record, the highest NINO.3 anomaly value in a January El Niño event is +3.5 K.

To ensure a clear and robust QBO response in our model experiments, we amplified these composite SST anomalies. El Niño anomalies were multiplied by 1.8 and La Niña anomalies by 1.4. These factors were chosen to approximate the peak SST anomalies observed during the strongest El Niño and La Niña events (Table S1, Figure R1). This approach allowed us to better isolate the impact of ENSO on the QBO. We emphasize that SST anomalies employed are not 'unrealistically' large in the sense that actual anomalies of this magnitude are observed on occasion.

Figure R1 shows the annual cycle of the amplified composite ENSO SST anomalies, compared to maximum/minimum observed values. While the compositing procedure cannot perfectly capture the evolution of individual ENSO events, the amplified anomalies exhibit realistic seasonal variations, with El Niño peaking during boreal winter. The variability in La Niña development is also reflected in the composite.

Therefore, while stronger than the anomalies used in K2019, our amplified SSTs remain within the realm of observed ENSO magnitudes, representing the peak values seen during strong events. This approach, like the use of amplified forcings in other climate modeling projects, allows us to better discern the QBO response to a substantial ENSO forcing within the constraints of computational resources and project timelines.

For example, both the QBOi Phase 1 project and various CMIP experiments have employed amplified forcings, such as $2xCO_2/+2K$ SST and even $4xCO_2/+4K$ SST, to investigate climate system responses. While a $4xCO_2/+4K$ SST scenario is unlikely in the near future, the insights gained from such experiments are valuable. Similarly, in the QBOi-ENSO experiments, we prioritize exploring the impacts of strong, yet realistic, ENSO events as a first step. By focusing on the upper end of observed ENSO variability, we can more effectively identify key mechanisms and sensitivities, laying a solid foundation for future research. Therefore, we believe this study offers valuable and meaningful insights into the ENSO-QBO relationship.

It seems that our rationale and conceptual framework were not adequately explained in the initial submission. We intend to address this thoroughly in the revised manuscript. Specifically, we will provide a clearer explanation of the experimental design choices, particularly the decision to use amplified ENSO SST anomalies. We will also articulate more clearly how this study fits within the broader QBOi-ENSO project goals. Furthermore, we will expand on the strategic reasons for focusing on strong ENSO events, given the constraints of computational resources and project timelines. We believe these revisions will significantly improve the clarity and impact of our work.

Also note that we are expanding our study to include analysis of existing AMIP runs with the QBOi models. This analysis will address concern about the imposed SST anomalies in a direct way. See general discussion point (IV) below for more details.

| Month | Jan | Feb | Mar | Apr | May | Jun | Jul | Aug | Sep | Oct | Nov | Dec |
|---|---|---|---|---|---|---|---|---|---|---|---|---|
| El Niño month | 17 | 15 | 11 | 13 | 14 | 18 | 18 | 17 | 18 | 18 | 18 | 17 |
| Max | 3.2 | 2.6 | 2.1 | 1.8 | 2.1 | 2.0 | 2.5 | 2.9 | 3.0 | 3.3 | 3.6 | 3.5 |
| La Niña month | 16 | 15 | 13 | 14 | 12 | 12 | 15 | 15 | 16 | 16 | 16 | 16 |
| Min | -1.8 | -1.5 | -1.0 | -1.3 | -1.4 | -2.0 | -1.6 | -1.6 | -1.3 | -1.6 | -1.7 | -1.8 |

**Table R1**: The number of El Niño and La Niña months during 1950-2016. Max and Min indicate maximum NINO3 anomalies for El Niño and minimum for La Niña (unit: K).

[Figure]

**Figure R1**: The red and blue lines show the delta SSTs in our (a) El Niño and (b) La Niña experiments and represent typical moderate El Niño and La Niña anomalies multiplied by a factor of 1.8 and 1.4, respectively. These monthly delta SSTs are smoothed in time with a 1-2-1 filter. Black lines represent the maximum/minimum observed monthly values during the entire record as shown in Table R1. For visualization, two (exactly repeated) full cycles are shown.

**(III) Adding more detailed analysis of the seasonal and QBO phase dependence of the ENSO influence on the QBO, following Taguchi (2010)**

Following the excellent suggestion to provide a more detailed analysis of the ENSO effects on the QBO mean flow evolution, we will present the same figures as in Taguchi (2010) for our QBOi models, as well as for FUB observational data from 1953 to 2022—extending the dataset by an additional 14 years compared to Taguchi (2010). This allows

us to examine the modulation of QBO amplitude and period as a function of both QBO phases and seasons during El Niño and La Niña.

We have actually now completed this analysis and Figure R3 presents two-dimensional plots of the mean amplitude and mean rate of phase progression for each category, classified by season and QBO phases at 50 hPa (see Fig. R2 below for the QBO phase definition). In our Fig. R3 red shading indicates stronger QBO amplitudes, or faster downward phase progression, during El Niño compared to La Niña.

FUB observations show a weaker QBO amplitude during El Niño in most seasons and phases. Compared to observations, GISS and LMDz simulate significantly enhanced QBO amplitudes during El Niño, as was also evident in Fig. 4 of the original manuscript.

The QBO phase progression rate in FUB observations indicates that QBO phases generally propagate faster during El Niño than during La Niña across most QBO phases and seasons. Most models capture this characteristic behavior. ECHAM5sh produces results consistent with observations for the W and EW phases, but deviates in the E and WE. EMAC also exhibits phase-dependent variations in QBO behavior.

Much faster phase propagation during El Niño is observed, particularly in westerly phases at 50 hPa. The ENSO-related variability of the QBO is most pronounced during the W phase, when the easterly phase is descending from higher levels. Among the models, EC-EARTH appears to best reproduce these characteristics.

In the revised manuscript we will show all these more detailed results for the seasonal and QBO phase-dependent modulation of the QBO by ENSO. We also will show results applying the analysis to earlier AMIP runs with the same model as described in general discussion point (IV) below.

[Figure]

**Figure R2**: Part of a figure from Taguchi (2010) that we reproduce here to show the definitions of WE, E, EW and W phases. (c) Reconstructed QBO zonal wind. (d) Reconstructed wind at 50 (solid line) and 20 (dashed line) hPa. See more details in Taguchi (2010).

**QBO Amplitude**

[Figure]

solid line 95%
dashed line 90%

[Figure]

**Fig.R3**: Two-dimensional plots of mean amplitude and mean rate of phase progression for each category determined by the seasons and QBO phases at 50 hPa. Outlined boxes denote that the numbers are judged to be significantly biased at a 95 % (thick outline) or 90 % (thin outline) confidence level. Line plots for El Nino (red) and La Nina (blue) below the panels show seasonal dependencies when values are averaged in the quadrant (phases). Quadrant (phase) dependencies are similarly shown at the right-hand sides. See more details in Taguchi (2010).

**(IV) Relation between the SST amplitude and the response of the QBO**

An important consideration is the question of how linear with respect to the strength of the SST anomalies is the ENSO modulation of the QBO. For example, Taguchi (2010, https://doi.org/10.1029/2010JD014325) investigated this issue by repeating his observational analyses but only including strong El Niño and La Niña cases.

Taguchi (2010) didn't show these results but states that "The examination shows that the results are generally insensitive to the definitions of the ENSO cases, since the weaker amplitude and faster phase propagation of the QBO are also obtained for the stronger EL conditions." We feel that the limited observational record presents a difficulty in drawing firm conclusions on this issue, but Taguchi's result at least suggests that the use of strong SST anomalies in our experiments is not unreasonable.

While a comprehensive exploration of this is beyond the scope of the present study, we will add a discussion in the revised manuscript. As the editor suggested, one way to investigate how the SST anomaly amplitude modulate the QBO is to run some of the models for ~20 years with moderate SSTs and compare the amplitude of wave forcing. We have access to relevant results from two of the models, namely MIROC-AGCM and MIROC-ESM, which have been run with the present amplified SST anomalies and with the moderate ones used in K2019. Note that unfortunately, the model versions for MIROC-AGCM used in our present study and in K2019 are not identical. For MIROC-ESM the identical model version was used in K2019.

In QBOi-ENSO experiments, MIROC-AGCM and MIROC-ESM show longer periods of the QBO during La Niña than El Niño by about 3.09 months and 1.55 months, respectively. On the other hand, in K2019 experiments using modest SST anomalies, the differences are 2.2 month and 0.4 month (statistically insignificant) respectively. So in these models the ENSO effect on QBO period seems to depend on the strength of the imposed SST anomalies in an intuitively reasonable sense (stronger forcing associated with bigger ENSO-related period change).

Another approach we believe is to apply the same analyses as Taguchi (2010) to the AMIP runs that were conducted earlier as QBOi "Experiment 1". Experiment 1 employed observed SSTs, providing realistic ENSO conditions, from January 1979 to February 2009 (Butchart et al. 2018). Fig. R4 shows the same results as Fig. R3 but for Experiment

1 using the MIROC-AGCM-LL. Our analysis of the AMIP results has just begun and this result for MIROC-AGCM-LL is based on a single realization of ~30 years. But note that we have available integrations for a total of 3 realizations for MIROC-AGCM-LL and for several other of the models, and in our revision we will include results for all available runs.

**Fig. R4:** The same as Fig. R3 but showing results from the QBOi Experiment 1 (an AMIP experiment with observed SST from January 1979 to February 2009) using one realization of the MIROC-AGCM-LL model simulation.

**(V) Using ERA5 reanalysis data instead of ERA-I**

Both reviewers suggested using ERA5 reanalysis data instead of ERA-I. Accordingly, we will use ERA5 data in the revised manuscript. Additionally, to address Reviewer #2's comment, *"What is new?"*, as recommended by the editor, we will incorporate observational results from ERA5 into Figures 11 and 14.

ERA5 provides zonal mean *"Mean eastward wind tendency due to parameterizations"*, which includes not only non-orographic gravity waves but also other parameterized forcing. Therefore, we cannot include reanalysis results in Fig. 12 but we can potentially include them in Fig. 14 to illustrate resolved and parameterized forcing. We will include explanations and a note of caution regarding this distinction in the revised manuscript.

**Reply to anonymous referee #1**

*Review of the article QBOi El Nino Southern Oscillation experiments Part I: Overview of experiment design and ENSO modulation of the QBO by Yoshio Kawatani et al.*

*General comments*

*This work is a continuation of the publication series produced by the QBOi project, based on an experimental protocol and several models known to simulate the QBO. This work is focused on the El Niño/La Niña effects on the QBO as simulated in 9 models. Most of the article describes the common and different features found in the different model simulations, and its structure follows the work of Kawatani et al. (2019), hereafter K2019, where they investigated the El Niño/La Niña effects on the QBO in MIROC models.*

*The experimental design chosen here is however deviating from that of K2019. Here they decided to use amplified mean El Niño/La Niña SST anomalies. This makes any direct comparison to K2019 and to observations difficult. Whether or not the QBO response should be linear to the amplitude of the SST anomaly pattern is not discussed. Probably it would have been better to use the same anomalies as in K2019. (Or an entirely different design based on SST fields of selected El Niño and La Niña years.)*

Please refer to our general discussion point (II) for this response. We will include additional explanations regarding the comparison between the ENSO SST anomalies in K2019 and the present study in the revised manuscript.

*They find that El Niño/La Niña effects on the QBO period are qualitatively similar with respect to the period, with El Niño leading to a shorter period despite of the increased tropical upwelling in the tropical lower stratosphere, from which it is clear that El Niño must also produces a stronger wave mean flow interaction. No common response is found for the QBO amplitude.*

*An interesting part is the discussion and analysis of the reasons for the described results: The more equatorial precipitation and the weaker Walker circulation found during El Niño conditions. These features are found in all models, and they probably are independent of the skill of a model to simulate a QBO. The discussion of the wave mean flow processes is however more complicated, because of the rather different ways that*

*resolved and parameterized waves generate the QBOs in the different models. And therefore not so much can be learned from this part, except that there exists still a considerable difference in the way that models generate QBOs. Further, as acknowledged by the authors, more detailed model diagnostics would be needed to learn more about the underlying reasons for the found behaviours. But this additional diagnostics was not part of the protocol, or the modelling groups could not produce these diagnostics.*

We briefly mentioned the datasets used in the present analysis and referenced Butchart et al. (2018). As noted in Butchart et al. (2018), 6-hourly data on temperature, as well as zonal, meridional, and vertical wind, are also available in the QBOi-ENSO experiments. We will provide a more detailed explanation in the revised manuscript. For further on this point refer to our general discussion point (I) above.

*Overall I think that the publication is worthwhile, as it creates a baseline for further work on this topic. Some minor corrections are needed before publication.*

We sincerely appreciate the Reviewer's positive evaluation and valuable comments. We are grateful that you find our work worthwhile.

*Detailed comments and questions*

*Abstract*
*L40 Stratosphere-troposphere Processes And their Role in Climate (SPARC) ...*
*As we know QBOi has been started as a SPARC project. But SPARC has changed its name to APARC and QBOi is now listed as an APARC project. Maybe it is worth to add a remark or a footnote on this aspect.*

We will fix this.

*L45 ... models -models ... should probably be ... models. Models ...*

We will fix this.

*1 Introduction*
*L64 ... that QBO facilitates ... → ... that the QBO facilitates ...*

We will fix this.

*L140 ... Conducting a common ENSO-QBO experiment across a range of QBO-resolved climate models could help elucidate the role of non-orographic GWP in driving the oscillation. ...*
*The work of Richter et al. (2020) on the climate warming effects on the QBO unfortunately showed that the differences between GWPs are considerable and probably responsible for the rather different QBO responses to the warming. As it seems it was not possible to decide which GWPs were "wrong" or "right". Now a similar exercise is presented aiming at El Niño/La Niña variations in SST as the external forcing instead of a warmer SST and increased atmospheric CO2. Why should we expect a scientifically more robust result if Richter et al. (2020) have shown that differences in parameterizing non-orographic gravity drag can lead to very different results? Simply because El Niño/La Niña cycles exist in the historical period for which the models have been tuned?*

We acknowledge your point regarding the challenges in fully elucidating the role of non-orographic GWP. While Richter et al. (2020) examined QBO modulation in a future climate—where direct observational data to validate changes are unavailable—our QBOi-ENSO experiments can be partially validated using existing observations, such as the observed shortening of QBO periods during El Niño compared to La Niña. Although our experimental design is somewhat idealized, it allows us to identify models that produce longer QBO periods during El Niño runs as potentially problematic, prompting a closer evaluation of their GWP parameterizations. We believe this is a key advantage of the QBOi-ENSO experiments over the future climate scenario examined in Richter et al. (2020). Based on your feedback, we will revise the manuscript to clarify this reasoning.

**2. Model Description and Experimental Design**
*L179 ... These factors bring the peak composite anomaly SSTs closer to the anomalies observed during the most intense El Niño and La Niña events. ...*
*Using amplification factors is problematic. This makes a comparison to observations or to the work by K2019 difficult. It seems necessary to add some remarks about the linearity between the SST pattern amplitude and the response of the QBO. Can this be assumed? Alternatively you could have chosen specific years with strong El Niño and La Niña SST anomalies. Then there would be no need to amplify the SST anomaly, and there would be less of a risk to construct an SST anomaly pattern that mixes the different types of El Niños, which are discussed in literature.*

Please refer to our general discussion point (II) above.

*L199 ... For clarity and conciseness, we will refer to these models as CESM1, EC-EARTH, ECHAM, EMAC, GISS, LMDz, MIROC-AGCM, MIROC-ESM and MRICESM1, respectively. ...*
*The abbreviated model names are introduced here, but not used consequently. Tables, Figures, and also some sentences use the full model names. Please decide whether short names shall be used or not. But if you decide to use short names, then please use these in all places: tables, figures, and text.*

We appreciate your comments and will make sure the revised version uses a consistent system of model names.

*L203 ... Launch levels for parameterized gravity waves varied across models, ranging from 450 to 700 hPa or 1000 to 100 hPa. ...*
*To which model(s) do the two pressure ranges relate? Please clarify.*

Table 1 shows the launch levels. We will add "(see Table 1)" in the revised manuscript.

*Table 1. lunched level → launch level*

We will fix this.

*Table 2. What does the entry for GISS-E2-2G and Residual stream function (5-1115✓) mean?*

We will fix this.

*L243 ... from the ERA-Interim (ERA-I; Dee et al. 2011) reanalyses ...*
*Why is ERA-I used for this comparison, when ERA-5 is now available? Newer reanalyses are generally improved compared to earlier ones.*

Please refer to our general discussion point (V) for this response. We will use ERA-5 in the revised manuscript.

*L247 … Importantly, the composite ERA-I and CMAP data were not scaled …*
*This is a kind of a flaw in the experimental design. If the response to the SST anomaly patterns is non-linear to the amplitude, then the applied scaling is hindering a direct comparison to observations or analyses. If, however the signal is linear, then the signals derived from ERA-I should be scaled like the SST patterns used for the simulations.*

Please refer to general discussion points (II) and (IV) for this response.

**3. ENSO Modulation of the QBO and Climatological Mean Field Differences**

*L266 – L276 These lines discuss deficiencies in the structure of the simulated QBO, as occurring in El Niño or La Niña simulations of ECHAM, GISS, and LMDz. In my opinion it is necessary to point out another deficiency, which is an unrealistic period, although a regular pattern of downward propagating westerly and easterly jets is simulated. Taking the displayed 20 years (Fig. 2) of the El Niño and La Niña simulations together, we have 40 years for which on average (40years / 28 months) we would expect about 17 cycles. A count of the cycles shown in Figure 2 can now serve as an additional measure for the quality of the simulations. If we allowed a range of 15 to 19, then the following models (here excluding ECHAM, GISS and LMDz) would fail: EC-EARTH: 20, MIROC-AGCM-LL: 26, MIROC-ESM: 21. Please extend your discussion of problematic simulations in this direction, so that the reader knows from the beginning which model simulations need to be viewed a bit more critically.*

We appreciate your comments regarding the unrealistic QBO periods in QBOi model simulations. A detailed analysis of simulated QBO periods in QBOi "Experiment 2" (with a climatological annual cycle of SSTs) was previously conducted by Bushell et al. (2020). However, we recognize the importance of addressing this issue directly in the context of our current study on the QBOi-ENSO experiment. Therefore, we will incorporate a more extensive discussion of the simulated QBO periods in the revised manuscript. This will provide readers with a clearer understanding of the model limitations and enable them to assess the simulations more critically from the outset.

Bushell, A. C., J. A. Anstey, N. Butchart, Y. Kawatani, S. M. Osprey, J. H. Richter, F. Serva, P. Braesicke, C. Cagnazzo, C.-C. Chen, H.-Y. Chun, R. R. Garcia, L. J. Gray, K. Hamilton, T. Kerzenmacher, Y.-H. Kim, F. Lott, C. McLandress, H. Naoe, J. Scinocca, T. N. Stockdale, S. Watanabe, K. Yoshida, S. Yukimoto: Evaluation of the Quasi‐Biennial Oscillation in global climate models for

the SPARC QBO‑initiative, Quarterly Journal of the Royal Meteorological Society, https://doi.org/10.1002/qj.3765, 2020

*L311 ... Next, we consider ENSO modulation of QBO amplitude, which is known less robust ... → ... Next, we consider the ENSO modulation of the QBO amplitude, which is known to be less robust ...*

We will fix this.

*L323 ... GISS, LMDz, and CESM1, all of which have variable GWP sources. ...*
*I think it should be added that MIROC-AGCM-LL has variable gravity waves too, though these are explicitly simulated, within the given resolution, instead of parameterized. Thus variability of gravity waves not necessarily leads to a strong amplitude difference between El Niño and La Niña. And one needs to wonder if a strong change is indicating that the variability of gravity waves is important aspect for a GWP, or whether this effect is rather a result of other aspects of parameterizing gravity wave. Please add some thoughts on this problem.*

While MIROC-AGCM-LL can simulate a QBO-like oscillation without parameterized non-orographic GWP, previous studies have indicated that its resolution (T106L72, corresponding to a 1.25-degree horizontal resolution and 550-meter vertical resolution) is insufficient to capture wave forcing as effectively as higher-resolution models, such as T213L256 or even higher-resolution models (e.g., Kawatani et al., 2010). In MIROC-AGCM-LL, most of the unresolved gravity wave forcing that is parameterized in other models is not explicitly simulated. Therefore, we believe that the MIROC-AGCM-LL results do not necessarily support the conclusion that 'variability of gravity waves does not necessarily lead to a strong amplitude difference between El Niño and La Niña.'

Parameterized GWP represents sub-grid-scale processes on much smaller scales than ~100 km. The variable source of parameterized GWP is often linked to cumulus convection, which also generates gravity waves on much smaller scales than those resolved in MIROC-AGCM-LL. We will clarify this distinction in the revised manuscript.

Kawatani, Y., K. Sato, T. J. Dunkerton, S. Watanabe, S. Miyahara and M. Takahashi, 2010: The roles of equatorial trapped waves and internal inertia-gravity waves in driving the quasi-biennial oscillation. Part I: Zonal mean wave forcing, J. Atmos. Sci., 67, 963-980., https://doi.org/10.1175/2009JAS3222.1

*L383 … although models tend to simulate the precipitation peak to the east of the observed one over the central Pacific in the El Niño run. …*
*It should also be mentioned that the precipitation peak in the model simulations is higher than in observations, which indicates that the local forcing by latent heat release in the simulations is higher than that explained by the observed precipitation. Quite likely this is related to the amplified El Niño/La Niña SST strength.*

We acknowledge your point that the precipitation peak in the model simulations is higher than in observations. It is important to note that the observed precipitation data we are using represents averaged values weighted by the NINO3.4 index, reflecting precipitation patterns during "moderate" ENSO events. While we could alternatively present precipitation distributions during strong El Niño events, such as the 1997 event, we believe that this level of detail is not essential for the core message of this manuscript. However, we will add further explanation regarding this issue in the revised manuscript to ensure clarity for the reader

*L422 … significantly deep westerly difference … → … significantly deeper westerly difference …*

We will fix this.

*L432 … for (left top) ERA-I …*
*ERA-I is "center top"*

We will fix this.

**4 Contrasting wave forcing and residual mean meridional circulations in El Niño and La Niña from QBOi models**

*L458 … The X term represents any other unresolved forcing. … Do you mean here parameterized momentum diffusion and effects from numerical diffusion and damping operators?*

Indeed this represents all the other possible contributions including explicitly parameterized diffusion and any other contributions from the numerical schemes

employed.

*L470 … in El Niño and La Niña simulations. …*
*La Niña simulations are nor shown, but differences of the El Niño and La Niña simulations.*

We will change the text, thank you.

*L477 … La Niña c annual … → … La Niña annual …*

We will change the text, thank you.

*L498 … which both use variable sources in their GWP, …*
*Do you mention this because you think that this is the reason for the differences? Often other differences in the formulation of the non-orographic gravity wave drag parameterizations can cause substantial differences already.*

We acknowledge that we mentioned the variable GWP sources in these models based on what was visually apparent in the figure, without directly attributing them as the sole cause of the observed differences. We agree with your assessment that other differences in the formulation of non-orographic gravity wave drag parameterizations could also contribute substantially to these differences. While it is difficult to definitively determine the specific reasons at this stage, we will take this point into account in the revised manuscript.

*L509 … averaged over 20°S–20°N …*
*Maybe it is worth to explain why a band of 20°S – 20°N is chosen, while earlier diagnostics/figures used narrower bands. (I guess this is made in order to remove residuals of the secondary meridional circulation of the QBO.)*

We selected the 20°S–20°N latitude band for averaging because this region encompasses the area where the QBO amplitude is most pronounced. While other latitude bands, such as 10°S–10°N or 15°S–15°N, could also be used, they tend to exhibit more noise. However, we have confirmed that our basic conclusions remain largely unaffected by the choice of these latitudinal ranges. In the revised manuscript, we will add a brief explanation regarding this choice, acknowledging the potential rationale behind selecting

wider bands to minimize noisier vertical structures.

*L510 ... ranging from approximately 0.2 mm s¹ in MIROC-AGCM to approximately 0.4 mm s¹ ⁻in LMDz. ...*
*This strong difference in the tropical upwelling implies also a strong difference in the strength of wave mean flow interaction that is necessary to simulate a QBO with a realistic period. This aspect is not discussed here, and maybe this El Niño/La Niña related article is the wrong place. Still it directly shows that the wave mean flow interaction must work at different strengths.*

Weak tropical upwelling in the MIROC model is discussed in Kawatani et al. (2010). In addition, the tropical upwelling is also different among reanalysis, as discussed in the S-RIP final report SPARC (2022). We will add a brief discussion about tropical upwelling related with a QBO in the revised manuscript.

SPARC, 2022: SPARC Reanalysis Intercomparison Project (S-RIP) Final Report. Masatomo Fujiwara, Gloria L. Manney, Lesley J. Gray, and Jonathon S. Wright (Eds.), SPARC Report No. 10, WCRP-6/2021, doi: 10.17874/800dee57d13, available at www.sparc-climate.org/publications/sparc-reports.

*L522 ... However, the specific altitudes at which $w*$ changes would most strongly influence the overall QBO period remain unclear. ...*
*Sentence unclear.*

We will modify this sentence to
"However, it remains unclear which specific altitudes where changes in w* occur have the strongest influence on the overall QBO period."

*L579 ... While output data of parameterized gravity wave fluxes in LMDz were not available at the time of this analysis, this model, which also uses variable parameterized wave sources related to precipitation activity, showed similar structures affected by precipitation distributions (Dr. Lott, personal communication). ...*
*Francois Lott is a co-author of this study. Please include the LMDz results in Figure 12.*

Francois Lott informed us that the datasets were quantitatively incorrect due to inadequate processing. However, we have confirmed that the qualitative distribution is related to precipitation, similar to what is observed in CESM1.

*L620 … parameterized wave forcing below is stronger …*
*What does "below" refer to? Maybe the sentence needs to be rephrased.*

We delete "below" in this sentence.

*L639 … As discussed for Fig. 14, … → … As discussed for Fig. 13, …*

We will fix this.

**5. Summary and concluding remarks**

*L685 … remained consistent …*
*"consistent" seems to be the wrong term, because this could have different meanings. (If El Niño/La Niña influences the ozone distribution, then the same ozone field cannot be consistent with El Niño and La Niña conditions at the same time.) "unchanged" would express more clearly that these fields simply have not been changed.*

We will change from consistent to unchanged following your suggestion, thank you.

**Reply to anonymous Reviewer #2**

*Review of the manuscript "QBOi El Nino Southern Oscillation experiments Part I: Overview of experiment design and ENSO modulation of the QBO" by Kawatani et al.*

*Summary*

*The manuscript presents an overview of the experimental design from the new SPARC QBOi project and examines the modulation of the QBO by ENSO using nine climate models. The findings indicate that the QBO period is longer during La Nina compared to El Nino across all models, consistent with observations. However, changes in the QBO amplitude remain inconclusive. Overall, I find the experiment intriguing, the manuscript well-written, and the results clearly explained. Most of my comments are minor and focus on improving consistency between different parts of the manuscript and aligning the figures with the text.*

We sincerely appreciate the Reviewer's positive evaluation and valuable comments. We are grateful that you find our work worthwhile.

*One major comment, however, concerns the lack of deeper insights into ENSO modulation of the QBO. The authors attribute this to the simplicity of the analyses and the limited availability of model output data, which they suggest prevents a full explanation of the quantitative differences in QBO between El Nino and La Nina. While future studies are mentioned as a potential avenue to address this, I argue that if a more detailed analysis is feasible, it should be included in this paper, as it was the primary motivation for the experiment and study.*

Please refer to our general discussion point (I) above for a consideration of the role of this paper within the broader QBOi program which helps motivate our work. Our plans for revision include significantly more analysis. In particular, following the Editor's suggestion we will repeat the more detailed analysis of Taguchi (2010) to elucidate the seasonal effects in the QBO-ENSO connections. We also are in the process of expanding our study to include analysis of the earlier AMIP runs (QBOi "Experiment 1") to supplement the new "annually repeating" runs that was the main focus of our paper.

***Major Comments***

*1) Lack of additional insight into the mechanisms of ENSO modulation of the QBO*

*My primary concern is whether this paper and the associated experiments provide any additional insight into the mechanisms by which ENSO modulates the QBO. At the start of the paper, I had hoped—likely in line with the motivation behind designing and implementing these experiments—that this study would offer a deeper understanding of these mechanisms. However, the study appears to be an extension of Kawatani et al. (2019), with potentially more models included beyond MIROC, yet missing important analyses due to data limitations.*

*There are repeated statements such as: "Further investigation of these models is hampered by the incomplete model variables in the available data sets", "This simple analysis with limited model output data cannot fully explain quantitative differences in QBO periods between El Nino and La Nina", and "Detailed zonal-time spectral analyses of model fields, like those performed in Kawatani et al. (2019), remain a subject for future study."*

*If such analyses are indeed possible, this paper is the appropriate venue to present them, rather than postponing them to future studies. For example, as the authors mentioned, detailed spectral analyses of the EP flux, gravity wave parameterization fluxes, precipitation, or momentum budgets based on the TEM framework could offer crucial insights into the intermodel spread of QBO period and amplitude.*

*To provide further context on my disappointment, Kawatani et al. (2019) noted: "It would be interesting to analyze the ENSO modulation of the three-dimensional wave forcing as well as tropical upwelling, which must show large differences between El Nino and La Nina. This may be investigated in a future study". Now, five years later, this study states: "A detailed investigation of the three-dimensional distributions of parameterized wave fluxes modulated by ENSO, including model dependence, would be of interest and remains a topic for future research."*

*It feels like an opportunity has been missed to address these outstanding questions. If there is a way to conduct these analyses, I strongly encourage the authors to include them in this paper.*

We do believe that our manuscript already contains a substantial amount of useful material and it provides the background and introduction for related QBOi studies please refer to our general discussion point (I) above. However, we understand the reviewer's concern and will include further analysis of the simulations following the editor's suggestion, as explained in our general discussion point (III) above as well as a substantial expansion of our study to include analysis of earlier AMIP runs as explained in general discussion point (IV) below

*2) Lack of use of recent data and citations of recent studies*
*Some aspects of the study, including citations and the data used, feel somewhat outdated. For instance, the use of ERA-I instead of ERA5. Additionally, the study only uses observed data up to 2012. If this limitation is due to avoiding the QBO disruptions, there are still several years of data available between 2012 and the end of 2015, as well as between 2020 and 2024. While including these additional years may not change the main conclusions of the paper, it would enhance the robustness of the analysis, particularly for slowly evolving phenomena like ENSO and QBO, where even a few more samples could provide valuable insights. Moreover, the citations miss some relevant and recent studies, such as Zhou et al. (2024), and a few others noted in my review.*

We appreciate this concern and we will repeat all our analyses with ERA5 data as explained in general discussion point (V). We will also include the recent references suggested.

***Minor comments***
*L47-49: It can also be mentioned that "all models simulate stronger equatorial tropical upwelling in El Nino compared to La Nina up to ~10 hPa".*

We will mention this.

*L85: Small-scale gravity waves also contribute significantly to the forcing of the QBO westerly (e.g., Pahlavan et al. (2021))*

We will refer to this paper.

*L95: As a good reference on this you can cite Coy et al. (2020).*

We will refer to this paper.

*In general, the figures can be significantly improved by reducing redundancy, which would allow for larger, clearer panels. For instance, in Figure 2, use "El Nino" as the title for the left column and "La Nina" for the right, rather than repeating them for each panel. Similarly, list model names only on the left side of the figure and show the y-axis (0–20) only on the bottom panels, instead of repeating it in every panel. These changes can enhance readability and apply to other figures as well.*

We will fix these issues in the revised manuscript, thank you.

*The other general issue with the figures is the presence of too many contours, which reduces readability. In particular, in Figures 4, 8, 9, 10, and 13, the contours over the shadings can be removed, similar to Figure 12, to improve clarity.*

In our revised version we will adopt these suggestions to improve legibility of these figures.

*Figure 3: Have you analyzed each phase of the QBO separately? For example, do both phases of the QBO become shorter during El Nino?*

This is an excellent suggestion and was also made by the Editor. We have now repeated the analysis segregated by QBO phase and season and will include these results in the revised version. Please refer to general discussion point (III) for details.

*Figure 3: Have you considered using a Fourier Transform to determine the period instead of relying on zero wind line crossing (e.g., as done in Lee et al. (2024))? While it likely won't change the conclusions, it might be a better option, particularly when the QBO becomes more irregular/unrealistic, as seen during El Nino in ECHAM.*

We appreciate the suggestion to use a Fourier Transform to determine the QBO period, as demonstrated in Lee et al. (2024). We agree that FFT methods are particularly useful when defining the phase transition of the QBO is challenging (we employed FFT methods in our QBOi phase-1 paper, Richter et al. 2020). However, in the current QBOi-ENSO

experiments, all models have a clearly defined QBO phase transition at 20 hPa. Therefore, we prefer to determine the QBO period by phase transition, following the approach of K2019. As you noted, this choice is unlikely to alter the overall conclusions of our study.

*L412: Will the cooler anomaly around 60°N–90°N in ERA-I, which is not observed in the models, change if more data is included, such as using ERA-5 from 1940 to 2024?*

We did not find this even in ERA-5, as we will show in the revised manuscript.

*For Figures 10, 11, 13, and 14, you could consider including results from reanalysis (e.g., ERA5) as a reference, similar to what is done in Figures 8 and 9.*

We will do this. Once again refer to our general discussion point (V).

*Figure 12: Could you add the total flux for El Nino and La Nina (i.e., averaged over 10°S–10°N and all longitudes) to the bottom panels? If so, is it consistently larger during El Nino?*

Thank you for the suggestion. We appreciate the value of including the total flux for El Niño and La Niña (averaged over 10°S–10°N and all longitudes) in Figure 12. While we are open to adding this information, we believe it would be best presented in a separate supplementary figure to avoid overcrowding the existing figure. We will carefully consider this option and its potential impact on the clarity of the manuscript.

*Editorial comments*
*L84: "respectively" seems redundant.*

We will fix this.

*L107: You can cite (Richter et al., 2020) again to avoid ambiguity.*

We will fix this.

*L108: SST is not yet defined.*

We will fix this.

*L140: "QBO-resolved" -> "QBO-resolving"*

We will fix this.

*L164: GWP is already defined.*

We will fix this.

*L165: What is experiment 2?*

This means what is defined as "experiment 2" in the QBOi phase 1 project. We will explain this in the revised manuscript.

*L165: SST needs to be defined at L108.*

We will fix this.

*L198: The model name "CESM15-110L" is mentioned here, while "WACCM5-110L" is used in the results (figures and tables). I suggest selecting one naming convention for consistency.*

We will fix this, thank you.

*L200: Using the concise version of model names is a great choice, but it would be helpful to maintain this approach consistently in the results (figures and tables) as well. Currently, there is a discrepancy where the text uses concise names while the results use the full model names, making it harder to follow.*

As we mentioned in our response to the other reviewer, we will have a consistent nomenclature for the models in the revised version.

*L198: "CESM15-110L" is mentioned here but in the results (figures and tables) "WACCM5-110L" is used. I suggest choosing one for consistency.*

We will fix this.

*L200: It is great to use the more concise version of the model names but it would be great to use the concise version in the results as well (figures and tables) to make it easier to follow. Now, there is this discrepancy between the text and results, the former using the concise version, and the latter the full name of the models.*

As mentioned above, we will fix this and have a consistent nomenclature in our revised manuscript.

*L220: Palmerio et al. (2022) is not in the bibliography.*

We will fix this, thank you.

*L238: TEM is already defined.*

We will fix this.

*Table 2: what is "5-1115" in front of GISS.*

This is our mistake. We will fix this.

*L284: "larger" -> "longer"(?)*

We will fix this.

*L295: "with" -> "for"*

We will fix this.

*L375: ITCZ not defined yet.*

We will fix this.

*L378: "(left-top)" -> "(center-top)"*

We will fix this.

*L392: Any reference for this statement?*

We will refer to the relevant paper in the revised manuscript.

*Figure 7: "PRCP" not defined.*

We will fix this.

*L427: BDC is already defined.*

We will fix this.

*L431: "(left-top)" -> "(center-top)"*

We will fix this.

*L442: A point after 4 is missing. "In El Nino and La Nina from QBOi models" is redundant. Also, capitalize the first letter to maintain consistency with the other titles.*

We will fix this.

*L443: "eddy forcing" -> "wave forcing", to be consistent with the other sections.*

We will fix this.

*L443: "mean zonal" -> "zonal mean"*

We will fix this.

*L444: TEM is already defined.*

We will fix this.

*L445: "eddies" -> "waves"*

We will fix this.

*L566-567: "WACCM" -> CESM1*

We will fix this.

*L577: ITCZ should be defined earlier at L375.*

We will fix this.

*L620: "below" should be removed.*

We will fix this.

*L621: "below" -> "above" (?)*

We will fix this.

*L639: "Fig. 14" -> "Fig. 13"*

We will fix this.

*Caption of Fig. 14: "eddy" -> "wave". "resolved motions" -> "resolved forcing".*

We will fix this.

*References*

*Coy, L., Newman, P. A., Strahan, S., & Pawson, S. (2020). Seasonal Variation of the Quasi-Biennial Oscillation Descent. Journal of Geophysical Research: Atmospheres, 125(18), e2020JD033077.https://doi.org/10.1029/2020JD033077*

*Geller, M. A., Zhou, T., & Yuan, W. (2016). The QBO, gravity waves forced by tropical convection, and ENSO. Journal of Geophysical Research: Atmospheres, 121(15), 8886–8895. https://doi.org/10.1002/2015JD024125*

*Kawatani, Y., Hamilton, K., Sato, K., Dunkerton, T. J., Watanabe, S., & Kikuchi, K. (2019). ENSO Modulation of the QBO: Results from MIROC Models with and without Nonorographic Gravity Wave Parameterization. https://doi.org/10.1175/JAS-D-19-0163.1*

*Lee, H.-K., Chun, H.-Y., Richter, J. H., Simpson, I. R., & Garcia, R. R. (2024). Contributions of Parameterized Gravity Waves and Resolved Equatorial Waves to the QBO Period in a Future Climate of CESM2. Journal of Geophysical Research: Atmospheres, 129(8), e2024JD040744. https://doi.org/10.1029/2024JD040744*

*Pahlavan, H. A., Wallace, J. M., Fu, Q., & Kiladis, G. N. (2021). Revisiting the Quasi-Biennial Oscillation as Seen in ERA5. Part II: Evaluation of Waves and Wave Forcing. Journal of the Atmospheric Sciences, 78(3), 693–707. https://doi.org/10.1175/JAS-D-20-0249.1*

*Richter, J. H., Butchart, N., Kawatani, Y., Bushell, A. C., Holt, L., Serva, F., et al. (2020). Response of the Quasi-Biennial Oscillation to a warming climate in global climate models. Quarterly Journal of the Royal Meteorological Society, n/a(n/a). https://doi.org/10.1002/qj.3749*

*Zhou, T., DallaSanta, K. J., Orbe, C., Rind, D. H., Jonas, J. A., Nazarenko, L., et al. (2024). Exploring the ENSO modulation of the QBO periods with GISS E2.2 models. Atmospheric Chemistry and Physics, 24(1), 509–532. https://doi.org/10.5194/acp-24-509-2024*

We appreciate these relevant recent references. We will include them in the revised manuscript.

---

## Author Comment (AC3)

**Authors' response to Editor's comments on "QBOi El Nino Southern Oscillation experiments: Overview of experiment design and ENSO modulation of the QBO" by Y. Kawatani et al.**

We thank the Editor for carefully itemizing the concerns raised and offering valuable suggestions for improving the revised manuscript. Our responses to each of the Editor's comments are included below. For clarity, we have included the Editor's comments in *blue italics* and our responses in regular font. Note that our responses here refer to numbered "general discussion points" (I-V) that are included in our responses to the Reviewers' comments.

*Dear Prof. Kawatani and co-authors:*
*Both Anonymous referees have posted their comments on your manuscript (WCD 2024-3270). As per WCD policy, you are now to post a response on how you will address the referee's comments – after which I will make a decision on the manuscript.*

*Both reviewers have made excellent comments on the manuscript and call for major revisions which I agree with, particularly concerning the issues of "what is new? (Reviewer #2)", the unexplored impact of unrealistically large forcing on the response (Reviewer #1), the lack of a discussion of agreement between the simulated QBOs and those observed (particularly the period) (Reviewer #1), and the use of very old reanalysis data throughout the manuscript (both Reviewers). To provide guidance in revising the manuscript so that it is acceptable for publication in WCD, below I itemize the issues that I expect will addressed in a revised manuscript. I will also post these on the WCD page for the manuscript. `*

We appreciate your efforts in thoroughly itemizing the issues.

*1. Reviewer #1 asks for an examination and justification of the impact of applying an ENSO forcing with an unrealizable amplitude. This should be addressed in the revised manuscript. One way to do this quantitatively is to run some of the models for ~20 years with realistic forcing and show how this impacts the wave forcing, say by producing Figs. 11 and 12 and comparing the amplitude of the wave forcing in the 3x El Nino with that with a 1x El Nino (and ditto for La Nina forcing).*

We address this concern in our response to the Reviewers, specifically in our general

discussion point (II). Your reference to "3 x El Nino" and "ENSO forcing with an unrealizable amplitude" is not completely clear to us. We constructed our El Niño forcing SST anomalies by first compositing over all El Niño events which produced a result appropriate for a typical moderate El Niño. Then, to amplify the effect, we multiplied that composite SST field by a factor 1.8. This choice avoids an "unrealizable" amplitude as our imposed peak NINO3 anomalies are then similar to those in the strongest observed El Niño events. The same considerations led us to multiply our composited La Niña SST by a factor of 1.4.

We will add to our revised manuscript a series of calculations that analyze the El Niño vs La Niño contrast in QBO amplitude and period as simulated in earlier AMIP runs by each of the models included in our study. Actually, this will amount to a repetition of Taguchi's (2010) analysis of the observed record. This directly addresses the issue of analyzing model runs with realistic SST anomalies. The comparison of the AMIP results with results from our runs with larger (on average) "annually repeating" SST will be interesting. Also, in alignment with the Editor's comment #3 below, we will include in this analysis the seasonal effects as revealed in Taguchi's analysis.

*2. Reviewer #1 calls for a necessary discussion of a deficiency in half the model models to simulate a QBO with a period that is consistent with that observed (see Reviewer #1 comments on Lines 266-76), and I agree. The reviewer's argument shows that four of the models have QBO periods that are unrealistic (EC-EARTH: 20, MIROC-AGCM-LL: 26, MIROC-ESM: 21). I note that GISS is on the edge of being disqualified by this measure, and the unrealistic QBO in the La Nina simulations is a reason to add this model to the disqualified list. Hence, three of the six models in which wave driving is examined in Figs. 13 and 14 have QBOs with unrealistic periods. These issues should be mentioned in the abstract, in section 3, and in the discussion/summary in section 5.*

In our revision we will include a discussion of these issues in line with your suggestions here.

*3. Reviewer #2 asks whether the changes in the QBO period are sensitive to the method used to define a QBO cycle and whether ENSO impacts all phases of the QBO simulated by the models. I expect the method used to provide a robust period. However, it is not clear from the analysis presented whether El Nino accelerates all phases of the QBO, as is seen in the observations – or whether it impacts certain phases (such as the downward*

*propagation of the westerly shear zone). In most of the models, the slope of the constant phase lines in the vertical-time plots with El Nino forcing experiment is indistinguishable from the slope in the La Nina forcing experiment (e.g., in Fig 2 for CESM1, EMAAC, MIROC-AGCM, MIROC-ESM and MRIESM2). Only in EC-Earth3.3 is the slope of the phase lines steeper with El Nino forcing than with La Nina forcing, as is also the case in the observations (Taguchi 2010, cf Fig. 9a with 9b). Repeating the straightforward analysis of Taguchi on the model results in this study would add considerable information on this issue.*

We appreciate your helpful suggestions. In the revised manuscript, we will incorporate new analysis results based on the methodology described in Taguchi (2010). Please see our response to the Reviewers general discussion point (III) for further details. As noted above, we plan to apply this analysis to our "annually repeating" runs and to earlier AMIP runs conducted with the same models.

*4. Though not explicitly discussed by the Anonymous Reviewers, the changes in the period of the QBO due to the phase of ENSO (El Nino vs. La Nina) in six of the nine models are small compared to that observed, despite the 3x forcing applied. This should be noted in the abstract, in section 3, and noted and discussed in the summary in section 5. For example, the observational analysis in Taguchi (2010) suggests that a QBO in a perpetual El Nino would have a period of 25 months – 7 months faster than during a perpetual La Nina (a 26% change). Only three of the models in this study feature this amount of change (even under 3 times the observed ENSO forcing), only one of which has a GWD parameterization that responds to changes in convective activity. [Interestingly, all three of these models are also the only models to have an average QBO period that is consistent with that observed (~28 months).]*

We appreciate your insightful comments regarding the changes in the QBO period due to the ENSO phase. You correctly point out that the changes in the QBO period in six of the nine models are small compared to observations. We agree that this discrepancy should be addressed in the revised manuscript.

As you suggest, we will add a note to the abstract and section 3, and provide a more detailed discussion in the summary (section 5). We will specifically address the observational analysis in Taguchi (2010), which can be interpreted as indicating a 26% change in the QBO period between perpetual El Niño and La Niña conditions. We will

also discuss the fact that only three models exhibit this magnitude of change, and that these models are also the only ones with an average QBO period consistent with observations.

*5. Both reviewers call for ERA 1 reanalysis data to be replaced with ERA 5 data throughout the manuscript and I agree. Also, Reviewer #2 provides references to recent literature that is relevant to this study.*

We will use ERA5 in the revised manuscript. Please see our general discussion point (V) in our response to the Reviewers for further details.

*6. Reviewer #2 questions what is learned from this study, given that it is already well established that gravity waves that cannot be explicitly resolved in (even high resolution) AGCMs are important for the driving of the observed QBO and that the response of the QBO to forcing is sensitive to the gravity wave parameterization scheme. The reviewer laments that this phase of the project did not deliver on the promise of a quantitative analysis of the spectral properties of the wave driving in each of the models, which would have made the current study novel. Though I am sympathetic to the Reviewer's concerns, I do see value in the current study, but the revised manuscript should persuasively argue for the merits of the study, given the superficial nature of the analysis. [Certainly, the inability of 4 or 5 of 9 models to simulate a QBO with a realistic period is further evidence of the sensitivity of the QBO to the parameterization of gravity waves (see comment 3 above). See also point 7 below.]*

We thank the Editor for acknowledging the value of this study. We will strengthen the manuscript by more clearly articulating the merits of our work, recognizing the limitations of our current analysis. As noted earlier, we also will quite significantly expand the scope of our study by including analysis of the QBO effects of ENSO as simulated in earlier AMIP runs. We also acknowledge the Editor's point that the inability of several models to simulate a realistic QBO period further emphasizes the QBO's sensitivity to gravity wave parameterization, and we will emphasize this connection in our revised discussion.

*7. Reviewer #2 has made some good suggestions to improve the figure presentations. Moreover, adding observational results (from ERA5) to Figures 11, 12 and 14 would add important observational evidence for how ENSO actually does affect the wave driving,*

*and provide important information for evaluating the efficacy of the ENSO impact on wave driving in the models. These plots would be sufficient to assuage Reviewer #2's comment "What is new?".*

Thank you for the suggestions. We will follow the Editor's advice and include ERA5 results in our revised Figures to provide observation-based context for the impact of ENSO on wave driving. We believe this addition will help address Reviewer #2's concerns regarding the novelty of the study.

*8. Figures 8, 9 and 10 are not necessary for the paper. That AGCMs reproduce the observed zonal average changes in circulation has been documented over and again, and the changes in these figures and not useful/used in understanding the impact of ENSO on the QBO (Figs. 11 and 12 are sufficient). Similarly, the text on lines 443-504 should be deleted (it detracts from paper).*

Thank you for your suggestions. We acknowledge your point that Figures 8, 9, and 10, and the corresponding text (lines 443-504), may not be strictly necessary for this paper. However, we believe these figures, illustrating the fundamental zonal wind, temperature, wave forcing, and residual circulation, are valuable and relevant. These figures align with those presented in K2019 and are intended to support future QBOi-ENSO research.
In light of your feedback, we are prepared to move these figures and the related text to the Supplementary Materials to streamline the manuscript and maintain a clear focus.

*Finally, a minor comment on statistical significance: On Line 240, we find "Emphasis will be placed on ... statistically significant at the 95% confidence level." But elsewhere you mention 99% (e.g., Fig. 3 caption and on Line 287). Which is it? Line 241-242 goes on to say "Statistical significance is determined using a two-sided Student's t-test, sampling the maximum individual yearly mean data (e.g., 100 data points for models with 100-year integrations) for both the El Niño and La Niña runs". This is fine for differences in the climatological mean, but not for discerning whether the period of the QBO is different in El Nino vs. La Nina, which has degrees of freedom equal to the number of QBO cycles (minus 1) in each respective regime. Using these degrees of freedom for each model, I find that all of the differences in Fig. 2 are indeed statistically significant at 99%.*

You are correct in identifying the discrepancy regarding statistical significance levels. The 99% confidence level is indeed appropriate for Figure 3 and the QBO period

differences, while the 95% level applies to the climatological mean differences. We appreciate your careful attention to detail and will explain this more clearly in the revised manuscript.

---

## Author Response (AR1)

**Authors' response to Editor's comments on "QBOi El Nino Southern Oscillation experiments: Overview of experiment design and ENSO modulation of the QBO" by Y. Kawatani et al.**

We thank the Editor for carefully summarizing the concerns raised and offering valuable suggestions for improving our manuscript. Our responses to each of the Editor's comments are included below. For clarity, we have included the Editor's comments in *blue italics* and our responses in regular font. Note that our responses here refer to numbered "general discussion points" (I-V) that are included in our responses to the Reviewers' comments.

*Dear Prof. Kawatani and co-authors:*
*Both Anonymous referees have posted their comments on your manuscript (WCD 2024-3270). As per WCD policy, you are now to post a response on how you will address the referee's comments – after which I will make a decision on the manuscript.*

*Both reviewers have made excellent comments on the manuscript and call for major revisions which I agree with, particularly concerning the issues of "what is new? (Reviewer #2)", the unexplored impact of unrealistically large forcing on the response (Reviewer #1), the lack of a discussion of agreement between the simulated QBOs and those observed (particularly the period) (Reviewer #1), and the use of very old reanalysis data throughout the manuscript (both Reviewers). To provide guidance in revising the manuscript so that it is acceptable for publication in WCD, below I itemize the issues that I expect will addressed in a revised manuscript. I will also post these on the WCD page for the manuscript. `*

We have appreciated your efforts in thoroughly summarizing the issues.

*1. Reviewer #1 asks for an examination and justification of the impact of applying an ENSO forcing with an unrealizable amplitude. This should be addressed in the revised manuscript. One way to do this quantitatively is to run some of the models for ~20 years with realistic forcing and show how this impacts the wave forcing, say by producing Figs. 11 and 12 and comparing the amplitude of the wave forcing in the 3x El Nino with that with a 1x El Nino (and ditto for La Nina forcing).*

We address this concern in our response to the Reviewers, specifically in our general

discussion point (II). The reference in the comments to "3 x El Nino" and "ENSO forcing with an unrealizable amplitude" is not completely clear to us. We constructed our El Niño forcing SST anomalies by first compositing over all El Niño events which produced a result appropriate for a typical moderate El Niño. Then, to amplify the effect, we multiplied that composite SST field by a factor 1.8. This choice avoids an "unrealizable" amplitude as our imposed peak NINO3 anomalies are then similar to those in the strongest observed El Niño events. The same considerations led us to multiply our composited La Niña SST by a factor of 1.4.

We have added to our revised manuscript a series of calculations that analyze the El Niño vs La Niño contrast in QBO amplitude and period as simulated in earlier AMIP runs by each of the models included in our study. Actually, this amount to a repetition of Taguchi's (2010) analysis of the observed record. This directly addresses the issue of analyzing model runs with realistic SST anomalies. The comparison of the AMIP results with results from our runs with larger (on average) "annually repeating" SST is discussed in our revised manuscript.. Also, in alignment with the Editor's comment #3 below, we have included in this analysis the seasonal effects as revealed in Taguchi's analysis.

*2. Reviewer #1 calls for a necessary discussion of a deficiency in half the model models to simulate a QBO with a period that is consistent with that observed (see Reviewer #1 comments on Lines 266-76), and I agree. The reviewer's argument shows that four of the models have QBO periods that are unrealistic (EC-EARTH: 20, MIROC-AGCM-LL: 26, MIROC-ESM: 21). I note that GISS is on the edge of being disqualified by this measure, and the unrealistic QBO in the La Nina simulations is a reason to add this model to the disqualified list. Hence, three of the six models in which wave driving is examined in Figs. 13 and 14 have QBOs with unrealistic periods. These issues should be mentioned in the abstract, in section 3, and in the discussion/summary in section 5.*

We appreciate the Editor's thoughtful comments regarding the realism of QBO periods in some of the model simulations. In response to the Editor and Reviewer's concerns, we have added discussion  of the unrealistic QBO periods in the abstract, in Section 3 and in the summary. Please also see the reply to reviewer 1.

*3. Reviewer #2 asks whether the changes in the QBO period are sensitive to the method used to define a QBO cycle and whether ENSO impacts all phases of the QBO simulated by the models. I expect the method used to provide a robust period. However, it is not*

*clear from the analysis presented whether El Nino accelerates all phases of the QBO, as is seen in the observations – or whether it impacts certain phases (such as the downward propagation of the westerly shear zone). In most of the models, the slope of the constant phase lines in the vertical-time plots with El Nino forcing experiment is indistinguishable from the slope in the La Nina forcing experiment (e.g., in Fig 2 for CESM1, EMAAC, MIROC-AGCM, MIROC-ESM and MRIESM2). Only in EC-Earth3.3 is the slope of the phase lines steeper with El Nino forcing than with La Nina forcing, as is also the case in the observations (Taguchi 2010, cf Fig. 9a with 9b). Repeating the straightforward analysis of Taguchi on the model results in this study would add considerable information on this issue.*

We appreciate these helpful suggestions. In the revised manuscript, we have incorporated new analysis results based on the methodology described in Taguchi (2010). Please see our response to the Reviewers general discussion point (III) for further details. As noted above, we have applied this analysis to our "annually repeating" runs and to earlier AMIP runs conducted with the same models.

*4. Though not explicitly discussed by the Anonymous Reviewers, the changes in the period of the QBO due to the phase of ENSO (El Nino vs. La Nina) in six of the nine models are small compared to that observed, despite the 3x forcing applied. This should be noted in the abstract, in section 3, and noted and discussed in the summary in section 5. For example, the observational analysis in Taguchi (2010) suggests that a QBO in a perpetual El Nino would have a period of 25 months – 7 months faster than during a perpetual La Nina (a 26% change). Only three of the models in this study feature this amount of change (even under 3 times the observed ENSO forcing), only one of which has a GWD parameterization that responds to changes in convective activity. [Interestingly, all three of these models are also the only models to have an average QBO period that is consistent with that observed (~28 months).]*

We appreciate these insightful comments regarding the changes in the QBO period due to the ENSO phase. In the revision we have added the following sentences:

"*The simulated La Niña periods range from 6.9 % to 42.5 % longer than those during El Niño, compared to an observed difference of approximately 27.2 %*" [added to the abstract.

"*Yuan et al. (2014) estimated long-term means for the QBO period of 25 months for El*

*Niño conditions and 31.8 months for La Niña conditions, corresponding to a 27.2% difference. Only three of the nine models (EC-EARTH, LMDz, and ECHAM) simulate La Niña–El Niño differences in QBO period that approach this observed sensitivity, even under the amplified ENSO forcing used in this study*" [added tosection 3].

"*It is noted here that only three of the nine models (EC-EARTH, LMDz, and ECHAM) simulate La Niña–El Niño differences in QBO period that approach the observed sensitivity (~27 %), even under the amplified ENSO forcing. The remaining six models exhibit more modest ENSO modulation of the QBO period*" [Added to the summary section].

*5. Both reviewers call for ERA 1 reanalysis data to be replaced with ERA 5 data throughout the manuscript and I agree. Also, Reviewer #2 provides references to recent literature that is relevant to this study.*

We have used ERA5 for the observations throughout the revised manuscript. Please see our general discussion point (V) in our response to the Reviewers for further details.

*6. Reviewer #2 questions what is learned from this study, given that it is already well established that gravity waves that cannot be explicitly resolved in (even high resolution) AGCMs are important for the driving of the observed QBO and that the response of the QBO to forcing is sensitive to the gravity wave parameterization scheme. The reviewer laments that this phase of the project did not deliver on the promise of a quantitative analysis of the spectral properties of the wave driving in each of the models, which would have made the current study novel. Though I am sympathetic to the Reviewer's concerns, I do see value in the current study, but the revised manuscript should persuasively argue for the merits of the study, given the superficial nature of the analysis. [Certainly, the inability of 4 or 5 of 9 models to simulate a QBO with a realistic period is further evidence of the sensitivity of the QBO to the parameterization of gravity waves (see comment 3 above). See also point 7 below.]*

We thank the Editor for acknowledging the value of this study. We have strengthened the manuscript by more clearly articulating the merits of our work, while recognizing the limitations of our current analysis. As noted earlier, we also have quite significantly expanded the scope of our study by including analysis of the QBO effects of ENSO as simulated in earlier AMIP runs. We also acknowledge the Editor's point that the inability

of several models to simulate a realistic QBO period further emphasizes the QBO's sensitivity to gravity wave parameterization, and we have emphasized this connection in our revised discussion.

*7. Reviewer #2 has made some good suggestions to improve the figure presentations. Moreover, adding observational results (from ERA5) to Figures 11, 12 and 14 would add important observational evidence for how ENSO actually does affect the wave driving, and provide important information for evaluating the efficacy of the ENSO impact on wave driving in the models. These plots would be sufficient to assuage Reviewer #2's comment "What is new?".*

Thank you for the suggestions. We have followed the Editor's advice and included ERA5 results in our revised figures to provide observation-based context for the impact of ENSO on wave driving. We believe this addition helps address Reviewer #2's concerns regarding the novelty of the study.

*8. Figures 8, 9 and 10 are not necessary for the paper. That AGCMs reproduce the observed zonal average changes in circulation has been documented over and again, and the changes in these figures and not useful/used in understanding the impact of ENSO on the QBO (Figs. 11 and 12 are sufficient). Similarly, the text on lines 443-504 should be deleted (it detracts from paper).*

Thank you for your suggestions. We acknowledge your point that original Figures 8, 9, and 10, and the corresponding text, may not be strictly necessary for this paper. However, we believe these figures, illustrating the fundamental zonal wind, temperature, wave forcing, and residual circulation, are valuable and relevant. These figures align with those presented in K2019. In light of this feedback, we have moved these figures and the related text to the Supplementary Materials to streamline the manuscript and maintain a clear focus.

*Finally, a minor comment on statistical significance: On Line 240, we find "Emphasis will be placed on ... statistically significant at the 95% confidence level." But elsewhere you mention 99% (e.g., Fig. 3 caption and on Line 287). Which is it? Line 241-242 goes on to say "Statistical significance is determined using a two-sided Student's t-test, sampling the maximum individual yearly mean data (e.g., 100 data points for models with 100-year integrations) for both the El Niño and La Niña runs". This is fine for differences*

*in the climatological mean, but not for discerning whether the period of the QBO is different in El Nino vs. La Nina, which has degrees of freedom equal to the number of QBO cycles (minus 1) in each respective regime. Using these degrees of freedom for each model, I find that all of the differences in Fig. 2 are indeed statistically significant at 99%.*

The Editor is correct in identifying the discrepancy regarding statistical significance levels. We modified the sentence as follows:

"*Consistent with the observational study by Taguchi (2010), all models simulated longer periods during La Niña compared to El Niño runs, a difference statistically significant at the ≥ 99% confidence level for each model (based on a two-sided Student's t-test using the number of QBO cycles in each simulation as the degrees of freedom)*"

**Authors' response to the Reviewers' comments on "QBOi El Nino Southern Oscillation experiments: Overview of experiment design and ENSO modulation of the QBO"**
**by Y. Kawatani et al.**

Corresponding author: Yoshio Kawatani (kawatani@ees.hokudai.ac.jp)

We are grateful to the two official referees for their helpful comments/suggestions and to the Editor, Dr. Battisti for summarizing the issues to be addressed in a revised manuscript. We earlier submitted our initial detailed response to the Reviewers' comments, with an outline of our proposed revisions to the original manuscript. Now we have completed the revised version and this response largely reproduces our earlier comments but now with a description of how the revised manuscript was actually changed.

We would also like to note that our revised version includes a new Supplementary Material document consisting of five sections, one table (Table S1), and seven figures (Figs. S1–S7). All supplementary figures and the table are cited in the main text. The main text remains focused on presenting the core results and conclusions, while the supplementary material provides additional background and context.

In this reply, we reproduce each Reviewer's comments in *blue italics*, while our responses are in standard font. We include some figures and tables as part of this response. These figures and tables are labelled "Fig. R1-R7" and " Table R1". The other figure and table numbers refer to the original manuscript.

Before presenting our responses to each referee, let us address some overall issues that we would like to share under the general discussion points (I) through (V) below. Our specific responses to individual comments begin on page 17 below.

**(I) Role of this paper within the QBOi program**

The QBOi Phase 1 project focused on evaluating various aspects of QBO simulation (quality of the QBO in control integrations, modulation associated with global warming, seasonal projection, wave activity etc., see references below) across a range of global circulation models. This foundation enabled the next phase that involves more targeted studies, such as the present QBOi-ENSO project, to explore specific forcings and their

impact on the QBO.

The QBOi-ENSO experiments utilize a simplified framework, adding somewhat typical intense El Niño or La Niña SST anomalies to the observed annual cycle 1979-2009 climatological SST.

This deliberate simplification isolates the impact of ENSO on the QBO, allowing for a clearer interpretation of the results. While direct comparison with K2019 and observations is indeed more challenging with this design, the amplified anomalies are expected to increase the signal-to-noise ratio of the response, providing a robust signal to identify key mechanisms and model sensitivities.

This present paper is meant as the initial component of several linked papers, and it specifically aims to provide a comprehensive overview of the QBOi-ENSO experimental design, the participating models, and the fundamental ENSO influences on the QBO and related meteorological phenomena. This foundational information will be essential for subsequent QBOi-ENSO publications, just as Butchart et al. (2018) served as an overview of the experiment design for QBOi Phase 1. In this paper, Section 2 includes details on the experimental protocol, model descriptions, and data information, similar to Butchart et al. (2018), but also expands upon this with some scientific analysis of ENSO modulation of the QBO, as indicated in the title.

The Reviewers and Editor raised the issue of the complexity of wave-mean flow interactions and the need for more detailed diagnostics and that such analysis could add to the contribution of this paper. While such in depth analysis was not within the scope of the present paper, it is an important next step. We are currently planning follow-up studies involving detailed wave analyses, similar to Holt et al. (2020) in QBOi Phase-1, using the QBOi-ENSO datasets to address these specific questions. During the international QBOi conference held earlier this year (March 2025; Cambridge, UK) QBOi project participants discussed in detail how to finalize the present paper, and how to proceed in the future. At this meeting a team of co-authors was established to write the next wave analysis paper. We believe this tiered approach, starting with a broad overview and followed by more specialized investigations, will be the most effective way to disseminate the findings of the QBOi-ENSO project. Therefore, we have reserved more detailed wave analysis for these future publications.

On the other hand, we understand the Reviewers' concern about the extent of new contributions made by this paper and we have added significant new analyses to our revised manuscript. The additional analysis is described in general discussion points (III) and (IV) below

Here is a list of the published core papers (2018-2021) in QBOi phase-1

<Protocol paper>

Butchart, N. et al., 2018: Overview of experiment design and comparison of models participating in the SPARC Quasi-Biennial Oscillation initiative (QBOi), GMD, https://doi.org/10.5194/gmd-11-1009-2018.<Five core papers>

Bushell, A. C. et al. 2020: Evaluation of the Quasi‐Biennial Oscillation in global climate models for the SPARC QBO‐initiative, QJRMS, https://doi.org/10.1002/qj.3765.

Richter, J. H. et al., 2020: Response of the quasi-biennial oscillation to a warming climate in global climate models, QJRMS, https://doi.org/10.1002/qj.3749.

Anstey, J. A. et al., 2021: Teleconnections of the quasi‐biennial oscillation in a multi‐model ensemble of QBO‐resolving models, QJRMS, https://doi.org/10.1002/qj.4048.

Holt, L. et al. 2020: An evaluation of tropical waves and wave forcing of the QBO in the QBOi models, QJRMS, https://doi.org/10.1002/qj.3827.

Stockdale, T. N., et al. 2020: Prediction of the quasi‐biennial oscillation with a multi‐model ensemble of QBO‐resolving models, QJRMS, https://doi.org/10.1002/qj.3919.

Here is the list of papers that have been published in WCD discussion, as productions of the QBOi-ENSO project:

Kawatani et al. 2025: QBOi El Nino Southern Oscillation experiments Part I: Overview of experiment design and ENSO modulation of the QBO, EGUsphere [preprint], https://doi.org/10.5194/egusphere-2024-3270, 2024. (Note that "Part I" is removed in the revised title)

Naoe et al. 2025: QBOi El Niño Southern Oscillation experiments: Teleconnections of the QBO, EGUsphere [preprint], https://doi.org/10.5194/egusphere-2025-1148, 2025.

Elsbury et al. 2025: QBOi El Niño Southern Oscillation experiments: Assessing relationships between ENSO, MJO, and QBO, EGUsphere [preprint], https://doi.org/10.5194/egusphere-2024-3950, 2025.

**(II) Strength of ENSO SSTs**

We understand concerns raised by the Reviewers and Editor regarding the strength of the ENSO SST anomalies used in the QBOi-ENSO experiments as compared to those used in Kawatani et al. 2019 (referred to as K2019, hereafter). We chose to use amplified anomalies to maximize the ENSO signal in the QBO response so helping our efforts to clarify the underlying mechanisms. Here we explain our rationale and the procedure in detail.

The SST anomalies in K2019 represent a "moderate" ENSO based on observations from all El Niño/La Niña SST anomalies. ENSO SST composites were constructed using data from 1950-2016, based on the Japan Meteorological Agency (JMA) ENSO indices. For each calendar month, El Niño and La Niña events were identified according to the JMA definition. Monthly SST anomalies were then weighted by the corresponding NINO.3 index and averaged to create monthly composite SST anomalies.

This process resulted in "moderate" composite ENSO SST anomalies, as illustrated by the January El Niño example. Seventeen January El Niño events were identified between 1950-2016, with NINO3.4 anomalies ranging from 0.4K to 3.2K. The resulting composite NINO.3 SST anomaly for January was 1.92K, representative of a moderate El Niño event. In the observational record, the highest NINO.3 anomaly value in a January during an El Niño event is +3.5 K.

To ensure a clear and robust QBO response in our model experiments, we amplified these composite SST anomalies. El Niño anomalies were multiplied by 1.8 and La Niña anomalies by 1.4. These factors were chosen to approximate the peak SST anomalies observed during the strongest El Niño and La Niña events (Table R1, Fig. R1). This approach allowed us to better isolate the impact of ENSO on the QBO. We emphasize that SST anomalies employed are not 'unrealistically' large in the sense that actual anomalies of this magnitude are observed on occasion.

Figure R1 shows the annual cycle of the amplified composite ENSO SST anomalies, compared to maximum/minimum observed values. While the compositing procedure cannot perfectly capture the evolution of individual ENSO events, the amplified anomalies exhibit realistic seasonal variations, with El Niño peaking during boreal winter. The variability in La Niña development is also reflected in the composite.

Therefore, while stronger than the anomalies used in K2019, our amplified SSTs remain within the realm of observed ENSO magnitudes, representing the peak values seen during strong events. This approach, like the use of amplified forcings in other climate modeling projects, allows us to better discern the QBO response to a substantial ENSO forcing within the constraints of computational resources and project timelines.

For example, both the QBOi Phase 1 project and various CMIP experiments have employed amplified forcings, such as $2xCO_2/+2K$ SST and even $4xCO_2/+4K$ SST, to investigate climate system responses. While a $4xCO_2/+4K$ SST scenario is unlikely in the near future, the insights gained from such experiments are valuable. Similarly, in the QBOi-ENSO experiments, we prioritize exploring the impacts of strong, yet realistic, ENSO events as a first step. By focusing on the upper end of observed ENSO variability, we can more effectively identify key mechanisms and sensitivities, laying a solid foundation for future research. Therefore, we believe this study offers valuable and meaningful insights into the ENSO-QBO relationship.

It seems that our rationale and conceptual framework were not adequately explained in the initial submission. We have addressed this in the revised manuscript. Specifically, we have provided a clearer explanation of the experimental design choices, particularly the decision to use amplified ENSO SST anomalies. We have also articulated more clearly how this study fits within the broader QBOi-ENSO project goals. Furthermore, we have expanded on the strategic reasons for focusing on strong ENSO events, given the constraints of computational resources and project timelines. We believe these revisions should significantly improve the clarity and impact of our work. Most of these clarifications have been added to Section 2: "Model Description and Experimental Design," as well as to Supplementary Section 1: "Strength of ENSO SSTs used in the experiments."

Also note that we have expanded our study to include analysis of existing AMIP runs with the QBOi models. This analysis addresses the concern about the imposed SST anomalies in a direct way. See general discussion point (IV) below for more details.

| Month | Jan | Feb | Mar | Apr | May | Jun | Jul | Aug | Sep | Oct | Nov | Dec |
|---|---|---|---|---|---|---|---|---|---|---|---|---|
| El Niño month | 17 | 15 | 11 | 13 | 14 | 18 | 18 | 17 | 18 | 18 | 18 | 17 |
| Max | 3.2 | 2.6 | 2.1 | 1.8 | 2.1 | 2.0 | 2.5 | 2.9 | 3.0 | 3.3 | 3.6 | 3.5 |
| La Niña month | 16 | 15 | 13 | 14 | 12 | 12 | 15 | 15 | 16 | 16 | 16 | 16 |
| Min | -1.8 | -1.5 | -1.0 | -1.3 | -1.4 | -2.0 | -1.6 | -1.6 | -1.3 | -1.6 | -1.7 | -1.8 |

**Table R1**: The number of El Niño and La Niña months during 1950-2016. Max and Min indicate maximum NINO3 anomalies for El Niño and minimum for La Niña, respectively (unit: K). This table has been incorporated as Table S1 in the Supplementary Material of the revised manuscript.

[Figure]

**Figure R1**: The red and blue lines show the delta SSTs in our (a) El Niño and (b) La Niña experiments and represent typical moderate El Niño and La Niña anomalies multiplied by a factor of 1.8 and 1.4, respectively. These monthly delta SSTs are smoothed in time with a 1-2-1 filter. Black lines represent the maximum/minimum observed monthly values during the entire record as shown in Table R1. For visualization, two (exactly repeated) full cycles are shown. Original Figures 1c,d were replaced with this Figure R1 in the revised manuscript.

**(III) Adding more detailed analysis of the seasonal and QBO phase dependence of the ENSO influence on the QBO, following Taguchi (2010)**

Following the excellent suggestion to provide a more detailed analysis of the ENSO effects on the QBO mean flow evolution, we first extended Taguchi's (2010) analysis of the observed FUB/KIT radiosonde-based equatorial zonal wind time series to data through 2022. Then we repeated his analysis for the existing AMIP runs for each of our QBOi models. Then, in the revised manuscript we present the same figures as in Taguchi (2010) for our QBOi models, as well as for FUB/KIT radiosonde observational data (https://www.atmohub.kit.edu/english/807.php) from 1953 to 2022. This allows us to examine the modulation of QBO amplitude and period as a function of both QBO phases and seasons during El Niño and La Niña.

Fig. R2 and R3 present two-dimensional plots of the mean rate of phase progression and mean amplitude for each category, respectively, classified by season and QBO phases at 50 hPa (see Fig. R4 for the QBO phase definition). In our Figs. R2 and R3 red shading indicates faster downward phase progression, or stronger QBO amplitudes, during El Niño compared to La Niña.

Radiosonde observations show a weaker QBO amplitude during El Niño in most seasons and phases (Fig. R3). Contrary to this observed feature, GISS and LMDz simulate significantly enhanced QBO amplitudes during El Niño, as was also evident in Fig. 4 of the original manuscript.

The QBO phase progression rate in the FUB/KIT observations indicates that QBO phases generally propagate faster during El Niño than during La Niña across most QBO phases and seasons. Much faster phase propagation during El Niño is observed, particularly in westerly phases at 50 hPa (see phases presented in Fig. R4), when the easterly phase is descending from higher levels.

Most models capture this characteristic behavior. Among the models, EC-EARTH appears to best reproduce these characteristics. ECHAM produces results consistent with observations for the W and EW phases, but deviates in the E and WE (see figure R4 below as well as the section S2 of the Supplementary Material for the definition of W, EW, WE, E phases).

In the method of Taguchi (2010), an empirical orthogonal function (EOF) analysis is applied to QBO zonal wind data from 70 hPa to 10 hPa. The amplitude and phase progression rate of the QBO are derived from the first and second EOF components. However, in the El Niño experiment of the ECHAM, the EOF coefficients for the QBO deviate significantly from an elliptical trajectory in phase space; that is, the shape appears distorted or collapsed. The equatorial zonal wind data from the El Niño experiment (see Fig. 2) indicate that the westerly phase is weak around 30 hPa, and the easterly phase exhibits a vertically confined structure. Since the phase progression rate is measured as the angle around the origin in phase space, this distortion likely results in less useful estimates.

In Section 3 of the revised manuscript, we have included all these more detailed results for the seasonal and QBO phase-dependent modulation of the QBO by ENSO.

[Figure]

**Figure R2**: Composite differences in QBO phase progression rate (unit: degrees per month) between El Niño and La Niña conditions, classified by QBO phase at 50 hPa and season. The top-center panel shows results from radiosonde observations, and the remaining panels show outputs from nine QBOi models. Red (blue) shading indicates faster (slower) phase progression during El Niño. Green outlines denote statistically significant differences at the 90 % confidence level. Right and bottom subpanels show seasonal and QBO-phase averages, respectively. The two values in the lower-right corner

indicate domain-mean values for La Niña (top) and El Niño (bottom), and the line plot shows their distributions across season and QBO phase (blue for La Niña, red for El Niño). This figure has been incorporated as Fig.5 in the main text of the revised manuscript.

[Figure]

**Figure R3**: Same as Fig. R2, but for QBO amplitude, defined as the distance from the origin in the two-dimensional phase space constructed from the first two EOF components of equatorial zonal wind anomalies. Red (blue) shading indicates stronger

(weaker) amplitude during El Niño. Mean values and composite lines follow the same format as in Fig. 5. This figure has been incorporated as Fig.6 in the main text of the revised manuscript.

[Figure]

**Figure R4**: Part of a figure from Taguchi (2010) that we reproduce here to show the definitions of WE, E, EW and W phases. (a) Reconstructed QBO zonal wind. (b) Reconstructed wind at 50 (solid line) and 20 (dashed line) hPa. See more details in Taguchi (2010). This figure has been incorporated as Fig.S1 in the Supplementary Material of the revised manuscript.

**(IV) Relation between the SST amplitude and the response of the QBO**

An important consideration is the question of how linear with respect to the strength of the SST anomalies is the ENSO modulation of the QBO. For example, Taguchi (2010, https://doi.org/10.1029/2010JD014325) investigated this issue by repeating his observational analyses but only including strong El Niño and La Niña cases.

Taguchi (2010) did not explicitly show these results, but states that "The examination shows that the results are generally insensitive to the definitions of the ENSO cases, since the weaker amplitude and faster phase propagation of the QBO are also obtained for the stronger EL conditions." We feel that the limited observational record presents a difficulty in drawing firm conclusions on this issue, but Taguchi's result at least suggests that the use of strong SST anomalies in our experiments is not unreasonable.

As the Editor suggested, one way to investigate how the SST anomaly amplitude modulate the QBO is to run some of the models for ~20 years with moderate SSTs and

compare the amplitude of wave forcing. We have access to relevant results from two of the models, namely MIROC-AGCM and MIROC-ESM, which have been run with the present amplified SST anomalies and with the moderate ones used in K2019. Note that unfortunately, the model versions for MIROC-AGCM used in our present study and in K2019 are not identical. For MIROC-ESM the identical model version was used in K2019.

In QBOi-ENSO experiments, MIROC-AGCM and MIROC-ESM show longer periods of the QBO during La Niña than El Niño by about 3.09 months and 1.55 months, respectively. On the other hand, in K2019 experiments using more modest SST anomalies, the differences are 2.2 month and 0.4 month (statistically insignificant) respectively. So in these models the ENSO effect on QBO period seems to depend on the strength of the imposed SST anomalies in an intuitively reasonable sense (stronger forcing associated with bigger ENSO-related period change).

As described above, another approach we adopted to investigate this issue was to apply the same analyses as Taguchi (2010) to the AMIP runs that were conducted earlier as QBOi "Experiment 1 (Exp1)". Exp1 employed observed SSTs, providing realistic ENSO conditions, from January 1979 to February 2009, with each ensemble integrated for 31 years (Butchart et al. 2018).

The EC-EARTH and GISS models did not participate in Exp1, so results from the remaining seven models are discussed here. One ensemble member is available for each of ECHAM5sh, EMAC, LMDz, and MRI-ESM, while three ensemble members are available for each of MIROC-AGCM, MIROC-ESM, and CESM1. All these datasets were used in our analyses.

To identify El Niño and La Niña events, we applied the same classification method as in Taguchi (2010), selecting months corresponding to these conditions from each 31-year AMIP simulation. Although this approach allows us to isolate ENSO phases for comparison, it should be noted that the total number of El Niño and La Niña samples is considerably smaller than in the perpetual ENSO experiments, leading to increased sampling uncertainty.

Moreover, while the perpetual ENSO experiments are specifically designed to isolate the effects of ENSO by prescribing fixed SST anomalies, the AMIP simulations in Exp1 are

forced by observed SSTs, sea ice, and external radiative forcings. As a result, they include the influence of other interannual and decadal climate variability in addition to ENSO. This makes the ENSO signal in Exp1 potentially less robust and more affected by external variability, thereby complicating direct comparison with the more idealized ENSO-only experiments. Nevertheless, we believe that the comparison remains meaningful and provides valuable insight into ENSO-related modulation of the QBO under more realistic boundary conditions.

Figs. R5 and R6 show the same results as Figs. R2 and R3 but for Experiment 1. Several categories lack data due to the reduced number of samples compared to the ENSO experiments. Similar characteristics to those found in the ENSO experiments are also observed in Exp1. The QBO amplitude differences between El Niño and La Niña vary across models, while all models show a faster phase progression rate for the mean values (averaged over all analyzed periods).

EMAC, LMDz, MIROC-AGCM, and CESM1 all display larger amplitude differences in the ENSO experiments compared to Exp1 (note that ECHAM5sh also shows larger differences, but with the opposite sign). Regarding the phase progression rate, ECHAM5sh, LMDz, and MIROC-AGCM show larger differences, while EMAC, MIROC-ESM, and CESM1 show much smaller differences.

Based on these results, in addition to our previous discussion on the perpetual ENSO experiments (particularly the reduced ENSO amplitude in MIROC-AGCM and MIROC-ESM) and the findings of Taguchi (2010), we conclude that ENSO amplitude influences the qualitative modulation of the QBO. However, the effect may not be simply linear.

While a comprehensive exploration of this issue is beyond the scope of the present study, we added a discussion in Section 6.1: "Discussion" in the revised main text. A more detailed explanation with additional figures (Figs. S5-7) is also included in the supplementary Section 5: "ENSO modulation of the QBO in the QBOi experiment 1".

[Figure]

**Figure R5:** The same as Fig. R2, but showing results from the QBOi Experiment 1 (an AMIP experiment with observed SST from January 1979 to February 2009).

[Figure]

**Figure R6:** The same as Fig. R3, but showing results from the QBOi Experiment 1.

[Figure]

**Figure R7:** Mean QBO amplitude and phase progression rate in the ENSO experiments and Exp1, along with FUB/KIT observations. Dashed lines represent FUB/KIT observations. Filled squares indicate results from the ENSO experiments, while open circles represent Exp1. Upward-pointing vectors correspond to stronger amplitude or faster phase progression during El Niño compared to La Niña, and vice versa for downward-pointing vectors.

**(V) Using ERA5 reanalysis data instead of ERA-I**

Both reviewers suggested using ERA5 reanalysis data instead of ERA-I. Accordingly, we repeated all our calculations using ERA5 data and use these results throughout the revised manuscript. Additionally, to help address Reviewer #2's comment, *"What is new?"*, we have followed the recommendation of the Editor and incorporated observational results from ERA5 into our revised versions of original Figures 11 and 14.

ERA5 provides the zonal mean of the quantity characterized as *"Mean eastward wind*

*tendency due to parameterizations"*. This quantity includes not only non-orographic gravity waves, but also other parameterized forcing. Therefore, we cannot include reanalysis results in Fig. 12 but we can include them in Fig. 14 to illustrate resolved and parameterized forcing. We have included explanations and a note of caution regarding this distinction in the revised manuscript.

**Reply to anonymous referee #1**

*Review of the article QBOi El Nino Southern Oscillation experiments Part I: Overview of experiment design and ENSO modulation of the QBO by Yoshio Kawatani et al.*

*General comments*

*This work is a continuation of the publication series produced by the QBOi project, based on an experimental protocol and several models known to simulate the QBO. This work is focused on the El Niño/La Niña effects on the QBO as simulated in 9 models. Most of the article describes the common and different features found in the different model simulations, and its structure follows the work of Kawatani et al. (2019), hereafter K2019, where they investigated the El Niño/La Niña effects on the QBO in MIROC models.*

*The experimental design chosen here is however deviating from that of K2019. Here they decided to use amplified mean El Niño/La Niña SST anomalies. This makes any direct comparison to K2019 and to observations difficult. Whether or not the QBO response should be linear to the amplitude of the SST anomaly pattern is not discussed. Probably it would have been better to use the same anomalies as in K2019. (Or an entirely different design based on SST fields of selected El Niño and La Niña years.)*

Please refer to our general discussion point (II) for this response. We have included additional explanations regarding the comparison between the ENSO SST anomalies in K2019 and the present study in the revised manuscript.

*They find that El Niño/La Niña effects on the QBO period are qualitatively similar with respect to the period, with El Niño leading to a shorter period despite of the increased tropical upwelling in the tropical lower stratosphere, from which it is clear that El Niño must also produces a stronger wave mean flow interaction. No common response is found for the QBO amplitude.*

*An interesting part is the discussion and analysis of the reasons for the described results: The more equatorial precipitation and the weaker Walker circulation found during El Niño conditions. These features are found in all models, and they probably are independent of the skill of a model to simulate a QBO. The discussion of the wave mean flow processes is however more complicated, because of the rather different ways that resolved and parameterized waves generate the QBOs in the different models. And therefore not so much can be learned from this part, except that there exists still a*

*considerable difference in the way that models generate QBOs. Further, as acknowledged by the authors, more detailed model diagnostics would be needed to learn more about the underlying reasons for the found behaviours. But this additional diagnostics was not part of the protocol, or the modelling groups could not produce these diagnostics.*

We briefly mentioned the datasets used in the present analysis and referenced Butchart et al. (2018). As noted in Butchart et al. (2018), 6-hourly data on temperature, as well as zonal, meridional, and vertical wind, are also available in the QBOi-ENSO experiments. We have provided a more detailed explanation in Section 2 of the revised manuscript, indicating that "*The requested spatial and temporal resolution and output period (e.g., monthly, daily, 6-hourly three- or two-dimensional data) align with those outlined in Butchart et al. (2018)*".  For further information on this point refer to our general discussion point (I) above.

*Overall I think that the publication is worthwhile, as it creates a baseline for further work on this topic. Some minor corrections are needed before publication.*

We sincerely appreciate the Reviewer's positive evaluation and valuable comments.

*Detailed comments and questions*

***Abstract***
*L40 Stratosphere-troposphere Processes And their Role in Climate (SPARC) …*
*As we know QBOi has been started as a SPARC project. But SPARC has changed its name to APARC and QBOi is now listed as an APARC project. Maybe it is worth to add a remark or a footnote on this aspect.*

In our revised abstract, we now refer to the "Atmosphere Processes And their Role in Climate (APARC) Quasi-Biennial Oscillation initiative (QBOi) project," while the relationship between SPARC and APARC is now briefly explained in Section 1.

*L45 … models -models … should probably be … models. Models …*

We have fixed this.

***1 Introduction***

*L64 ... that QBO facilitates ... → ... that the QBO facilitates ...*

We have fixed this.

*L140 ... Conducting a common ENSO-QBO experiment across a range of QBO-resolved climate models could help elucidate the role of non-orographic GWP in driving the oscillation. ...*
*The work of Richter et al. (2020) on the climate warming effects on the QBO unfortunately showed that the differences between GWPs are considerable and probably responsible for the rather different QBO responses to the warming. As it seems it was not possible to decide which GWPs were "wrong" or "right". Now a similar exercise is presented aiming at El Niño/La Niña variations in SST as the external forcing instead of a warmer SST and increased atmospheric CO2. Why should we expect a scientifically more robust result if Richter et al. (2020) have shown that differences in parameterizing non-orographic gravity drag can lead to very different results? Simply because El Niño/La Niña cycles exist in the historical period for which the models have been tuned?*

We acknowledge the Reviewer' point regarding the challenges in fully elucidating the role of non-orographic GWP. While Richter et al. (2020) examined QBO modulation in a future climate—where direct observational data to validate changes are unavailable—our QBOi-ENSO experiments can be partially validated using existing observations, such as the observed shortening of QBO periods during El Niño compared to La Niña. Although our experimental design is somewhat idealized, it allows us to identify models that produce longer QBO periods during El Niño runs as potentially problematic, prompting a closer evaluation of their GWP parameterizations. We believe this is a key advantage of the QBOi-ENSO experiments over the future climate scenario examined in Richter et al. (2020). Based on the Reviewer's feedback, we have revised the manuscript to clarify this reasoning. The sentence below was added in the Section 1.

"*While previous work (e.g., Richter et al., 2020) has highlighted large inter-model differences in QBO responses to climate warming scenarios due to divergent representations of non-orographic GWP, such future scenarios lack observational benchmarks. In contrast, the QBOi-ENSO experiments are informed by well-documented observational evidence, particularly the observed shortening of QBO periods during El Niño compared to La Niña (e.g., Taguchi, 2010; Yuan et al., 2014). This allows at least partial validation of model behavior. In this sense, the ENSO-focused experiments offer a scientifically more tractable approach to evaluating model GWP compared to warming*

*scenario experiments.*"

*2. Model Description and Experimental Design*

*L179 ... These factors bring the peak composite anomaly SSTs closer to the anomalies observed during the most intense El Niño and La Niña events. ...*
*Using amplification factors is problematic. This makes a comparison to observations or to the work by K2019 difficult. It seems necessary to add some remarks about the linearity between the SST pattern amplitude and the response of the QBO. Can this be assumed? Alternatively you could have chosen specific years with strong El Niño and La Niña SST anomalies. Then there would be no need to amplify the SST anomaly, and there would be less of a risk to construct an SST anomaly pattern that mixes the different types of El Niños, which are discussed in literature.*

Please refer to our general discussion point (II) above.

*L199 ... For clarity and conciseness, we will refer to these models as CESM1, EC-EARTH, ECHAM, EMAC, GISS, LMDz, MIROC-AGCM, MIROC-ESM and MRICESM1, respectively. ...*
*The abbreviated model names are introduced here, but not used consequently. Tables, Figures, and also some sentences use the full model names. Please decide whether short names shall be used or not. But if you decide to use short names, then please use these in all places: tables, figures, and text.*

We appreciate this helpful suggestion and have ensured that the revised manuscript uses a consistent system of model names. In the revised version, we include the following explanation in Section 2:
"*For clarity and conciseness, we refer to these models in the text as CESM1, EC-EARTH, ECHAM, EMAC, GISS, LMDz, MIROC-AGCM, MIROC-ESM, and MRI, respectively. The original model names are retained in figures and tables*"
This approach was chosen to improve clarity for readers who may consult figures and tables independently of the main text.

*L203 ... Launch levels for parameterized gravity waves varied across models, ranging from 450 to 700 hPa or 1000 to 100 hPa. ...*
*To which model(s) do the two pressure ranges relate? Please clarify.*

Table 1 shows the launch levels. We have added "(see Table 1)" in the revised manuscript.

*Table 1. lunched level → launch level*

We have fixed this.

*Table 2. What does the entry for GISS-E2-2G and Residual stream function (5-1115✓) mean?*

We have fixed this.

*L243 ... from the ERA-Interim (ERA-I; Dee et al. 2011) reanalyses ...*
*Why is ERA-I used for this comparison, when ERA-5 is now available? Newer reanalyses are generally improved compared to earlier ones.*

Please refer to our general discussion point (V) for this response. We have used ERA5 for all relevant calculations in the revised manuscript.

*L247 ... Importantly, the composite ERA-I and CMAP data were not scaled ...*
*This is a kind of a flaw in the experimental design. If the response to the SST anomaly patterns is non-linear to the amplitude, then the applied scaling is hindering a direct comparison to observations or analyses. If, however the signal is linear, then the signals derived from ERA-I should be scaled like the SST patterns used for the simulations.*

Please refer to general discussion points (II) and (IV) for this response.

**3. ENSO Modulation of the QBO and Climatological Mean Field Differences**

*L266 – L276 These lines discuss deficiencies in the structure of the simulated QBO, as occurring in El Niño or La Niña simulations of ECHAM, GISS, and LMDz. In my opinion it is necessary to point out another deficiency, which is an unrealistic period, although a regular pattern of downward propagating westerly and easterly jets is simulated. Taking the displayed 20 years (Fig. 2) of the El Niño and La Niña simulations together, we have 40 years for which on average (40years / 28 months) we would expect about 17 cycles. A count of the cycles shown in Figure 2 can now serve as an additional measure for the quality of the simulations. If we allowed a range of 15 to 19, then the following models*

*(here excluding ECHAM, GISS and LMDz) would fail: EC-EARTH: 20, MIROC-AGCM-LL: 26, MIROC-ESM: 21. Please extend your discussion of problematic simulations in this direction, so that the reader knows from the beginning which model simulations need to be viewed a bit more critically.*

We appreciate the reviewer's comment highlighting the importance of assessing whether the simulated QBO periods in each model are consistent with observations. While a previous evaluation of QBO periods in QBOi "Experiment 2" (which used climatological SSTs) was reported by Bushell et al. (2020), we agree that it is essential to address QBO period realism directly in the context of the QBOi-ENSO simulations. Accordingly, the following sentence has been added to the explanation of Figs. 2 and 3:

"*The mean QBO period differs among models, and some simulate QBO periods that fall notably outside the observed range, which has a mean of approximately 28 months and varies from 18 to 34 months (Baldwin et al., 2001; Anstey et al., 2021). In particular, MIROC-AGCM (16.6 to 19.7 months for El Niño and La Niña means) and MIROC-ESM (22.5 to 24.9 months) exhibit systematically shorter periods than observed. The mean QBO periods in both El Niño and La Niña runs for EMAC and GISS are also somewhat shorter than ~28 months. While these models reproduce realistic downward propagation of QBO phases (Fig. 2), the short periodicity constitutes a structural limitation that should be taken into account when interpreting their simulation results. Nevertheless, the primary focus of this study is on the relative differences in QBO characteristics between El Niño and La Niña conditions within each model, rather than on absolute agreement with observed QBO behavior. Accordingly, even models with biases in mean QBO period can still provide meaningful insights into the modulation of the QBO if they produce internally consistent and interpretable differences between the two ENSO phases.*"

We also add the sentence below just after discussion of Fig. 3:

"*In this context, it is also worth noting that a comprehensive evaluation of QBO period characteristics across multiple climate models participating in the QBOi project was conducted by Bushell et al. (2020). That study analyzed QBO periods in both QBOi Experiment 1 (AMIP-type simulations with observed SSTs) and Experiment 2 (simulations with climatological SSTs). Their results provide a broader reference for understanding how model formulation and boundary conditions influence simulated QBO periodicity. Readers interested in the model-dependent behavior of QBO periods across these*

*different experimental designs are encouraged to consult Bushell et al. (2020) for further context*"

This addition is intended to help readers more critically assess which model simulations may require caution when interpreting the results.

Bushell, A. C., J. A. Anstey, N. Butchart, Y. Kawatani, S. M. Osprey, J. H. Richter, F. Serva, P. Braesicke, C. Cagnazzo, C.-C. Chen, H.-Y. Chun, R. R. Garcia, L. J. Gray, K. Hamilton, T. Kerzenmacher, Y.-H. Kim, F. Lott, C. McLandress, H. Naoe, J. Scinocca, T. N. Stockdale, S. Watanabe, K. Yoshida, S. Yukimoto: Evaluation of the Quasi‐Biennial Oscillation in global climate models for the SPARC QBO‐initiative, Quarterly Journal of the Royal Meteorological Society, https://doi.org/10.1002/qj.3765, 2020

*L311 ... Next, we consider ENSO modulation of QBO amplitude, which is known less robust ... → ... Next, we consider the ENSO modulation of the QBO amplitude, which is known to be less robust ...*

We have fixed this.

*L323 ... GISS, LMDz, and CESM1, all of which have variable GWP sources. ...*
*I think it should be added that MIROC-AGCM-LL has variable gravity waves too, though these are explicitly simulated, within the given resolution, instead of parameterized. Thus variability of gravity waves not necessarily leads to a strong amplitude difference between El Niño and La Niña. And one needs to wonder if a strong change is indicating that the variability of gravity waves is important aspect for a GWP, or whether this effect is rather a result of other aspects of parameterizing gravity wave. Please add some thoughts on this problem.*

While MIROC-AGCM-LL can simulate a QBO-like oscillation without parameterized non-orographic GWP, previous studies have indicated that its resolution (T106L72, corresponding to a 1.25-degree horizontal resolution and 550-meter vertical resolution) is insufficient to capture wave forcing as effectively as higher-resolution models, such as T213L256 or even higher-resolution models (e.g., Kawatani et al., 2010). In MIROC-AGCM-LL, most of the unresolved gravity wave forcing that is parameterized in other models is not explicitly simulated. Therefore, we believe that the MIROC-AGCM-LL results do not necessarily support the conclusion that 'variability of gravity waves does

not necessarily lead to a strong amplitude difference between El Niño and La Niña.'

Parameterized GWP represents sub-grid-scale processes on much smaller scales than ~100 km. The variable source of parameterized GWP is often linked to cumulus convection, which also generates gravity waves on much smaller scales than those resolved in MIROC-AGCM-LL. We added short explanation below in the revised manuscript.

"*Due to its limited resolution (T106L72), MIROC-AGCM cannot fully capture the high-frequency, small-scale gravity wave spectrum that is typically represented by parameterized GWP schemes. As a result, much of the gravity wave that would influence the QBO remains unresolved*"

Kawatani, Y., K. Sato, T. J. Dunkerton, S. Watanabe, S. Miyahara and M. Takahashi, 2010: The roles of equatorial trapped waves and internal inertia-gravity waves in driving the quasi-biennial oscillation. Part I: Zonal mean wave forcing, J. Atmos. Sci., 67, 963-980., https://doi.org/10.1175/2009JAS3222.1

*L383 ... although models tend to simulate the precipitation peak to the east of the observed one over the central Pacific in the El Niño run. ...*
*It should also be mentioned that the precipitation peak in the model simulations is higher than in observations, which indicates that the local forcing by latent heat release in the simulations is higher than that explained by the observed precipitation. Quite likely this is related to the amplified El Niño/La Niña SST strength.*

We acknowledge the Reviewer's point that the precipitation peak in the model simulations is higher than in observations. It is important to note that the observed precipitation data used in our analysis represents simple averages over all El Niño and La Niña months, first at each calendar month and then annually, reflecting precipitation patterns during "moderate" ENSO events. We have added explanation here in the revised manuscript as follows: "The magnitude of the precipitation peak is also generally larger in the models than in observations, which may reflect the amplified SST anomalies used in the simulations".

*L422 ... significantly deep westerly difference ... → ... significantly deeper westerly difference ...*

This sentence has been moved to the supplementary materials in the revised manuscript, and we have fixed this.

This figure has been moved to the supplementary materials in the revised manuscript, and we have fixed this.

**4 Contrasting wave forcing and residual mean meridional circulations in El Niño and La Niña from QBOi models**

*L458 … The X term represents any other unresolved forcing. … Do you mean here parameterized momentum diffusion and effects from numerical diffusion and damping operators?*

Indeed, this represents all the other possible contributions including explicitly parameterized diffusion and any other contributions from the numerical schemes employed. We changed the sentence in the revised manuscript as follows:
"*The $\bar{X}$ term represents any other unresolved forcing including explicitly parameterized diffusion and any other contributions from the numerical schemes employed.*"

*L470 … in El Niño and La Niña simulations. …*
*La Niña simulations are nor shown, but differences of the El Niño and La Niña simulations.*

This sentence is moved to the supplementary materials, and we have changed the text here to correct this as follows:
"*Figure 8 of Kawatani et al. (2019) showed the EP flux vectors for El Niño conditions, along with the differences between El Niño and La Niña simulations.*"

This figure has been moved to the supplementary material. We have changed the text as suggested

*L498 ... which both use variable sources in their GWP, ...*
*Do you mention this because you think that this is the reason for the differences? Often other differences in the formulation of the non-orographic gravity wave drag parameterizations can cause substantial differences already.*

Indeed we discussed the variable GWP sources in these models based on what was visually apparent in the figure, without directly attributing them as the sole cause of the observed differences. We agree with the Reviewer's assessment that other differences in the formulation of non-orographic gravity wave drag parameterizations could also contribute substantially to these differences. While it is difficult to definitively determine the specific reasons at this stage, we simply removed the relevant sentence from the main text. here.

*L509 ... averaged over 20°S–20°N ...*
*Maybe it is worth to explain why a band of 20°S – 20°N is chosen, while earlier diagnostics/figures used narrower bands. (I guess this is made in order to remove residuals of the secondary meridional circulation of the QBO.)*

We selected the 20°S–20°N latitude band for averaging because this region encompasses the area where the QBO amplitude is most pronounced. While other latitude bands, such as 10°S–10°N or 15°S–15°N, could also be used, they tend to exhibit more noise. However, we have confirmed that our basic conclusions remain largely unaffected by the choice of these latitudinal ranges. In the revised manuscript, we have added a brief explanation as follows:

"*This latitude band was chosen to reduce noise from the secondary meridional circulation associated with the QBO. The main conclusions are not sensitive to the choice of meridional averaging width*"

*L510 ... ranging from approximately 0.2 mm s¹ in MIROC-AGCM to approximately 0.4 mm s¹ ⁻in LMDz. ...*
*This strong difference in the tropical upwelling implies also a strong difference in the strength of wave mean flow interaction that is necessary to simulate a QBO with a realistic period. This aspect is not discussed here, and maybe this El Niño/La Niña related article is the wrong place. Still it directly shows that the wave mean flow interaction must work at different strengths.*

We agree that the substantial inter-model spread in tropical upwelling implies differences in the strength of wave–mean flow interaction, which is essential for simulating a realistic QBO period. Although a detailed discussion of this aspect is beyond the scope of this ENSO-focused study, we consider this an important point and have added a brief mention of it in the revised manuscript. Weak tropical upwelling in the MIROC model has been discussed in Kawatani et al. (2010). In addition, differences in tropical upwelling among reanalysis datasets have been reported in the S-RIP final report (SPARC, 2022).

We have added one sentence here as follows:

"*This inter-model spread in tropical upwelling may reflect differences in the strength of wave–mean flow interaction, which is critical for simulating the QBO with a realistic period (e.g., Kawatani et al., 2010). While a detailed examination of this aspect is beyond the scope of the present study, it may partly explain model differences in QBO characteristics*"

SPARC, 2022: SPARC Reanalysis Intercomparison Project (S-RIP) Final Report. Masatomo Fujiwara, Gloria L. Manney, Lesley J. Gray, and Jonathon S. Wright (Eds.), SPARC Report No. 10, WCRP-6/2021, doi: 10.17874/800dee57d13, available at www.sparc-climate.org/publications/sparc-reports.

*L522 ... However, the specific altitudes at which $w*$ changes would most strongly influence the overall QBO period remain unclear. ...*
*Sentence unclear.*

We have modified this sentence to read:
"*However, it remains unclear which specific altitudes of $\overline{w}^*$ change have the strongest influence on the overall QBO period*"

*L579 ... While output data of parameterized gravity wave fluxes in LMDz were not available at the time of this analysis, this model, which also uses variable parameterized wave sources related to precipitation activity, showed similar structures affected by precipitation distributions (Dr. Lott, personal communication). ...*
*Francois Lott is a co-author of this study. Please include the LMDz results in Figure 12.*

Francois Lott informed us that the datasets were quantitatively incorrect due to inadequate processing. However, we have confirmed that the qualitative distribution is related to

precipitation, similar to what is observed in CESM1.

*L620 … parameterized wave forcing below is stronger …*
*What does "below" refer to? Maybe the sentence needs to be rephrased.*

We delete "below" in this sentence.

*L639 … As discussed for Fig. 14, … → … As discussed for Fig. 13, …*

We have fixed this.

*5. Summary and concluding remarks*

*L685 … remained consistent …*
*"consistent" seems to be the wrong term, because this could have different meanings. (If El Niño/La Niña influences the ozone distribution, then the same ozone field cannot be consistent with El Niño and La Niña conditions at the same time.) "unchanged" would express more clearly that these fields simply have not been changed.*

We have changed the wording here from "consistent" to "unchanged" following the Reviewer's suggestion.

**Reply to anonymous Reviewer #2**

*Review of the manuscript "QBOi El Nino Southern Oscillation experiments Part I: Overview of experiment design and ENSO modulation of the QBO" by Kawatani et al.*

*Summary*

*The manuscript presents an overview of the experimental design from the new SPARC QBOi project and examines the modulation of the QBO by ENSO using nine climate models. The findings indicate that the QBO period is longer during La Nina compared to El Nino across all models, consistent with observations. However, changes in the QBO amplitude remain inconclusive. Overall, I find the experiment intriguing, the manuscript well-written, and the results clearly explained. Most of my comments are minor and focus on improving consistency between different parts of the manuscript and aligning the figures with the text.*

We sincerely appreciate the Reviewer's positive evaluation and valuable comments.

*One major comment, however, concerns the lack of deeper insights into ENSO modulation of the QBO. The authors attribute this to the simplicity of the analyses and the limited availability of model output data, which they suggest prevents a full explanation of the quantitative differences in QBO between El Nino and La Nina. While future studies are mentioned as a potential avenue to address this, I argue that if a more detailed analysis is feasible, it should be included in this paper, as it was the primary motivation for the experiment and study.*

Please refer to our general discussion point (I) above for a consideration of the role of this paper within the broader QBOi program which helps motivate our work. Our revisions have involved significantly more analysis. In particular, following the Editor's suggestion we have repeated the more detailed analysis of Taguchi (2010) to elucidate the seasonal effects in the QBO-ENSO connections. We also have expanded our study to include analysis of the earlier AMIP runs (QBOi "Experiment 1") to supplement the new "annually repeating" runs that represented the main focus of our original manuscript.

*Major Comments*

*1) Lack of additional insight into the mechanisms of ENSO modulation of the QBO*

*My primary concern is whether this paper and the associated experiments provide any additional insight into the mechanisms by which ENSO modulates the QBO. At the start of the paper, I had hoped—likely in line with the motivation behind designing and implementing these experiments—that this study would offer a deeper understanding of these mechanisms. However, the study appears to be an extension of Kawatani et al. (2019), with potentially more models included beyond MIROC, yet missing important analyses due to data limitations.*

*There are repeated statements such as: "Further investigation of these models is hampered by the incomplete model variables in the available data sets", "This simple analysis with limited model output data cannot fully explain quantitative differences in QBO periods between El Nino and La Nina", and "Detailed zonal-time spectral analyses of model fields, like those performed in Kawatani et al. (2019), remain a subject for future study."*

*If such analyses are indeed possible, this paper is the appropriate venue to present them, rather than postponing them to future studies. For example, as the authors mentioned, detailed spectral analyses of the EP flux, gravity wave parameterization fluxes, precipitation, or momentum budgets based on the TEM framework could offer crucial insights into the intermodel spread of QBO period and amplitude.*

*To provide further context on my disappointment, Kawatani et al. (2019) noted: "It would be interesting to analyze the ENSO modulation of the three-dimensional wave forcing as well as tropical upwelling, which must show large differences between El Nino and La Nina. This may be investigated in a future study". Now, five years later, this study states: "A detailed investigation of the three-dimensional distributions of parameterized wave fluxes modulated by ENSO, including model dependence, would be of interest and remains a topic for future research."*

*It feels like an opportunity has been missed to address these outstanding questions. If there is a way to conduct these analyses, I strongly encourage the authors to include them in this paper.*

We do believe that our manuscript already contains a substantial amount of useful material and it provides the background and introduction for related QBOi studies; please

refer to our general discussion point (I) above. However, we understand the Reviewer's concern and have included further analysis of the simulations following the Editor's suggestion, as explained in our general discussion point (III) above, as well as a substantial expansion of our study to include analysis of earlier AMIP runs as explained in general discussion point (IV).

*2) Lack of use of recent data and citations of recent studies*
*Some aspects of the study, including citations and the data used, feel somewhat outdated. For instance, the use of ERA-I instead of ERA5. Additionally, the study only uses observed data up to 2012. If this limitation is due to avoiding the QBO disruptions, there are still several years of data available between 2012 and the end of 2015, as well as between 2020 and 2024. While including these additional years may not change the main conclusions of the paper, it would enhance the robustness of the analysis, particularly for slowly evolving phenomena like ENSO and QBO, where even a few more samples could provide valuable insights. Moreover, the citations miss some relevant and recent studies, such as Zhou et al. (2024), and a few others noted in my review.*

We appreciate this concern and we have repeated all our analyses with ERA5 data as explained in general discussion point (V). We have also included the recent references that the Reviewer has suggested.

***Minor comments***
*L47-49: It can also be mentioned that "all models simulate stronger equatorial tropical upwelling in El Nino compared to La Nina up to ~10 hPa".*

We have included one sentence below in the abstract.
"*All models also simulate stronger equatorial tropical upwelling in El Niño compared to La Niña up to ~10 hPa, consistent with ERA5 reanalysis.*"

*L85: Small-scale gravity waves also contribute significantly to the forcing of the QBO westerly (e.g., Pahlavan et al. (2021))*

We have now referred to this paper.

*L95: As a good reference on this you can cite Coy et al. (2020).*

We have now referred to this paper.

*In general, the figures can be significantly improved by reducing redundancy, which would allow for larger, clearer panels. For instance, in Figure 2, use "El Nino" as the title for the left column and "La Nina" for the right, rather than repeating them for each panel. Similarly, list model names only on the left side of the figure and show the y-axis (0–20) only on the bottom panels, instead of repeating it in every panel. These changes can enhance readability and apply to other figures as well.*

As suggested by the Reviewer we have fixed these issues in the revised manuscript.

*The other general issue with the figures is the presence of too many contours, which reduces readability. In particular, in Figures 4, 8, 9, 10, and 13, the contours over the shadings can be removed, similar to Figure 12, to improve clarity.*

In our revised version we have adopted these suggestions to improve the legibility of the figures, for example by using thinner contour lines. In original Figures 4, 8, 9, and 10, we retained the contours over the shading to indicate the sign and structure of the anomalies, as they help distinguish positive and negative values in the broader spatial patterns. However, the contour thickness and density have been carefully adjusted to improve visual clarity while preserving essential information.

*Figure 3: Have you analyzed each phase of the QBO separately? For example, do both phases of the QBO become shorter during El Nino?*

This is an excellent suggestion and was also made by the Editor. We have now repeated the analysis segregated by QBO phase and season and have included these results in Section 3 (new Figs. 5 and 6) in the revised version. Please refer to general discussion point (III) for details.

*Figure 3: Have you considered using a Fourier Transform to determine the period instead of relying on zero wind line crossing (e.g., as done in Lee et al. (2024))? While it likely won't change the conclusions, it might be a better option, particularly when the QBO becomes more irregular/unrealistic, as seen during El Nino in ECHAM.*

We appreciate the suggestion to use a Fourier Transform to determine the QBO period,

as demonstrated in Lee et al. (2024). We agree that FFT methods are particularly useful when defining the phase transition of the QBO is challenging (we employed FFT methods in our QBOi phase-1 paper, Richter et al. 2020). However, in the current QBOi-ENSO experiments, all models have a clearly defined QBO phase transition at 20 hPa. Therefore, we prefer to determine the QBO period by phase transition, following the approach of K2019. As the Reviewer noted, this choice is unlikely to alter the overall conclusions of our study.

*L412: Will the cooler anomaly around 60°N–90°N in ERA-I, which is not observed in the models, change if more data is included, such as using ERA-5 from 1940 to 2024?*

We find this even in ERA-5 from 1979 to 2022, as we have shown in the revised manuscript. Of course, ERA5 covers the period from 1950, but here we use data only beginning in 1979.

*For Figures 10, 11, 13, and 14, you could consider including results from reanalysis (e.g., ERA5) as a reference, similar to what is done in Figures 8 and 9.*

Once again refer to our general discussion point (V) for ERA5. ERA5 data were used to produce the revised versions corresponding to the original Figures 10, 11, and 14. However, we did not include reanalysis results in the revision of the original Figure 13, which presents QBO composites of zonal wave forcing, because it is not feasible to clearly separate El Niño and La Niña phases in a ±18-month composite using real-world data. The main purpose of Figure 13 is to compare the relative contributions of resolved and parameterized wave forcing during El Niño and La Niña. Therefore, the absence of results based on reanalyses does not affect the conclusions drawn from this figure.

*Figure 12: Could you add the total flux for El Nino and La Nina (i.e., averaged over 10°S–10°N and all longitudes) to the bottom panels? If so, is it consistently larger during El Nino?*

We thank the reviewer for this constructive suggestion. In response, we calculated the total (eastward plus westward) momentum fluxes at 100 hPa, averaged over 10°S–10°N and all longitudes, for both El Niño and La Niña runs. The results show that CESM1 exhibits a significantly larger total flux during El Niño. This is consistent with its use of variable non-orographic gravity wave sources related to convective activity. On the other

hand, the other three models (ECHAM5sh, MIROC-ESM, and MRI-ESM2.0), which employ Hines-type gravity wave parameterizations with fixed sources, show no significant differences in the total flux between the two ENSO phases. In the revision we have added this sentence:

"*When averaged over 10°S–10°N and all longitudes, the total (eastward plus westward) momentum flux at 100 hPa is significantly larger during El Niño only in CESM1, which uses variable non-orographic gravity wave sources. In contrast, the other three models with Hines-type schemes and fixed wave sources (ECHAM, MIROC-ESM, and MRI) do not show significant differences in total flux between El Niño and La Niña conditions*"

*Editorial comments*
*L84: "respectively" seems redundant.*

We agree and have fixed this in the revised version.

*L107: You can cite (Richter et al., 2020) again to avoid ambiguity.*

We have done this in the revised version.

*L108: SST is not yet defined.*

We have fixed this.

*L140: "QBO-resolved" -> "QBO-resolving"*

We have fixed this.

*L164: GWP is already defined.*

We have fixed this.

*L165: What is experiment 2?*

We add the explanation here as follows:
"*Phase-1 of QBOi consists of five experiments. Experiment 1 is an AMIP-type simulation*

*using observed sea surface temperatures (SSTs) for 1979–2009. Experiment 2 employs climatological annual cycles of SSTs, sea ice, and external forcings, while Experiments 3 and 4 explore global warming scenarios. Experiment 5 consists of seasonal prediction experiments with perturbed initial conditions*"

*L165: SST needs to be defined at L108.*

We have fixed this.

*L198: The model name "CESM15-110L" is mentioned here, while "WACCM5-110L" is used in the results (figures and tables). I suggest selecting one naming convention for consistency.*
*L200: Using the concise version of model names is a great choice, but it would be helpful to maintain this approach consistently in the results (figures and tables) as well. Currently, there is a discrepancy where the text uses concise names while the results use the full model names, making it harder to follow.*

As we mentioned in our response to the other reviewer, we have now adopted a consistent naming convention for the models throughout our revised manuscript.

*L220: Palmerio et al. (2022) is not in the bibliography.*

We have fixed this.

*L238: TEM is already defined.*

We have fixed this.

*Table 2: what is "5-1115" in front of GISS.*

This is our mistake. We have fixed this.

*L284: "larger" -> "longer"(?)*

We have fixed this.

*L295: "with" -> "for"*

We have fixed this.

*L375: ITCZ not defined yet.*

We have fixed this.

*L378: "(left-top)" -> "(center-top)"*

We have fixed this.

*L392: Any reference for this statement?*

In our revision we have included a reference to Kawatani et al. (2009) here.

*Figure 7: "PRCP" not defined.*

 In the revised version "PRCP" no longer appears in the figure label.

*L427: BDC is already defined.*

In the revision this sentence has been moved to the supplementary material.

*L431: "(left-top)" -> "(center-top)"*

We have fixed this.

*L442: A point after 4 is missing. "In El Nino and La Nina from QBOi models" is redundant. Also, capitalize the first letter to maintain consistency with the other titles.*

We have fixed this.

*L443: "eddy forcing" -> "wave forcing", to be consistent with the other sections.*

We have fixed this.

*L443: "mean zonal" -> "zonal mean"*

We have fixed this.

*L444: TEM is already defined.*

We have fixed this.

*L445: "eddies" -> "waves"*

We have fixed this.

*L566-567: "WACCM" -> CESM1*

We have fixed this.

*L577: ITCZ should be defined earlier at L375.*

We have fixed this.

*L620: "below" should be removed.*

We have fixed this.

*L621: "below" -> "above" (?)*

Yes we want to say "above" here.

*L639: "Fig. 14" -> "Fig. 13"*

The reference to Fig. 14 in the original manuscript is what we intended.

*Caption of Fig. 14: "eddy" -> "wave". "resolved motions" -> "resolved forcing".*

We have fixed this.

**References**

*Coy, L., Newman, P. A., Strahan, S., & Pawson, S. (2020). Seasonal Variation of the Quasi-Biennial Oscillation Descent. Journal of Geophysical Research: Atmospheres, 125(18), e2020JD033077.https://doi.org/10.1029/2020JD033077*

*Geller, M. A., Zhou, T., & Yuan, W. (2016). The QBO, gravity waves forced by tropical convection, and ENSO. Journal of Geophysical Research: Atmospheres, 121(15), 8886–8895. https://doi.org/10.1002/2015JD024125*

*Kawatani, Y., Hamilton, K., Sato, K., Dunkerton, T. J., Watanabe, S., & Kikuchi, K. (2019). ENSO Modulation of the QBO: Results from MIROC Models with and without Nonorographic Gravity Wave Parameterization. https://doi.org/10.1175/JAS-D-19-0163.1*

*Lee, H.-K., Chun, H.-Y., Richter, J. H., Simpson, I. R., & Garcia, R. R. (2024). Contributions of Parameterized Gravity Waves and Resolved Equatorial Waves to the QBO Period in a Future Climate of CESM2. Journal of Geophysical Research: Atmospheres, 129(8), e2024JD040744. https://doi.org/10.1029/2024JD040744*

*Pahlavan, H. A., Wallace, J. M., Fu, Q., & Kiladis, G. N. (2021). Revisiting the Quasi-Biennial Oscillation as Seen in ERA5. Part II: Evaluation of Waves and Wave Forcing. Journal of the Atmospheric Sciences, 78(3), 693–707. https://doi.org/10.1175/JAS-D-20-0249.1*

*Richter, J. H., Butchart, N., Kawatani, Y., Bushell, A. C., Holt, L., Serva, F., et al. (2020). Response of the Quasi-Biennial Oscillation to a warming climate in global climate models. Quarterly Journal of the Royal Meteorological Society, n/a(n/a). https://doi.org/10.1002/qj.3749*

*Zhou, T., DallaSanta, K. J., Orbe, C., Rind, D. H., Jonas, J. A., Nazarenko, L., et al. (2024). Exploring the ENSO modulation of the QBO periods with GISS E2.2 models. Atmospheric Chemistry and Physics, 24(1), 509–532. https://doi.org/10.5194/acp-24-509-2024*

We appreciate these relevant recent references. We have referred them in the revised manuscript.

---

## Editor Decision (ED1)

WCD decision EGUSPHERE-2024-3270 (Kawatani et al)

Dear Dr. Kawatani and co-authors:

Thank you for the considerable work you and your coauthors have done to revise your paper. I was very happy to see that you addressed almost all of the issues raised by the two anonymous reviewers, with which I concurred. In particular, including the quantitative analysis of the sensitivity of the QBO period and amplitude to the phase of ENSO adds substantively to the manuscript. My decision on the revised manuscript is "Publish subject to minor revisions", after which I will make a final decision to accept the paper for publication.

My recommendations for final edits are as follows (NOTE: All line numbers refer to the document with the Author Tracked Changes):

i)     Edit to bring the abstract more in line with the results of the paper, and remove extraneous information. Suggest the following change on lines 55-56: "…simulated La Nina periods tend to be longer than those observed during El Nino, although in most models the differences are small compared to that observed." This wording is more in line with the summary of the results in the text (e.g., line 345-6 "Only three of the nine models …simulate La Nina-El Nino differences in the QBO period that approach this observed sensitivity, even under the amplified ENSO forcing used in this study.", and the text on lines 917-919 "It is noted … the QBO period.")

ii)    Also in the Abstract, delete the sentence on lines 59-61 ("The models capture … wind and temperature."). As I said in my review of the original manuscript, all models capture the equatorial tropospheric anomalies associated with ENSO for at least the past 35 years. So this statement adds nothing to the manuscript (other than prompting the reader to ask "why are they surprised at this result?").

iii)   Adsf

iv)    A reference is required on the statement that ends on line 155.

v)     Line 216: "… representing the upper end of past variability…" refers to a vertical coordinate on "end". Change this to read "…representing the extreme of past variability."

vi)    Delete the paragraph starting on line 365. This point is made more succinctly by appending a short qualifier to line 365: "as seen in Fig. 2. For further information on how model formulation and boundary conditions influence the simulated QBO, see Bushell et al 2020."

vii)   Line 458, the word "uniform" is vague. Please elaborate.

viii) In reference to Fig. 11 (the original Fig. 12), Reviewer 2 asked about the total EP flux in the El Nino and La Nina simulations. You indicated in response to the reviewer that a (nicely worded) new paragraph was added to the revised text ("When averaged over … the total … is significantly larger during El Nino only in CESM1. In contrast, … between El Nino and La Nina conditions."). I couldn't find this text. If it isn't in the paper, please add it.

ix) Lines 715-end of paragraph. It should be noted that, unlike in all of the models, observations show that the easterly acceleration in the QBO is by resolved (K <= 20) waves. See Fig. 12 of Pahlavan et al.

x) Delete the gratuitous one-sentence paragraph on lines 831-832; it doesn't add any specific information.

xi) Move the text on lines 917-919 to line 853, so line 853 reads "… basic agreement with observations. However, only three of the nine models (EC-EARTH, LMDz, and ECHAM) simulate La Niña–El Niño differences in QBO period that approach the observed sensitivity (~27 %), even under the amplified ENSO forcing. The remaining six models exhibit more modest ENSO modulation 920 of the QBO period."

xii) Finally, in several places in the text, the results may be reported with too many significant digits. For example, one line 428. Let's say there are perhaps 40 QBO cycles in a 100-year simulation. The mean period will have some uncertainty, compounded by taking the difference in the period. I can't imagine there are more than two significant digits in the result, but please check. Similar comments apply to the quoted numbers on lines 222, 340-341

---

## Author Response (AR2)

**Authors' response to Editor's comments on "QBOi El Niño Southern Oscillation experiments: Overview of experiment design and ENSO modulation of the QBO" by Y. Kawatani et al.**

We very much appreciate the Editor's extensive efforts in carefully reading of the revised manuscript and making suggestions for improvement. Our responses to each of the Editor's comments are included below. For clarity, we have included the Editor's comments in *blue italics* and our responses in regular font.

*Thank you for the considerable work you and your coauthors have done to revise your paper. I was very happy to see that you addressed almost all of the issues raised by the two anonymous reviewers, with which I concurred. In particular, including the quantitative analysis of the sensitivity of the QBO period and amplitude to the phase of ENSO adds substantively to the manuscript. My decision on the revised manuscript is "publish subject to minor revisions", after which I will make a final decision to accept the paper for publication.*

We appreciate the Editor's positive assessment of our revised manuscript. We carefully considered each of the remaining comments and revised the manuscript where appropriate.

*i) Edit to bring the abstract more in line with the results of the paper, and remove extraneous information. Suggest the following change on lines 55-56: "...simulated La Nina periods tend to be longer than those observed during El Nino, although in most models the differences are small compared to that observed." This wording is more in line with the summary of the results in the text (e.g., line 345-6 "Only three of the nine models ...simulate La Nina- El Nino differences in the QBO period that approach this observed sensitivity, even under the amplified ENSO forcing used in this study.", and the text on lines 917-919 "It is noted ... the QBO period.")*

We have revised the sentence in the abstract to follow the Editor's suggestion, but slightly modified it as follows: "*The simulated QBO periods during La Niña tend to be longer than those during El Niño, although in most models the differences are small compared to that observed*".

*ii) Also in the Abstract, delete the sentence on lines 59-61 ("The models capture… wind and temperature."). As I said in my review of the original manuscript, all models capture the equatorial tropospheric anomalies associated with ENSO for at least the past 35 years. So this statement adds nothing to the manuscript (other than prompting the reader to ask "why are they surprised at this result?").*

We have deleted the sentence in the abstract as suggested.

*iii) Adsf*

We assume this was a typographical error.

*iv) A reference is required on the statement that ends on line 155.*

We agree and have added references to Kawatani et al. (2010b) and Kawatani et al. (2019) here.

*v) Line 216: "… representing the upper end of past variability…" refers to a vertical coordinate on "end". Change this to read "…representing the extreme of past variability."*

We have revised the sentence as suggested.

*vi) Delete the paragraph starting on line 365. This point is made more succinctly by appending a short qualifier to line 365: "as seen in Fig. 2. For further information on how model formulation and boundary conditions influence the simulated QBO, see Bushell et al 2020."*

As suggested, we have deleted the paragraph and added the recommended sentence here.

*vii) Line 458, the word "uniform" is vague. Please elaborate.*

We have revised the sentence to clarify the intended meaning as follows:
"*These results highlight that, unlike the relatively consistent ENSO-related changes in QBO period, in which all models simulate longer periods during La Niña than during El Niño, the QBO amplitude exhibits more model-dependent and phase-dependent variations.*"

*viii) In reference to Fig. 11 (the original Fig. 12), Reviewer 2 asked about the total EP flux in the El Nino and La Nina simulations. You indicated in response to the reviewer that a (nicely worded) new paragraph was added to the revised text ("When averaged over ... the total ... is significantly larger during El Nino only in CESM1. In contrast, ... between El Nino and La Nina conditions."). I couldn't find this text. If it isn't in the paper, please add it.*

We thank the Editor for pointing this out. Indeed we had intended to include the sentence mentioned in our response to Reviewer 2, but it was inadvertently omitted. We have now inserted the following sentence into the discussion of Fig. 11 as follows:

"*In addition to these spatial patterns, when averaged over 10°S–10°N and all longitudes, the total (eastward plus westward) momentum flux at 100 hPa is significantly larger during El Niño only in CESM1, which uses variable non-orographic gravity wave sources. In contrast, the other three models with Hines-type schemes and fixed wave sources (ECHAM, MIROC-ESM, and MRI) do not show significant differences in total flux between El Niño and La Niña conditions*"

*ix) Lines 715-end of paragraph. It should be noted that, unlike in all of the models, observations show that the easterly acceleration in the QBO is by resolved (K <= 20) waves. See Fig. 12 of Pahlavan et al.*

We respectfully note that this interpretation of Pahlavan et al. (2023) may reflect a misunderstanding. As shown in their Fig. 12 and described in the text, the easterly (westward) acceleration in the QBO is primarily driven by small-scale gravity (SSG) waves with zonal wavenumbers greater than 20, while contributions from larger-scale waves ($K \leq 20$), including Kelvin, MRG, and IG modes, are relatively minor. This conclusion is consistent with findings from other observational and modeling studies, and is now widely accepted in the QBO literature.

*x) Delete the gratuitous one-sentence paragraph on lines 831-832; it doesn't add any specific information.*

We have deleted this sentence as suggested.

*xi) Move the text on lines 917-919 to line 853, so line 853 reads "… basic agreement with observations. However, only three of the nine models (ECEARTH, LMDz, and ECHAM) simulate La Niña–El Niño differences in QBO period that approach the observed sensitivity (~27 %), even under the amplified ENSO forcing. The remaining six models exhibit more modest ENSO modulation 920 of the QBO period."*

We have moved the text as suggested.

*xii) Finally, in several places in the text, the results may be reported with too many significant digits. For example, one line 428. Let's say there are perhaps 40 QBO cycles in a 100-year simulation. The mean period will have some uncertainty, compounded by taking the difference in the period. I can't imagine there are more than two significant digits in the result, but please check. Similar comments apply to the quoted numbers on lines 222, 340-341*

We changed all percentage values in the main text to integers (e.g., 28% instead of 27.9%). For QBO periods in months, we now use one decimal place (e.g., $25.3 \pm 2.1$ mon). In Figure 3, we originally showed values to two decimal places, but changed them to one decimal place for consistency. We kept this level of detail to allow reasonable comparison across models, especially where standard deviations are included.